# Beyond a Million Tokens: Benchmarking and Enhancing Long-Term Memory in LLMs

**Mohammad Tavakoli**[1], **Alireza Salemi**[2], **Carrie Ye**[1], **Mohamed Abdalla**[1],
**Hamed Zamani**[2], **J. Ross Mitchell**[1]
[1]University of Alberta    [2]University of Massachusetts Amherst
{tavakol5, cye, mabdall2, jmitche2}@ualberta.ca
{asalemi, zamani}@cs.umass.edu

## Abstract

Evaluating the abilities of large language models (LLMs) for tasks that require long-term memory and thus long-context reasoning, for example in conversational settings, is hampered by the existing benchmarks, which often lack narrative coherence, cover narrow domains, and only test simple recall-oriented tasks. This paper introduces a comprehensive solution to these challenges. First, we present a novel framework for automatically generating long (up to 10M tokens), coherent, and topically diverse conversations, accompanied by probing questions targeting a wide range of memory abilities. From this, we construct BEAM, a new benchmark comprising 100 conversations and 2,000 validated questions. Second, to enhance model performance, we propose LIGHT–a framework inspired by human cognition that equips LLMs with three complementary memory systems: a long-term episodic memory, a short-term working memory, and a scratchpad for accumulating salient facts. Our experiments on BEAM reveal that even LLMs with 1M token context windows (with and without retrieval-augmentation) struggle as dialogues lengthen. In contrast, LIGHT consistently improves performance across various models, achieving an average improvement of 3.5%–12.69% over the strongest baselines, depending on the backbone LLM. An ablation study further confirms the contribution of each memory component.

## 1 Introduction

Large language models (LLMs) have been deployed across diverse applications, including open-domain conversational agents (Laban et al., 2025; Chen et al., 2025), retrieval-augmented generation (RAG) for open-domain question answering and fact checking (Lewis et al., 2020; Salemi et al., 2025; Salemi & Zamani, 2025; Kim et al., 2024b), long-document and code analysis (Li et al., 2025; Jelodar et al., 2025; Fang et al., 2024), and scientific or legal research (Rueda et al., 2025; Nguyen et al., 2025). Many of these tasks demand models capable of processing long inputs, motivating LLMs such as Gemini (DeepMind, 2025) with input windows of up to 1M tokens. Among these domains, conversational systems present an intuitive and critical need for extended context, as users often engage in protracted, multi-session dialogues that require consistent memory across lengthy interactions (Zhong et al., 2024; Xu et al., 2022; Du et al., 2024; Tan et al., 2025). This highlights the importance of evaluating how well LLMs can reason over and utilize long conversational histories.

While there are many prior efforts on studying and evaluating long-term memory of LLMs (Kim et al., 2024a; Xu et al., 2021; Maharana et al., 2024; Zhong et al., 2024; Xu et al., 2022; Du et al., 2024; Tan et al., 2025), existing benchmarks have fundamental limitations. Most extend conversation length by artificially concatenating short sessions of different users, producing dialogues with abrupt topic shifts and weak narrative coherence. Such a construction artificially simplifies evaluation because distinct segments are easily separable, reducing the need for true long-range reasoning. Furthermore, these datasets typically target narrow domains—often limited to personal-life scenarios—leaving many real-world application areas underrepresented. Finally, they emphasize simple context recall, overlooking other critical memory abilities such as contradiction resolution, recognizing evolving information, and instruction following.

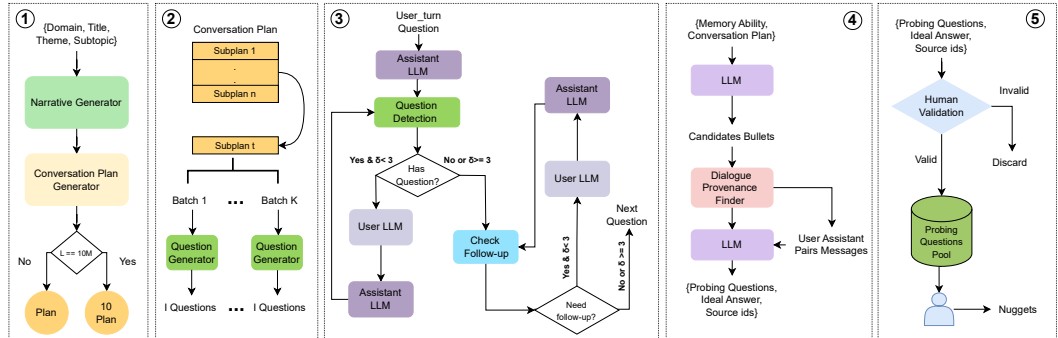

Figure 1: Overview of BEAM generation process. In the first stage, conversation plans are created for each chat seed. In the second stage, user utterances are generated from the conversation plans. In the third stage, assistant responses are produced. In the fourth stage, probing questions are generated based on the targeted memory abilities and corresponding conversation plans. In the final stage, invalid probing questions are filtered out, and nuggets are created for the validated set.

To address these limitations, this paper presents a framework for automatically generating long coherent conversations between a user and an AI assistant—scaling up to 10M tokens on diverse domains—with a set of probing questions designed to evaluate diverse memory abilities of any LLM on the generated dialogues. An overview of the data generation framework is shown in Figure 1. This framework begins by defining a high-level conversation plan—a narrative for a particular domain and a simulated user with generated attributes—that outlines the overall flow of the dialogue. This plan is recursively decomposed into finer sub-plans that specify the storyline and its progression. From these sub-plans we generate chronologically ordered user turns, which are then expanded with corresponding assistant responses. To increase realism, the system injects follow-up questions and clarifications from both sides. Finally, we automatically create a set of probing questions that target ten distinct memory dimensions, with a focus on complicated and multi-hop reasoning, which are then validated by human annotators to ensure high quality. Using this pipeline, we construct the BEAM dataset: 100 diverse conversations ranging from 100 K to 10 M tokens each, accompanied by 2000 probing questions to evaluate the memory capabilities of LLMs.

To improve LLM performance on probing questions, we introduce the LIGHT framework (Figure 2), which is applicable to both open-source and proprietary LLMs, inspired by research in human cognitive science and human's memorization and recall process (Sridhar et al., 2023; Binder & Desai, 2011). This framework integrates three complementary memories: (1) episodic memory, a long-term index of the full conversation used for retrieval; (2) working memory, capturing the most recent user–assistant turns; and (3) a scratchpad, where after each turn the model reasons over the dialogue and records salient facts for future use. At inference, the LLM draws jointly on retrieved episodic content, the working memory, and the accumulated scratchpad to generate accurate answers.

To evaluate LLM memory capabilities and the effectiveness of our method, we conduct experiments on the constructed dataset, BEAM, using both open-source and proprietary models. Results show that even LLMs with long context windows perform substantially worse as conversation length increases. Our method improves the LLM's performance in answering the probing questions by 3.5%–12.69% on average over the best-performing baseline, depending on the backbone model and conversation length. An ablation study further reveals the contribution of each LIGHT component on the performance. To support future work, we release all code, data, and evaluation scripts.[1]

## 2 BEAM: BENCHMARKING MEMORY CAPABILITIES OF LLMs

### 2.1 PROBLEM FORMULATION

Let $\mathcal{D} = \{T_i\}_{i=1}^{|\mathcal{D}|}$ denote a collection of $|\mathcal{D}|$ conversations between users and a conversational agent $\pi$. Each conversation is represented as $\mathcal{T} = \{t_i\}_{i=1}^{|\mathcal{T}|}$, where $t_i \in \mathcal{T}$ corresponds to the $i^{\text{th}}$ utterance

---

[1]Available at: https://github.com/mohammadtavakoli78/BEAM

(turn) in the dialogue. The objective of this work is to systematically evaluate a predefined set of memory abilities $\mathcal{M}$ exhibited by $\pi$ across conversations. For each memory ability $m \in \mathcal{M}$, we construct a probing dataset of size $N$, denoted as $\mathcal{Q}_m = \{(x_i, y_i)\}_{i=1}^N$, where $x_i$ is a probing question and $y_i$ is the corresponding ground-truth answer set. Each probing question $(x, y) \in \mathcal{Q}_m$ is appended as the $(|\mathcal{T}| + 1)^{\text{th}}$ turn in the dialogue, and the system generates a response $\hat{y} = \pi(x; \mathcal{T})$ based on the conversation. The generated response is then evaluated using an ability-specific scoring function $\mu_m$, producing a performance score $s = \mu_m(x, y, \hat{y})$. The goal of this work is to quantify the performance of conversational systems on each memory ability in $\mathcal{M}$.

## 2.2 BENCHMARK CREATION

Our goal is to evaluate how well LLMs can answer questions that depend on long-term conversational memory. We measure performance across ten complementary abilities, seven drawn from prior benchmarks and three newly introduced here—*Instruction Following*, *Event Ordering*, and *Contradiction Resolution* (see Table 2 in Appendix B.1). *Abstention* evaluates whether a model withholds answers when evidence is missing. *Contradiction Resolution* tests the capacity to detect and reconcile inconsistent statements across widely separated turns, maintaining global coherence. *Event Ordering* assesses whether a model can recognize and reconstruct the sequence of evolving information in the dialogue. *Information Extraction* measures recall of entities and factual details in long histories. *Instruction Following* examines sustained adherence to user-specified constraints over long contexts. *Information Update* evaluates revising stored facts as new ones appear. *Multi-hop Reasoning* probes inference that integrates evidence across multiple, non-adjacent dialogue segments. *Preference Following* captures personalized responses that adapt to evolving preferences. *Summarization* assesses the ability to abstract and compress dialogue content, while *Temporal Reasoning* tests reasoning about explicit and implicit time relations. Together, these abilities evaluate a system's capacity to maintain, update, and manipulate information throughout extended conversations (see Appendix B.6 for examples of each ability). Given these abilities and the formulation in Section 2.1, the benchmark requires three components: 1) a user–assistant conversation, 2) probing questions targeting key memory abilities, and 3) an evaluation methodology to assess the model's responses. The overall statistics of the constructed benchmark are summarized in Table 3 in Appendix B.1. The rest of this section details the process used to construct these components.

**Overview:** The overview of our framework for creating conversations, probing questions, and the evaluation strategy is illustrated in Figure 1. The process begins by generating a simulated conversation between a user and an assistant. Structured conversation plans are first produced to guide the flow of the synthetic interactions. Each plan specifies sufficient information to generate both user and assistant turns, ensuring a coherent and natural conversational trajectory. While a typical exchange consists of a user question followed by an assistant response, realistic dialogues often involve follow-ups for clarification, elaboration, or related subtopics. To capture this, we incorporate two interaction-control modules. The question-detection module identifies whether an assistant response includes a query that requires a user reply; if triggered, the system generates the corresponding user response. The follow-up detection module determines when the user would naturally pose a clarifying or elaborative question; if triggered, it produces an additional user query for the assistant. Together, these mechanisms produce conversations that exhibit interactive, bidirectional behavior beyond simple turn-taking. After the conversation is generated, an automated procedure constructs a candidate set of probing questions, each tailored to the specific memory abilities in the benchmark. These candidates are then reviewed by a human evaluator, who selects valid questions and formulates the associated evaluation rubrics used for subsequent benchmarking. A case study and an example of the different generated components of a conversation is provided in Appendix E.

### 2.2.1 CONVERSATION PLAN GENERATION

A *conversation plan* serves as the scaffold for each dialogue, providing a coherent storyline that unfolds chronologically. Each plan is generated using an LLM based on seed information, including: the conversation *domain*; a *title and theme*; *subtopics* outlining specific topics; a set of *narratives* defining evolving aspects (e.g., career progression, goals); a *user profile* with attributes such as name, age, gender, location, profession, and personality traits sampled from the Myers–Briggs Type Indicator (MBTI); a *relationship graph* linking the user to family, friends, and acquaintances, constrained for realism (e.g., age gaps); and an explicit *timeline* specifying the span of the conversation.

To generate candidate titles and themes, human annotators specify target domains, then GPT-4.1 (OpenAI, 2025a) generates candidate titles, themes, and subtopics using Listing 22. Human reviewers refine outputs for topical diversity. For each conversation, we generate 15-20 narratives using the open-source LLaMA-3.3 70B model (AI, 2024) with the prompt in Listing 23 (Appendix H). Given the conversation seed, this model produces narrative elements capturing the evolving storyline, forming the backbone of a coherent conversation.

Conversation plans consist of $N$ *sub-plans*, each representing a distinct stage in the conversation. Each sub-plan contains $M$ *bullet points*, defined by a *narrative*, a descriptive statement of its role in the storyline, and a *time anchor*. For conversations of 128K, 500K, and 1M tokens, a single plan is generated (line 4 in Algorithm 1, Appendix B.3.5) by conditioning the LLM on the conversation seed, profile, relationship graph, timeline, and specified counts of sub-plans, bullet points, and narratives (prompt in Listing 24, Appendix H). The number of sub-plans varies with domain and target length to meet the token requirement; e.g., coding domains generally require fewer turns than broader domains. For 10M-token conversations, one plan cannot capture the scope, so we create ten interlocking plans forming a coherent longer narrative. The process begins with a global seed defining the overall topic and theme, but a single seed is insufficient; instead, we derive ten distinct seeds—one per plan—so the narrative can evolve across stages. We propose two strategies:

- **Sequential Expansion:** The global seed defines the initial point in the conversation's chronology. Subsequent seeds represent successive events (e.g., a trip, job search, later milestones). Using the prompt in Listing 28 (Appendix H), each new seed is generated from the main seed, profile, and timeline. Plans are then produced sequentially (line 12 in Algorithm 1, Appendix B.3.5), with each plan conditioned on its predecessor to maintain continuity. Core relationships (e.g., parents) remain fixed, while new acquaintances are gradually introduced to reflect the evolving context.

- **Hierarchical Decomposition:** The main seed is decomposed into ten sub-seeds, each representing a distinct topical and temporal segment. Together, these sub-seeds span the full storyline (e.g., an international trip: first three for preparation, next five for trip events, final two for reflections). Similar to sequential expansion, the user's core relationships remain constant, while new acquaintances are introduced to reflect the evolving context. These ten sub-seeds are generated using the prompt in Listing 29 (Appendix H), conditioned on the main seed, profile, and timeline.

Each conversation plan is assigned explicit topical and temporal boundaries—encoded in the seed—to avoid redundancy and ensure sub-themes appear in the right narrative stage. For coherence, the LLM conditions on summaries of prior plans and future seeds when producing a new plan, allowing anticipation of upcoming events (e.g., reserving tickets for travel dates). This procedure is implemented in line 20 of Algorithm 1 (Appendix B.3.5). Plans are generated using the prompt in Listing 31 (Appendix H), conditioned on the main seed, current sub-seed, number of sub-plans, narrative set, user profile, core and new relationships, preceding and subsequent sub-seeds, previous plan, a summary of earlier plans, current sub-seed index, and a binary flag for the first plan (triggering user introduction). Since initial plans may not sufficiently test three key memory abilities—*contradiction resolution*, *information update*, and *instruction following*—we apply a two-stage augmentation: first generate the base plan, then use GPT-4.1 (Listing 27) to augment each sub-plan with three targeted bullet points. Performing augmentation separately improves coverage and fidelity. The refinement follows the prompt in Listing 27 (Appendix H), which takes plan as input and outputs the revised version. This stage corresponds to the first module in Figure 1, which forms the first step of the overall data-generation pipeline. The detailed process for plan generation is reported in Appendix B.3.2.

### 2.2.2 USER UTTERANCE GENERATION

Once conversation plans are constructed, user utterances are synthesized from the sub-plans. Each sub-plan contains $M$ bullet points, which are divided sequentially into $K$ contiguous batches of equal size. Batching narrows the LLM's focus, reducing repetition and low-quality outputs that can occur when conditioning on the entire sub-plan. For each batch, the LLM generates $I$ user questions (line 6 in Algorithm 2 in Appendix B.3.5) using the prompt in Listing 32 (Appendix H), conditioned on the conversation seed, the current batch, preceding batches, and context from earlier sub-plans. Each generated user question constitutes a user turn in the dialogue, ensuring coherence and continuity across extended conversations. Values of $K$ and $I$ are manually specified based on domain and target conversation length to meet the token budget, with configurations reported in

Table 6 (Appendix B). This provides fine-grained control over user interaction density, preventing under-generation or redundancy. To balance quality and cost, question generation uses the open-source LLaMA-3.3 70B model (AI, 2024), which produces high-quality outputs efficiently as the backbone LLM. This user-utterance construction aligns with the second stage in Figure 1. The details of this procedure for user utterance generation are provided in Appendix B.3.3.

### 2.2.3 ASSISTANT UTTERANCE GENERATION

Assistant-side responses are generated iteratively in a role-playing setup, where one LLM assumes the *assistant role* and another the *user role*. For each sub-plan, the assistant LLM is conditioned on the conversation seed (Section 2.2.1), prior sub-plans, a summary of the last $M$ turns, and a compressed summary of earlier ones (using the prompt in Listing 37 in Appendix B); for 10M-token conversations, additional summaries of prior plans are provided. The assistant first generates a response to the user's most recent question (line 9 in Algorithm 3 in Appendix B.3.5), which is analyzed by a *question-detection module* (line 11 in Algorithm 3 in Appendix B.3.5, using the prompt in Listing 35 Appendix B) to determine the presence of a counter-question. If detected, the response is passed to the user LLM, which generates a contextually consistent reply based on the current and prior sub-plans, relevant history, and conversation summaries (using the prompt in Listing 38 in Appendix B, line 14 in Algorithm 3 in Appendix B.3.5). This loop continues until no further assistant questions are detected or the threshold $\delta_1 = 2$ is reached, balancing realism and avoiding infinite cycles. In addition, a *follow-up detection module* (line 21 in Algorithm 3 in Appendix B.3.5, using the prompt in Listing 36 in Appendix B) evaluates whether a clarifying or elaborative user follow-up is warranted, based on factors such as subject complexity, ambiguity, or incomplete responses. When required, the module generates a follow-up query conditioned on the seed, current and prior sub-plans, the most recent $M$ turns, and earlier summaries (using the prompt in Listing 39 in Appendix B), which is then passed back to the assistant LLM. The number of follow-up exchanges is limited by a threshold $\delta_2 = 2$, analogous to $\delta_1$. Together, these modules yield dialogues with bidirectional dynamics, contextual referencing, and realistic clarifications, approximating human–AI interactions. This assistant-side generation maps to the third module in Figure 1. The details of this procedure are provided in Appendix B.3.4.

### 2.3 PROBING QUESTIONS GENERATION

After constructing conversations, we generate probing questions to evaluate memory abilities. The pipeline combines automated synthesis with human validation: an LLM first produces candidate probes, which annotators review to select valid ones. Probes are derived from both the conversation plan and chat to ensure each targets a specific ability, is grounded in dialogue turns, and includes explicit provenance. The process begins by passing the plan to GPT-4.1-mini (OpenAI, 2025b), which selects candidate bullet points conditioned on the ability under evaluation. For example, knowledge-update probes require bullet pairs encoding an initial fact and its later revision, while summarization and event-ordering probes span multiple bullets. Each bullet is linked to its corresponding user and assistant turns through indices introduced during user-assistant turn generation, enabling retrieval of the precise dialogue segments in which the content was created. Candidate bullet selection is performed using prompts 1–9, one per memory ability. For abstention, candidate selection is unnecessary; probes are created directly from the plan using the prompt shown in Listing 14 (Appendix H).

Given the selected bullet points and aligned dialogue snippets, GPT-4.1-mini generates the probing question, a candidate answer, and source identifiers citing the specific messages containing the answer. For 10M-token dialogues, candidate selection and synthesis are performed with a sliding window across the ten interlocking plans, processing a limited number at a time to preserve topical locality and scalability. Probe generation uses prompts 10–19 for each memory ability, mapping candidate bullet points and contexts into fully formed questions. Finally, a human evaluator reviews the generated candidates and selects those that are valid and consistent with the conversation. Samples of probing questions are provided in Appendix D, items 1–10.

### 2.4 EVALUATION

We evaluate LLMs on the probing questions using nugget evaluation, a common approach for long-form text assessment (Pradeep et al., 2024; 2025). Each probing question is manually validated:

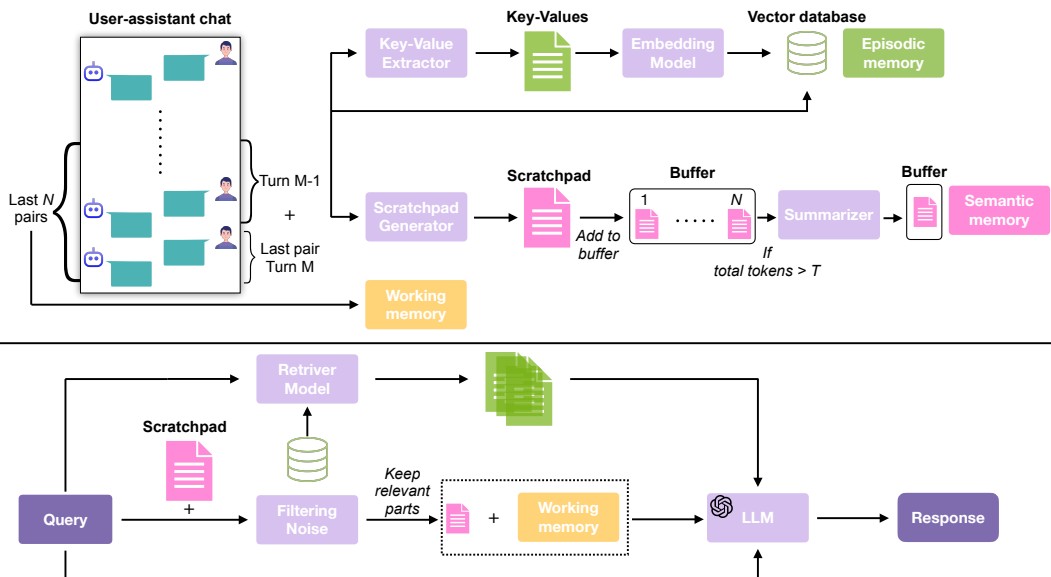

Figure 2: Overview of the LIGHT framework. The system combines (i) **episodic retrieval**, (ii) a **scratchpad** and (iii) a **working memory buffer**. At inference, relevant items from the index and scratchpad, together with the full working memory, are integrated to generate the final response.

invalid or unsupported questions are discarded, and minor inconsistencies are corrected. From the validated set, two questions per memory ability are chosen for each conversation, yielding 20 probing questions per conversation. Rubric nuggets are then derived for each question. A nugget is an atomic, self-contained criterion that a system response must satisfy. Annotators decompose the ideal reference answer into minimal semantic units, ensuring each nugget is both atomic and self-contained. System responses are scored against these nuggets by an LLM judge (Listing 20, Appendix H), which assigns 0 (unsatisfied), 0.5 (partially satisfied), or 1 (fully satisfied). Scores are averaged across nuggets to produce ability-level metrics. This nugget-based procedure applies to nine memory abilities; the exception is event ordering, where quality depends on both recall and correct sequence. We evaluate event ordering using the Kendall tau-b coefficient (Kendall, 1945), which considers both order and presence. To apply this metric, an LLM equivalence detector (using the prompt in Listing 21 in Appendix H) aligns events in system responses with nuggets, outputting `yes` if two snippets denote the same event/topic and `no` otherwise. Kendall tau-b is then computed over the aligned sequences, capturing both recall and ordering fidelity. Examples of nugget construction for each memory ability are provided in Appendix D.

## 3 LIGHT: IMPROVING MEMORY CAPABILITIES OF LLMS

Inspired by research in human cognitive science (Sridhar et al., 2023; Binder & Desai, 2011), humans employ two primary mechanisms for remembering and using knowledge: *episodic memory*, the ability to recall specific personal experiences along with their context, and *working memory*, the capacity to retain and manipulate information about recent events over short periods. In addition, maintaining notes on a *scratchpad* provides an external record that supports long-term recall and later retrieval. Since answering questions in long-context conversations similarly requires integrating past experiences and accumulated knowledge, we introduce a method that emulates these strategies by combining episodic recall, short-term working memory, and an external scratch-pad mechanism.

**Overview:** An overview of our method is shown in Figure 2. Given a question $x$ about a conversation $\mathcal{T} = \{t_i\}_{i=1}^{|\mathcal{T}|}$, where $|\mathcal{T}|$ is the total number of turns, the framework first queries a retrieval model $R$ to obtain $k$ relevant segments from $\mathcal{T}$, simulating recall from episodic memory: $E = R(x, k, \mathcal{T})$. Next, the most recent $z$ dialogue pairs of the conversation are selected to form the working memory, $W = \{t_{|\mathcal{T}|-i}\}_{i=0}^{z}$. In parallel, a pre-constructed scratchpad $S_{|\mathcal{T}|}$ contains up to $m$

salient notes. A filtering function $f$ retains only the items pertinent to $x$, yielding $S_x = f(S_{|\mathcal{T}|}, x)$. Finally, the LLM $\pi$ generates the answer by conditioning on the question and these three memory components, $y = \pi(x, E, W, S_x)$ using the prompt shown in Listing 44 in Appendix H. The remainder of this section details the construction and logic of each component in this pipeline.

### 3.1 RETRIEVAL FROM THE CONVERSATION

**Indexing the Conversation:** After each user–assistant turn (Figure 2, top), we apply Qwen2.5-32B-AWQ (Team, 2024) with the prompt in Listing 40 (Appendix H) to extract key–value pairs and a summary of the interaction. Keys represent entities and values capture attributes or descriptive details, providing fine-grained, event-level indices analogous to hippocampal memory traces (Teyler & DiScenna, 1986). These key–value pairs and summaries are embedded using the BAAI/bge-small-en-v1.5 embedding model (of Artificial Intelligence, 2023) and stored in a vector database as keys, while the original dialogue segments are kept as values to ensure faithful grounding.

**Retrieval from the Index:** To retrieve information from the conversation as episodic memory, we embed the question $x$ using the same embedding model and compare it against the stored keys in the index, and the original dialogue segments corresponding to the top $k$ nearest neighbors are returned.

### 3.2 SCRATCHPAD FORMATION AND UTILIZATION

**Construction:** In addition to episodic memory (Figure 2, middle pathway), we build a higher-level representation that preserves information beyond individual dialogue events. It integrates semantic knowledge (facts and concepts), autobiographical details (life events), prospective memory (future intentions), and contextual metadata (time, place, acquisition context) (Binder & Desai, 2011). For each dialogue pair, we use Qwen2.5-32B-AWQ with the prompt in Listing 41 (Appendix H) to reason over the current and preceding turn and extract salient content. The resulting "scratchpad" is iteratively merged with earlier versions; once content exceeds a 30K-token threshold—substantially shorter than the raw conversation—it is compressed into a 15K-token summary by GPT-4.1-nano using the prompt in Listing 42. This process maintains efficiency and long-term coherence, analogous to the gradual abstraction of semantic memory in humans. Unlike the episodic index, the scratchpad is not stored in a retrieval database but is provided directly as contextual input during inference.

**Filtering Scratchpad (function $f$):** During inference, the scratchpad is selectively filtered with respect to the question. It is first divided into semantically coherent chunks using *semantic chunking*.[2] Each chunk is evaluated by Qwen2.5-32B-AWQ with the prompt in Listing 43 (Appendix H), which assigns a binary relevance label (`yes`/`no`). Only the chunks judged relevant are retained, producing a condensed representation of scratchpad that is passed to the response generator.

## 4 EXPERIMENTS

### 4.1 EXPERIMENTAL SETUP

**Baselines:** We evaluate our approach against two types of baselines: long-context LLMs and a RAG method. For long-context LLMs, the entire conversation history is provided, followed by the probing question. We include two proprietary LLMs (*GPT-4.1-nano*, *Gemini-2.0-flash*, both 1M context). and two open-source models (*Qwen2.5-32B-AWQ*, *Llama-4-Maverick-fp8*). For long-context experiments, *Qwen2.5-32B-AWQ* is evaluated with a 128K context length, while for the RAG baseline and our proposed method a 32K context length is used. At the 10M-token, since none of the four models support this length, they are evaluated on the largest recent dialogue segment fitting their window.[3] For RAG baselines, each user–assistant turn pair is treated as a document, embedded and stored in a vector database. At inference, the top five most similar documents are retrieved and passed to the LLM using the prompt in Listing 44 (Appendix H).

---

[2]`SemanticChunker` in LangChain is used, which segments text into variable-length passages based on semantic rather than fixed token windows.

[3]Among available models, only *Llama-4-Scout* supports 10M-token context windows; however, due to its extreme computational requirements, we were unable to include it in our experiments.

Table 1: Comparison of different LLMs and methods across conversation lengths and memory abilities using the created benchmark. Methods with the best performance per evaluation are bolded.

| Length | Memory Ability | Qwen 2.5 | | | Llama Maverick | | | Gemini 2 Flash | | | GPT-4.1-nano | | |
|---|---|---|---|---|---|---|---|---|---|---|---|---|---|
| | | Vanilla | RAG | Ours | Vanilla | RAG | Ours | Vanilla | RAG | Ours | Vanilla | RAG | Ours |
| 100K | Abstention | 0.300 | **0.650** | 0.475 | 0.200 | **0.800** | 0.600 | **0.800** | 0.800 | 0.675 | 0.475 | **0.800** | 0.575 |
| | Contradiction Resolution | 0.031 | 0.025 | **0.037** | 0.025 | 0.031 | **0.031** | 0.006 | **0.050** | 0.018 | 0.012 | 0.018 | **0.031** |
| | Event Ordering | 0.192 | 0.201 | **0.205** | **0.190** | 0.162 | 0.166 | 0.181 | **0.191** | 0.166 | **0.181** | 0.169 | 0.177 |
| | Information Extraction | 0.425 | 0.338 | **0.479** | 0.510 | 0.392 | **0.518** | 0.333 | 0.341 | **0.464** | 0.273 | 0.362 | **0.538** |
| | Instruction Following | **0.400** | 0.375 | 0.362 | 0.412 | 0.375 | **0.412** | 0.275 | 0.287 | **0.362** | **0.425** | 0.350 | 0.400 |
| | Knowledge Update | **0.437** | 0.275 | 0.362 | 0.300 | 0.350 | **0.450** | 0.125 | **0.325** | 0.300 | 0.275 | 0.375 | **0.375** |
| | Multi-Hop Reasoning | 0.222 | 0.203 | **0.281** | 0.152 | 0.225 | **0.353** | 0.200 | 0.148 | **0.225** | 0.178 | 0.263 | **0.365** |
| | Preference Following | 0.554 | 0.379 | **0.566** | 0.450 | 0.512 | **0.625** | 0.300 | 0.416 | **0.462** | 0.437 | 0.550 | **0.625** |
| | Summarization | 0.128 | 0.074 | **0.232** | 0.065 | 0.111 | **0.238** | 0.018 | 0.093 | **0.139** | 0.028 | 0.083 | **0.202** |
| | Temporal Reasoning | 0.112 | **0.162** | 0.112 | 0.100 | **0.275** | 0.187 | **0.187** | 0.150 | 0.125 | 0.112 | 0.125 | **0.162** |
| | Average | 0.280 | 0.269 | **0.311** | 0.240 | 0.323 | **0.358** | 0.242 | 0.280 | **0.294** | 0.239 | 0.309 | **0.345** |
| 500K | Abstention | 0.314 | **0.728** | 0.571 | 0.185 | **0.785** | 0.628 | 0.714 | **0.800** | 0.685 | 0.557 | **0.828** | 0.600 |
| | Contradiction Resolution | **0.053** | 0.017 | 0.017 | 0.035 | 0.028 | **0.042** | 0.010 | 0.021 | **0.021** | 0.017 | 0.025 | **0.035** |
| | Event Ordering | 0.185 | 0.221 | **0.244** | **0.209** | 0.186 | 0.197 | **0.215** | 0.189 | 0.200 | 0.188 | 0.180 | **0.204** |
| | Information Extraction | 0.166 | 0.400 | **0.506** | **0.608** | 0.402 | 0.535 | 0.469 | 0.343 | **0.478** | 0.142 | 0.382 | **0.491** |
| | Instruction Following | 0.304 | **0.350** | 0.295 | 0.403 | **0.447** | 0.390 | 0.133 | **0.334** | 0.280 | 0.244 | 0.286 | **0.342** |
| | Knowledge Update | 0.111 | 0.226 | **0.278** | 0.276 | **0.338** | 0.264 | 0.171 | 0.180 | **0.223** | 0.107 | **0.288** | 0.240 |
| | Multi-Hop Reasoning | 0.125 | 0.187 | **0.214** | 0.219 | 0.313 | **0.350** | **0.198** | 0.135 | 0.157 | 0.070 | 0.233 | **0.266** |
| | Preference Following | 0.567 | 0.477 | **0.571** | 0.560 | 0.525 | **0.623** | 0.379 | 0.427 | **0.532** | 0.450 | 0.577 | **0.684** |
| | Summarization | 0.137 | 0.187 | **0.344** | 0.266 | 0.197 | **0.373** | 0.136 | 0.165 | **0.250** | 0.109 | 0.184 | **0.334** |
| | Temporal Reasoning | 0.035 | 0.114 | **0.121** | 0.064 | 0.078 | **0.190** | **0.150** | 0.078 | 0.092 | 0.057 | **0.161** | 0.154 |
| | Average | 0.200 | 0.291 | **0.316** | 0.283 | 0.330 | **0.359** | 0.257 | 0.267 | **0.292** | 0.194 | 0.314 | **0.335** |
| 1M | Abstention | 0.342 | **0.650** | 0.500 | 0.221 | **0.742** | 0.435 | 0.642 | **0.750** | 0.735 | 0.492 | **0.778** | 0.678 |
| | Contradiction Resolution | **0.035** | 0.035 | 0.021 | **0.046** | 0.028 | 0.042 | 0.010 | **0.028** | 0.007 | **0.050** | 0.028 | 0.021 |
| | Event Ordering | 0.183 | 0.195 | **0.200** | 0.214 | 0.179 | 0.193 | 0.190 | **0.198** | 0.185 | 0.191 | 0.179 | **0.211** |
| | Information Extraction | 0.138 | **0.407** | 0.366 | **0.489** | 0.431 | 0.474 | 0.374 | **0.380** | 0.341 | 0.153 | 0.399 | **0.410** |
| | Instruction Following | 0.383 | 0.300 | **0.419** | **0.440** | 0.338 | 0.433 | 0.120 | 0.290 | **0.380** | 0.226 | 0.271 | **0.394** |
| | Knowledge Update | 0.064 | **0.378** | 0.357 | 0.164 | 0.342 | **0.414** | 0.107 | **0.278** | 0.264 | 0.150 | 0.342 | **0.392** |
| | Multi-Hop Reasoning | 0.102 | 0.163 | **0.209** | 0.174 | 0.245 | **0.270** | 0.083 | 0.134 | **0.147** | 0.091 | **0.293** | 0.278 |
| | Preference Following | 0.486 | 0.491 | **0.551** | 0.535 | 0.514 | **0.610** | 0.273 | 0.470 | **0.472** | 0.435 | 0.513 | **0.576** |
| | Summarization | 0.122 | 0.157 | **0.316** | 0.207 | 0.145 | **0.315** | 0.091 | 0.125 | **0.224** | 0.060 | 0.152 | **0.290** |
| | Temporal Reasoning | 0.073 | 0.078 | **0.154** | 0.097 | 0.107 | **0.176** | **0.104** | 0.057 | 0.085 | 0.061 | 0.064 | **0.107** |
| | Average | 0.193 | 0.285 | **0.309** | 0.259 | 0.307 | **0.336** | 0.199 | 0.271 | **0.284** | 0.191 | 0.302 | **0.336** |
| 10M | Abstention | 0.250 | **0.600** | 0.550 | 0.050 | **0.700** | 0.450 | **0.750** | 0.650 | 0.650 | 0.450 | **0.650** | 0.400 |
| | Contradiction Resolution | **0.050** | 0.000 | 0.012 | **0.025** | 0.000 | 0.000 | 0.000 | **0.025** | 0.000 | 0.000 | 0.012 | **0.025** |
| | Event Ordering | 0.180 | **0.221** | 0.197 | 0.190 | **0.220** | 0.176 | 0.220 | **0.266** | 0.193 | **0.215** | 0.201 | 0.173 |
| | Information Extraction | 0.100 | 0.350 | **0.350** | 0.075 | **0.375** | 0.300 | 0.075 | **0.275** | 0.150 | 0.050 | 0.300 | **0.350** |
| | Instruction Following | 0.175 | 0.200 | **0.350** | 0.250 | 0.350 | **0.500** | 0.025 | 0.125 | **0.250** | 0.075 | 0.175 | **0.250** |
| | Knowledge Update | 0.100 | **0.300** | 0.275 | 0.100 | **0.375** | 0.325 | 0.050 | **0.325** | 0.200 | 0.050 | **0.325** | 0.300 |
| | Multi-Hop Reasoning | 0.125 | 0.050 | **0.125** | 0.000 | 0.075 | **0.125** | 0.000 | 0.125 | **0.125** | 0.012 | 0.091 | **0.135** |
| | Preference Following | 0.241 | 0.291 | **0.308** | 0.291 | 0.316 | **0.483** | 0.075 | **0.300** | 0.150 | 0.175 | 0.366 | **0.425** |
| | Summarization | 0.114 | 0.106 | **0.220** | 0.065 | 0.053 | **0.277** | 0.000 | 0.045 | **0.136** | 0.020 | 0.063 | **0.179** |
| | Temporal Reasoning | 0.000 | 0.000 | 0.000 | 0.000 | 0.025 | **0.025** | 0.025 | 0.025 | **0.075** | **0.050** | 0.000 | 0.025 |
| | Average | 0.133 | 0.211 | **0.238** | 0.104 | 0.249 | **0.266** | 0.122 | **0.216** | 0.192 | 0.109 | 0.218 | **0.226** |

**Inference Setup:** For inference, we use Nucleus (Holtzman et al., 2020) with temperature 0, except for conversation plan, user-turn, and assistant-turn generation, where temperature is 0.1 to encourage diversity. All open-source LLMs are served via VLLM for efficient inference. For Llama3.3-70B, we set the maximum output length to 6K tokens during user-turn generation, while for other LLMs we adopt their default maximum output length. For experiments involving both the RAG baseline and our proposed method, we employ FAISS as the vector database (Douze et al., 2024). For dense retrieval, we use the embedding model *BAAI/bge-small-en-v1.5* (Xiao et al., 2023).

## 4.2 EMPIRICAL FINDINGS

**Main Results:** Across all four conversation lengths (100K–10M tokens), our method consistently outperforms both long-context LLMs and RAG baselines (Table 1). At shorter contexts (100K), we observe strong gains, such as +49.1% for Llama-4-Maverick and +44.3% for GPT-4.1-nano over long-context baselines, showing that structured memory helps even when full history can be processed. The benefits grow with context length: at 1M tokens, improvements reach +75.9% for GPT-4.1-nano and +60.1% for Qwen2.5-32B. At 10M tokens—where no baseline natively supports the full context—our method achieves dramatic improvements, including +155.7% for Llama-4-Maverick and +107.3% for GPT-4.1-nano. The only exception is Gemini-2.0-flash at 10M, where our method surpasses the long-context baseline (+57.3%) but slightly trails RAG, likely due to model-specific retrieval behavior. Overall, these findings underscore the scalability and robustness of our framework across diverse architectures and extreme context lengths.

When evaluated across the ten memory abilities, our method shows the largest relative gains in summarization (+160.6%), multi-hop reasoning (+27.2%), and preference following (+76.5%). Strong improvements are also observed in information extraction (+56.7%), instruction following (+39.5%), and temporal reasoning (+56.3%). These results highlight that our method is particularly

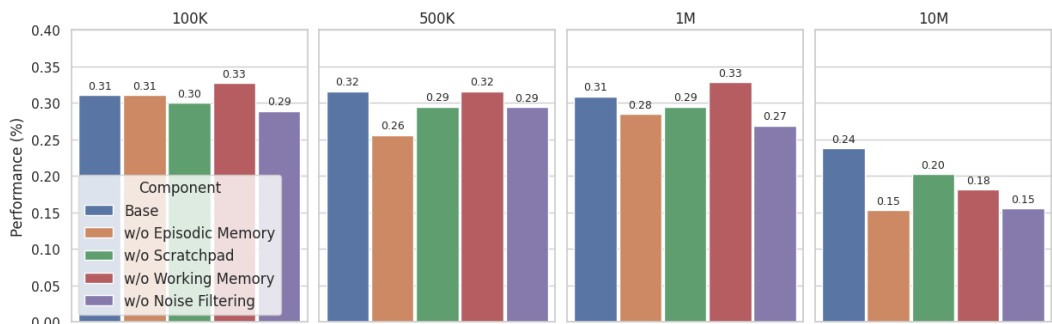

Figure 3: Ablation study illustrating the contribution of each component in LIGHT (retrieval, scratchpad, working memory, and noise filtering) across different conversation lengths.

effective for tasks requiring long-range recall and integration of dispersed information. In contrast, all methods—including ours—perform strongest in abstention and weakest in contradiction resolution, indicating that contradiction detection remains a challenging open problem.

**Ablation:** We conduct an ablation to assess the role of each component—*episodic memory*, *scratchpad*, *working memory*, and *noise filtering*—across conversation lengths (Figure 3). At 100K, removing retrieval does not change performance and it remains steady, since the scratchpad alone suffices, while removing scratchpad or noise filtering reduces performance (–1.1%, –2.2%). Working memory also degrades results here (–1.6%). At 500K, removing any component reduces performance except working memory, where removal enhances performance very slightly. At 1M, retrieval, scratchpad, and noise filtering remain beneficial, but removing working memory slightly improves performance, again reflecting its limited usefulness when few questions depend on the most recent turns. By 10M, all components are essential, with removals leading to large drops (–8.5% for retrieval, –3.7% for scratchpad, –5.7% for working memory, –8.3% for noise filtering). Overall, the ablations show that each module contributes increasingly as context length grows, and the full architecture consistently achieves the best performance. Detailed results across all memory abilities are provided in Table 8.

**Effect of Retrieval Budget:** We examine the effect of retrieval budget ($K$), testing 5, 10, 15, and 20 documents (Figure 4). Performance consistently improves when increasing $K$ from 5 to 15, with the best results at $K$=15 (+7.39%, +10.75%, +6.79%, and +6.3% at 100K, 500K, 1M, and 10M). Increasing further to $K$=20 slightly degrades performance, likely due to noisy context. Results at $K$=10 are mixed—helpful at 100K, 500K and 1M but harmful at 10M—indicating additional documents sometimes add noisy information. Full results across memory abilities are shown in Table 9. We also conducted complementary experiments analyzing the effect of retriever choice, where we observed that at 100K, 500K, and 1M token

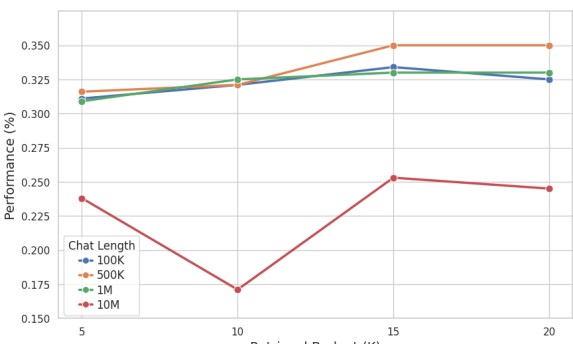

Figure 4: Effect of varying retrieval budget ($K$) on performance. The plot shows how the number of retrieved documents shapes the balance between recall and noise, highlighting different behaviors at short and long conversation lengths.

lengths, using a sparse retriever improves performance, whereas at 10M tokens, the dense retriever achieves better results. The full results and discussion are provided in Appendix C.2.

**Case Study** A case study demonstrating the usefulness of the scratchpad is provided in Appendix F.

Additional analyses on embedding choice, indexing setup, and a supplementary baseline are provided in Appendix C.3–C.5.

**Human Evaluation:** We conducted a human evaluation to assess the quality of the generated conversations. Three dimensions were considered: *Coherence and Flow*, *Realism*, and *Complexity and Depth*, each rated on a 5-point Likert scale (1 = lowest, 5 = highest). The average scores across all conversations were 4.53, 4.57, and 4.64, respectively, indicating consistently high quality. Details of the annotation protocol and inter-annotator agreement are provided in Appendix B.2. We further present a qualitative error analysis characterizing LIGHT 's failure modes across memory abilities in Appendix G.

## 5 RELATED WORK

The detailed related work is provided in Appendix A; here we present a concise summary.

Context windows of LLMs have expanded dramatically, from early limits of 512–2K tokens (GPT-2/3; (Radford et al., 2019; Brown et al., 2020)) to 128K–1M (Claude-3, GPT-4-Turbo, Gemini 2.0; (DeepMind, 2025; Anthropic, 2025; OpenAI, 2025a)) and even 10M (Llama 4; (Meta-AI, 2025)). This growth is driven by advances in efficient attention (sparse, linear, memory-optimized kernels; (Beltagy et al., 2020; Wang et al., 2020; Dao et al., 2022)), improved positional encodings (relative, rotary with scaling, ALiBi; (Dai et al., 2019; Peng et al., 2023b)), long-context training strategies (continued-training, curriculum learning; (Xiong et al., 2023; Ding et al., 2024)), and inference optimizations such as paged attention, KV-cache compression, and distributed attention (Kwon et al., 2023; Zhang et al., 2023; Li et al., 2024; Liu et al., 2023). Such capabilities are especially valuable for applications involving conversational histories, the main focus of our work.

Beyond expanding context windows, models incorporate additional mechanisms for persistent memory. These include recurrence and compression (Transformer-XL, Compressive Transformer; (Dai et al., 2019; Rae et al., 2019)), state-space architectures (RWKV, Mamba, Hyena; (Peng et al., 2023a; Gu & Dao, 2023; Poli et al., 2023)), external memory modules (Memformer, RETRO, RMT; (Wu et al., 2020; Borgeaud et al., 2022; Fan et al., 2024)), context summarization (AutoCompressor; (Chevalier et al., 2023)), and retrieval-augmented generation (REALM, RAG, HippoRAG; (Guu et al., 2020; Lewis et al., 2020; Jimenez Gutierrez et al., 2024)). These approaches complement larger windows by enabling scalable and persistent long-term reasoning.

Existing benchmarks such as DialSim, MSC, LoCoMo, MemoryBank, DuLeMon, PerLTQA, Long-MemEval, and MemBench (Kim et al., 2024a; Xu et al., 2021; Maharana et al., 2024; Zhong et al., 2024; Xu et al., 2022; Du et al., 2024; Tan et al., 2025) evaluate recall, temporal reasoning, and multi-session reasoning, but typically span narrow domains, exhibit shallow dependencies, and concatenate separate user sessions to simulate long context, reducing realism. Recent work such as MemoryCode (Rakotonirina et al., 2025) generates multi-session dialogues from template-driven instruction seeds to assess long-context reasoning, but focuses on a single domain. Our benchmark instead scales to 10M tokens across diverse topics and introduces new tasks such as contradiction resolution, event ordering, and instruction following, generating coherent, single-user conversations that preserve narrative continuity for a more faithful assessment of long-term conversational memory.

## 6 CONCLUSION

This paper addresses the shortcomings of existing benchmarks for evaluating long-term memory in conversational systems. We introduce a scalable framework to generate BEAM, a new benchmark with long, coherent dialogues (up to 10M tokens) and diverse memory probes. To improve LLMs performance, we develop LIGHT, a cognitive-inspired framework combining episodic, working, and scratchpad memories. Our experiments show that while standard LLMs' performance degrades over long contexts, LIGHT provides substantial improvements, boosting memory performance by an average of 3.5%-12.69%. By offering a more robust evaluation and an effective memory enhancement technique, this work helps the development of more reliable long-context conversational systems.

ACKNOWLEDGMENTS

Ross Mitchell is the Alberta Health Services Chair in Artificial Intelligence in Health and is supported by CIFAR, the University Hospital Foundation, the Alberta Machine Intelligence Institute (Amii), and the Canada Foundation for Innovation. Mohamed Abdalla is supported by a CIFAR AI Chair. This research is supported by the Canadian Institutes of Health Research (FRF 196047). Carrie Ye is supported by the CRAF (CIORA)–Arthritis Society Canada Clinician Investigator Award (CI-24-0013). This research is supported in part by the Center for Intelligent Information Retrieval, in part by NSF grant #2143434, in part by the Office of Naval Research contract #N000142412612, and with support from Google.org. Any opinions, findings, and conclusions or recommendations expressed in this material are those of the authors and do not necessarily reflect those of the sponsors.

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

# A  DETAILED RELATED WORK

**Long-Context Large Language Models.** The context window of LLMs has expanded from 512–2,048 tokens in early models (GPT-1/2/3, BERT, T5; (Radford et al., 2018; 2019; Brown et al., 2020; Devlin et al., 2019; Raffel et al., 2020)) to 128K–1M tokens in recent systems (Claude-3, GPT-4-Turbo, Gemini 1.5 Pro, Gemini 2.0 Flash, Claude-4, GPT-4.1; (Anthropic, 2024; Achiam et al., 2023; Team et al., 2024; DeepMind, 2025; Anthropic, 2025; OpenAI, 2025a)), with some reaching 10M tokens (Llama 4 Scout; (Meta-AI, 2025)). This growth has been enabled by innovations that address the quadratic cost of self-attention, including sparse mechanisms (Longformer, BigBird; (Beltagy et al., 2020; Zaheer et al., 2020)), linear approximations (Linformer, Performer; (Wang et al., 2020; Choromanski et al., 2020)) and memory-efficient kernels (FlashAttention; (Dao et al., 2022)). Advances in positional encoding, such as relative encodings (Transformer-XL; (Dai et al., 2019)), rotary embeddings (RoPE; (Su et al., 2024)) with scaling methods (YaRN, NTK; (Peng et al., 2023b)), and linear biases (ALiBi; (Press et al., 2021)), have extended usable context lengths. Training strategies like continued pre-training and curriculum learning (e.g., LLaMA-2-Long (Xiong et al., 2023), LongRoPE (Ding et al., 2024)) further expand capabilities, while inference optimizations such as PagedAttention (Kwon et al., 2023), KV-cache compression (H2O, SnapKV; (Zhang et al., 2023; Li et al., 2024)) and distributed approaches (Ring Attention; (Liu et al., 2023)) enable practical deployment at scale.

**Long-Term Memory Methods.** Researchers have developed approaches to enhance long-term memory beyond simply extending context windows. Architectural modifications include Transformer-XL (Dai et al., 2019), which introduced segment-level recurrence, and Compressive Transformer (Rae et al., 2019), which stored both recent states and compressed older information. State-space models such as RWKV (Peng et al., 2023a), Mamba (Gu & Dao, 2023), and Hyena (Poli et al., 2023) replace attention with recurrent dynamics, allowing linear scaling and theoretically unbounded memory. Memory-augmented transformers such as Memformer (Wu et al., 2020), RETRO (Borgeaud et al., 2022) and RMT (Fan et al., 2024) add external memory slots for explicit storage and recall. Context compression offers an orthogonal strategy by summarizing past information rather than storing it verbatim, as in AutoCompressor (Chevalier et al., 2023), which learns compact, information-preserving representations to reduce token usage. Retrieval-augmented generation (RAG) scales further by maintaining external knowledge stores: REALM (Guu et al., 2020) and RAG (Lewis et al., 2020) pioneered dense retrieval, RETRO (Borgeaud et al., 2022) integrated retrieval into transformers, and HippoRAG (Jimenez Gutierrez et al., 2024) incorporated structured knowledge graphs.

Building on these foundations, we propose a novel retrieval-augmented method that shows substantial improvements over baselines in long-memory evaluation.

**Long-Term Memory Benchmarks.** Several benchmarks have emerged to evaluate long-term memory capabilities in LLMs. DialSim (Kim et al., 2024a) derives evaluation data from multiparty television scripts, producing dialogues extending to 350K tokens with naturalistic patterns but limited topical diversity. MSC (Xu et al., 2021) introduces multisession human-assistant conversations testing memory across session boundaries, though with brief sessions and shallow dependencies. LoCoMo (Maharana et al., 2024) presents 50 conversations averaging 9K tokens in 35 sessions, while MemoryBank (Zhong et al., 2024) provides 300 sessions with 194 probing questions evaluating recall and temporal reasoning. DuLeMon (Xu et al., 2022) focuses on dialogue-level memory and forgetting curves, PerLTQA (Du et al., 2024) targets memory classification and retrieval, and LongMemEval (Wu et al., 2024) constructs multisession evaluations with 500 questions testing information extraction and temporal reasoning. More recently, MemBench (Tan et al., 2025) evaluates the memory of LLM-based agents by assessing their performance on information extraction, multihop reasoning, knowledge updating, preference following, and temporal reasoning. Recent work such as MemoryCode (Rakotonirina et al., 2025) generates multi-session dialogues from template-driven instruction seeds to assess long-context reasoning, but focuses on a single domain.

As summarized in Table 2, the existing benchmarks are largely based on concatenated short sessions with limited coherence, narrow personal and casual domains, and few memory abilities. They also lack realistic bidirectional interactivity. In contrast, our benchmark spans diverse domains, scales up to 10M tokens, and introduces three additional dimensions—contradiction resolution, event order-

Table 2: Comparison of our benchmark with existing long-term memory benchmarks. Memory abilities: IE = Information Extraction, MR = Multi-hop Reasoning, KU = Knowledge Update, TR = Temporal Reasoning, ABS = Abstention, CR = Contradiction Resolution, EO = Event Ordering, IF = Instruction Following, PF = Preference Following, SUM = Summarization.

| Benchmark | Domain | Chat Length | Memory Abilities | | | | | | | | | |
|---|---|---|---|---|---|---|---|---|---|---|---|---|
| | | | IE | MR | KU | TR | ABS | CR | EO | IF | PF | SUM |
| MSC (Xu et al., 2021) | Casual | ∼1K | ✗ | ✗ | ✗ | ✗ | ✗ | ✗ | ✗ | ✗ | ✗ | ✗ |
| DuLeMon (Xu et al., 2022) | Casual | ∼1K | ✗ | ✗ | ✗ | ✗ | ✗ | ✗ | ✗ | ✗ | ✗ | ✗ |
| MemoryBank (Zhong et al., 2024) | Personal life | ∼5K | ✓ | ✗ | ✗ | ✓ | ✗ | ✗ | ✗ | ✗ | ✗ | ✗ |
| PerLTQA (Du et al., 2024) | Personal life | N/A | ✓ | ✗ | ✗ | ✗ | ✓ | ✗ | ✗ | ✗ | ✗ | ✗ |
| LoCoMo (Maharana et al., 2024) | Personal life | ∼10K | ✓ | ✓ | ✗ | ✓ | ✓ | ✗ | ✗ | ✗ | ✗ | ✓ |
| DialSim (Kim et al., 2024a) | TV/Film scripts | ∼350K | ✓ | ✓ | ✗ | ✓ | ✓ | ✗ | ✗ | ✗ | ✗ | ✗ |
| LongMemEval (Wu et al., 2024) | Personal life | 115K, 1M | ✓ | ✓ | ✓ | ✓ | ✓ | ✗ | ✗ | ✗ | ✓ | ✗ |
| MemBench (Tan et al., 2025) | Personal life | ∼100K | ✓ | ✓ | ✓ | ✓ | ✗ | ✗ | ✗ | ✗ | ✓ | ✗ |
| **BEAM (This work)** | **Multi-domain:** Coding, Math, Health, Finance, Personal life, ... | **128K, 500K, 1M, 10M** | ✓ | ✓ | ✓ | ✓ | ✓ | ✓ | ✓ | ✓ | ✓ | ✓ |

ing, and instruction following—yielding a more comprehensive framework for evaluating long-term memory in conversational systems.

# B  BENCHMARK DESIGN

## B.1  DATASET STATISTICS

Table 3 summarizes the statistics of the generated dataset, including averages of user messages, assistant messages, assistant and user follow-up questions, and dialogue turns across different chat sizes.

Table 3: Statistics of the dataset. Reported values are averages per chat in each chat size. # User Messages and # Assistant Messages denote the average number of utterances from the user and assistant, respectively. # Answer Assistant Questions is the number of times the assistant posed a question that the user answered. # Followup Questions is the number of follow-up questions asked by the user. # Turns refers to the total number of dialogue turns.

| Chat Size | # User Messages | # Assistant Messages | # Answer Assistant Questions | # Followup Questions | # turns |
|---|---|---|---|---|---|
| 128K | 144 | 144 | 27 | 216 | 107 |
| 500K | 544 | 544 | 79 | 51 | 416 |
| 1M | 1067 | 1067 | 105 | 120 | 842 |
| 10M | 10435 | 10435 | 1151 | 1528 | 7757 |

## B.2  BENCHMARK QUALITY EVALUATION

To evaluate the quality of the generated conversations, we conducted a human assessment across all conversations. Two annotators rated each conversation on three dimensions using a 5-point Likert scale (1 = lowest, 5 = highest): *Coherence and Flow*, *Dialogue Realism*, and *Complexity and Depth*.

**Annotation Setup and Procedure.** We used an internal system for the annotators. The annotations were completed by two annotators with a computer science background. Human involvement are as follows:

- **Probing Question Validation.** The annotators were first provided with guidelines and instructions on how to perform the task, and a training session was held. They went through

a calibration set together, completed multiple examples, and discussed their opinions on each example.

- **Rubric Design.** The annotators were given guidelines on how to design rubrics along with examples and templates to make the task easier and more unified. A training and calibration session was then held, during which annotators reviewed many samples together, asked questions, and received clarification on ambiguous and edge cases.
- **Conversation Quality Evaluation.** For this task, the annotators were again provided with guidelines on how to evaluate the generated conversations and assign scores. Because the conversations are very long and it is impractical to read them in full, annotators were instructed to read the initial 25 dialogue turns to understand what the conversation is about and to become familiar with the theme and flow of the chat. They were instructed to check the follow-up questions proposed by both the user and the assistant, the adherence of the conversation to the conversation plan and user utterances, and whether any hallucinations or unexpected behaviors occur. After that, they were asked to read several random turns between the initial 25 turns and the middle of the conversation to assess whether the conversation is progressing according to the conversation plan. They then had to read 25 dialogue turns in the middle of the conversation, followed by additional random turns between the middle and the end, and finally 25 dialogue turns near the end of the conversation. In this way, annotators read different parts of each conversation and ensured good coverage.

For every conversation, annotators rated the three quality dimensions described below using a 5-point Likert scale.

- **Coherence and Flow**: Conversation continuity (each turn follows naturally from the previous one), smooth transitions across topics and responses, and thread consistency without abrupt or jarring shifts.
- **Dialogue Realism**: Naturalness of user queries (messages sound authentic), realistic progression of topics over time, human-like interactions (appropriate clarifications, follow-ups, etc.), and believability of scenarios.
- **Complexity and Depth**: Handling of multi-layered, interconnected topics, progressive increase in difficulty, and demonstration of domain expertise when required.

**Inter-Annotator Agreement.** Before performing the full evaluation, annotators independently assessed a set of 20 conversations. Agreement was measured using Cohen's Kappa. The observed agreement levels were:

- **Coherence and Flow:** $\kappa = 0.7044$
- **Dialogue Realism:** $\kappa = 0.7391$
- **Complexity and Depth:** $\kappa = 0.7849$

The aggregated evaluation results are reported in Table 4.

Table 4: Conversation quality human evaluation results (1–5 scale). Higher is better.

| Chat Size | Coherence and Flow | Dialogue Realism | Complexity and Depth |
|:---:|:---:|:---:|:---:|
| 128K | 4.4 | 4.55 | 4.35 |
| 500K | 4.49 | 4.4 | 4.63 |
| 1M | 4.66 | 4.54 | 4.6 |
| 10M | 4.6 | 4.8 | 5 |
| **Average** | 4.53 | 4.57 | 4.64 |

## B.3 Benchmark Creation Details

### B.3.1 Domain Coverage of the Dataset

To ensure broad coverage and realism, our dataset spans a diverse set of domains. The collection includes both technical and non-technical conversations, ranging from specialized domains such as coding, mathematics, financial investment and health to personal and social domains such as therapy,

lifestyle, and trip planning. In total, we designed 100 multi-turn chats distributed across 19 domains, each represented by a set of distinct titles that capture the thematic scope of the dialogues. The full list of domains and their associated chat titles is provided in Table 5.

Table 5: Domains and associated chat titles in our dataset (100 total chats).

| Domain | Chat Titles |
|---|---|
| Coding | Designing a Large-Scale Retrieval-Augmented Generation (RAG) System for Enterprise Search ● Creating a Self-Driving Car Simulation Environment ● Developing a Multi-Agent AI Research Platform ● Building a Multi-Language AI Chatbot with Contextual Memory ● Developing a Personalized News Aggregator with AI Summarization ● Creating an Autonomous Stock Trading Bot ● Implementing a Custom Image Captioning Model ● Building a Multiplayer Online Game with Real-Time Physics ● Building a Real-Time Chat Application with Node.js and Socket.io ● Creating an AI-Powered Resume Analyzer with Python and NLP ● Developing a Computer Vision App for Real-Time Object Detection ● Creating a Restaurant Recommendation System ● Automating Social Media Posts with Python ● Building a Personal Budget Tracker Web App in Python and Flask ● Creating a Command-Line To-Do List Manager in Go ● Developing a Weather Forecast App in JavaScript with OpenWeather API ● Training a Spam Email Classifier Using Python and Scikit-learn ● Building a Portfolio Website with HTML, CSS, and Bootstrap |
| Math | Partial Differential Equations (PDEs) in Depth ● Functional Analysis and Infinite-Dimensional Spaces ● Solving Ordinary Differential Equations (ODEs) ● Deep Dive into Number Theory ● Advanced Probability and Combinatorics ● Exploring Non-Euclidean Geometry ● Studying Multivariable Calculus ● Diving into Analytic Geometry ● Developing Skills in Mathematical Induction ● Exploring Conic Sections in Depth ● Understanding Sequences and Series ● Mastering Basic Differential Calculus ● Exploring the Geometry of Triangles ● Understanding the Basics of Probability ● Mastering Algebraic Equations for Everyday Problem Solving ● Learning the Foundations of Trigonometry ● Mastering Fractions, Decimals, and Percentages |
| Writing Assistant & Learning | Building a Portfolio-Ready Resume that Passes Any Applicant Tracking System ● Mastering the Art of Persuasive Academic Essay Writing ● Crafting a Standout Cover Letter for Competitive Job Markets ● Designing a Multi-Purpose Personal Statement for Global Opportunities ● Developing a Self-Editing System for Lifelong Writing Improvement |
| Therapy & Emotional Support | Recovering from Workplace Burnout and Chronic Stress ● Healing After the Loss of a Loved One ● Overcoming Childhood Trauma and Rebuilding Self-Trust ● Coping with Post-Breakup Emotional Pain and Relationship Trauma |
| Career & Professional Development | Advancing from Mid-Level to Senior Leadership Roles ● Building a Powerful Professional Network from Scratch ● Landing Your Next Job: From Resume to Job Offer ● Designing a 5-Year Career Growth Plan ● Positioning Yourself for a Promotion |
| Financial Investment | Building a Long-Term Stock Market Investment Strategy ● Getting Started in Real Estate Investing ● Navigating the World of Cryptocurrency ● Creating a Balanced Investment Portfolio |
| Health & Wellness | Creating a Personalized Nutrition and Meal Planning System ● Designing a Sustainable Fitness Routine ● Improving Sleep Quality for Better Health ● Understanding and Managing Chronic Illness ● Recognizing Symptoms and Seeking Medical Help Early |

| Domain | Chat Titles |
| --- | --- |
| Relationship & Family | Strengthening Communication in Romantic Relationships • Parenting Through Different Life Stages • Navigating In-Law and Extended Family Relationships • Rebuilding Relationships After Trust Has Been Broken |
| Education & Learning | Learning to Play a Musical Instrument from Scratch • Mastering a New Language for Real-World Communication • Becoming a Skilled Photographer • Exploring Performing Arts: Acting, Theater, and Dance |
| Home & Real Estate | Buying Your First Home with Confidence • Renting a Home or Apartment Without Stress • Selling Your Home for Maximum Value • DIY Home Improvement and Repairs • Making Your Home More Comfortable and Functional |
| Lifestyle | Designing a Daily Routine That Boosts Productivity and Well-Being • Building Healthy and Sustainable Lifestyle Habits • Balancing Social Life and Personal Time |
| Cooking | Mastering Quick and Healthy Weeknight Dinners • Baking Like a Pro at Home • Exploring Global Cuisines from Your Kitchen • Cooking for Special Diets and Allergies • Meal Prepping for the Week Ahead |
| Business & Entrepreneurship | Starting a Business from Scratch • Growing and Scaling Your Small Business • Building a Successful Startup |
| Trip Planning | Preparing for a Week-Long Hiking and Camping Adventure in Patagonia • Organizing a Cross-Country USA Road Trip • Planning a Cultural Immersion Trip to Japan • Planning a Budget Backpacking Trip Across Southeast Asia • Arranging a Luxury Honeymoon in the Maldives |
| Sport | Soccer – Playing, Watching, and Supporting the World's Most Popular Game • Basketball – From Street Courts to the NBA • Volleyball – Indoor, Beach, and Competitive Play • Hockey – Ice, Field, and Global Competitions • Tennis – From Local Courts to Grand Slams |
| Event Planning | Planning a Surprise 30th Birthday Party for a Close Friend • Coordinating a Destination Beach Wedding for 100 Guests • Organizing a Weekend Community Food and Music Festival • Planning a Cozy Christmas Eve Dinner for Extended Family |
| Asking Recommendation | Finding the Perfect Smartphone for Photography and Gaming • Choosing a Lightweight Laptop for Work, Travel, and Entertainment • Selecting a Must-Read Fiction Series for Winter Evenings • Finding the Best Streaming Movies for a Family Weekend • Choosing Comfortable and Stylish Sneakers for Daily Wear |
| Legal & Administrative | Filing for a Marriage-Based Green Card in the United States • Creating a Legally Valid Will and Estate Plan • Applying for a Patent to Protect a New Invention |
| Philosophical & Ethical Discussion | Deciding Whether to Use AI to Automate Hiring in My Company • Considering Whether to Believe in and Live by the Idea of Free Will |

### B.3.2 CONVERSATION PLAN GENERATION

A *conversation plan* serves as the central scaffold of each conversation, providing a coherent storyline that evolves chronologically. The process of constructing conversation plans is anchored by a *seed* that specifies the *domain* of the dialogue (e.g., sports, finance, programming, mathematics), a *title* representing the high-level topic, and a *theme* that provides a more detailed instantiation of the title. The seed also includes a set of *subtopics*, which enumerate finer-grained subtopics and details to ensure topical diversity. However, a title, theme, and subtopics alone are insufficient to support detailed and information-rich conversations. To enrich the narrative, we introduce *narratives set* that define the evolving aspects of a conversation (e.g., career progression, goals, relationships). Each narrative is paired with descriptive details that specify its scope and trajectory.

In addition to the seed and narrative set, each conversation incorporates a *user profile*, a *relationship graph*, and an explicit *timeline*. The user profile includes attributes such as name, age, gender, location, profession, and personality traits. To avoid redundancy, personality traits are grounded in the Myers–Briggs Type Indicator (MBTI). Specifically, we randomly select six MBTI types, provide their descriptions, and instruct an LLM to synthesize a composite trait profile, enabling the creation of 8,008 unique user profiles. Relationship graphs are then constructed, linking the main user to family members (parents, partner, children), friends, and acquaintances, subject to constraints (e.g., plausible age gaps) to preserve realism. The timeline specifies the temporal span of the conversation, defining the range between its beginning and end.

In order to generate titles and themes of the chats, target domains are first specified by human. Given these domains, GPT-4.1 (OpenAI, 2025a) is prompted using the prompt shown in Listing 22 in Appendix H, to produce candidate titles, themes, and subtopics. These candidates are refined by human to ensure topical diversity by removing the similar chat titles and selecting diverse chat titles. Finally, for each conversation, we generate 15–20 narratives using open-source LLaMA-3.3 70B (AI, 2024) with the prompt shown in Listing 23 to save cost. In this prompt, given the conversation seed as input, the LLM produces narratives that capture evolving aspects of the storyline, providing the backbone for constructing coherent conversation plans.

Conversation plans are structured as a sequence of $N$ *sub-plans*, where each sub-plan corresponds to a distinct stage of the conversation. Each sub-plan contains a fixed number of $M$ *bullet-points*, and each bullet-point is defined by a *narrative* and a descriptive statement specifying how that narrative unfolds in the storyline. To maintain temporal coherence, each sub-plan also includes a *time anchor* specifying a concrete date or period.

For conversations of sizes 128K, 500K, and 1M tokens, a single conversation plan is generated, as shown in line 4 of Algorithm 1 in Appendix B.3.5. The plan is produced by conditioning the LLM on the conversation seed, user profile, relationship graph, timeline, the number of sub-plans, the number of bullet points within each sub-plan and narrative set, using the prompt shown in Listing 24 in Appendix H. The number of sub-plans is not fixed but varies with both the domain and the target conversation length, in order to adhere to the length budget. For instance, domains such as coding typically require fewer dialogue turns to reach the same token budget compared to more general domains.

For 10M-token conversations, a single plan cannot adequately capture the scope and continuity required at this scale. To address this, we construct ten distinct yet interlocking conversation plans that together produce a coherent long-term narrative. While the process begins with a main seed that defines the global topic and theme of the conversation, a single seed is insufficient for producing ten plans. Instead, we generate ten distinct conversation seeds—one for each plan—so that the narrative can unfold across multiple stages. The procedure for deriving these seeds—and the plans that follow—differs depending on the strategy. We propose two strategies for constructing them:

• **Sequential Expansion:** The conversation seed is used as the first seed in the sequence. The remaining seeds are generated to represent successive stages of the user's life, extending the storyline chronologically. For instance, if the main seed concerns an international trip, the first plan covers the trip itself, the second covers the period after returning (e.g., job search), and subsequent seeds correspond to later milestones. We generate these seeds using the prompt shown in Listing 28, which conditions on the main seed, user profile, and timeline to produce a sequence of temporally aligned seeds. Each conversation plan is then generated sequentially, with every plan conditioned on its predecessor to maintain continuity, as specified in line 12 of Algorithm 1

in Appendix B.3.5. The plans are generated using the prompt shown in Listing 30, yielding a temporally ordered series of interconnected narrative arcs. To maintain realism, the user's core relationships (e.g., parents, children, partner) remain fixed across plans, while new acquaintances are gradually introduced.

- **Hierarchical Decomposition:** Instead of extending the seed chronologically, the main seed is decomposed into ten sub-seeds, each corresponding to a distinct topical or temporal slice of the overall storyline. Together, these seeds span the full narrative. For example, if the main seed concerns an international trip, the first three seeds may cover preparation steps (e.g., reservations, document gathering), the next five capture events during the trip, and the final two represent post-trip activities (e.g., reflections, recounting experiences). Like in Sequential Expansion, the user's core relationships (e.g., parents, children, partner) remain fixed across plans, while new acquaintances are gradually introduced. We generate these ten sub-seeds using the prompt shown in Listing 29, which takes the main seed, user profile, and timeline, and outputs ten derived seeds.

Each plan is assigned explicit topical and temporal boundaries to prevent redundancy or thematic overlap, ensuring that sub-themes unfold in the correct stage of the narrative. These boundaries are encoded in the conversation seed itself. For coherence, summaries of all prior plans are provided to the LLM when generating a new plan, allowing contextual references to past events. Moreover, when generating each plan, future seeds are also supplied, encoding their own topical and temporal boundaries. This design allows earlier plans to anticipate upcoming events with consistent references (e.g., booking tickets for the correct travel dates before the trip actually occurs). This strategy is implemented in line 20 of Algorithm 1 in Appendix B.3.5. Conversation plans are generated using the prompt shown in Listing 31, which takes as input the main seed, the current sub-seed, the number of sub-plans, the narrative set, the user profile, core and newly introduced relationships, the preceding and subsequent sub-seeds, the previous plan, the summary of all previous plans, the index of the current sub-seed, and a binary indicator specifying whether the plan is the first in the sequence (in which case the introduction of the user is included). The output is a fully specified conversation plan.

After the conversation plan is constructed, it is expanded into user-turn questions and subsequently assistant responses, yielding complete dialogues that can be used to evaluate memory abilities. However, in its initial form, the plan may not include sufficient information to evaluate three critical memory abilities: *contradiction resolution*, *knowledge update*, and *instruction following*. To address this, after the initial plan generation, we pass the plan to GPT-4.1 to generate high-quality plans and augment each sub-plan with additional bullet points specifically designed to enable evaluation of these abilities. Importantly, this augmentation is performed in a second stage rather than during the initial plan generation, since incorporating such information directly in a single-pass generation leads to lower quality and less reliable coverage of these abilities. The augmentation is implemented using the prompt shown in Listing 27, which takes an existing conversation plan as input and outputs a revised version where each sub-plan includes three additional bullet points targeting these abilities.

### B.3.3 USER UTTERANCE GENERATION

Once conversation plans are constructed, user turns are synthesized directly from them. Each sub-plan within a conversation plan consists of $M$ bullet-points, which are partitioned into $K$ contiguous batches of equal size. Partitioning is performed sequentially, such that each batch corresponds to a consecutive segment of the sub-plan. Partitioning is necessary because conditioning the LLM on an entire sub-plan at once tends to yield repetitive or low-quality questions; batching mitigates this by narrowing the focus of generation. For each batch, the LLM produces $I$ user questions (line 6 of Algorithm 2 in Appendix B.3.5) using the prompt presented in Listing 32. The model is conditioned on the conversation seed, the current batch specification, preceding batches within the same sub-plan, and contextual information from earlier sub-plans. This setup ensures that generated questions remain grounded in prior context, yielding conversations that are coherent and continuous over extended spans.

The values of $K$ and $I$ vary depending on the domain and the target conversation length, in order to adhere to the overall length budget. We specify the values for $K$ and $I$ manually. The specific configurations of $K$ and $I$ across domains and conversation sizes are reported in Table 6. This provides fine-grained control over the density of user interactions and helps prevent both under-

generation and excessive redundancy. Additonally, to better capture domain-specific conversational patterns, we incorporate domain-specific features during question generation:

- **Programming:** To reflect realistic developer–assistant interactions, we incorporate questions that involve sharing code snippets. These include (i) buggy code requiring debugging assistance, (ii) correct code seeking optimization, and (iii) natural language descriptions of desired functionality for which code is requested. We use the prompt shown in Listing 33 to generate questions specific to the programming domain.

- **Mathematics.** To capture authentic problem-solving dynamics, we incorporate questions that involve sharing mathematical work, requesting corrections, asking for the next logical step in a solution, or introducing problems to be solved. We use the prompt shown in Listing 34 to generate questions specific to the mathematics domain.

To reduce computational cost while maintaining generation quality, question generation is performed using the open-source LLaMA-3.3 70B model (AI, 2024), which produces high-quality questions.

### B.3.4 ASSISTANT UTTERANCE GENERATION

After generating user-side questions, assistant-side responses are generated in an iterative, role-playing framework where one LLM assumes the *assistant role* and another assumes the *user role*. For each sub-plan, the assistant LLM is conditioned on the seed as explained in Section 2.2.1, prior sub-plans of the conversation plan, a summary of the most recent $M$ dialogue turns, and a compressed summary of older turns (generated using the prompt shown in Listing 37). For 10M-token conversations, additional summaries of prior plans are also provided.

The response generation process unfolds as an iterative interaction between the assistant and user roles. First, the assistant LLM produces an answer to the user's most recent question (line 9). This output is then analyzed by a *question-detection module*, which determines whether the assistant's response contains a counter-question directed at the user (line 11), using the prompt shown in Listing 35 that takes the assistant response as input and outputs yes if a question is present and no otherwise. If such a counter-question is detected, the response—together with the current and previous sub-plans, relevant past context, and conversation summaries—is passed to the user LLM, which generates a realistic reply that reflects the storyline and contextual details using the prompt shown in Listing 38 (line 14). This new user reply is subsequently passed back to the assistant LLM, continuing the conversation. This loop repeats until no further assistant questions are detected or the predefined threshold $\delta_1$ (which is set to two) is reached, preventing infinite cycles. For $\delta_1$ we tested values 2, 3 and 5 which we selected 2 as it produces more realistic dialogues.

Beyond direct question–answer exchanges, a *follow-up detection module* (line 21) evaluates whether, in a realistic setting, the user would naturally ask a clarifying or elaborative follow-up. The need for a follow-up is determined using the prompt shown in Listing 36, which takes as input the seed, dialogue history, and the assistant's most recent response, and outputs yes or no. This decision is guided by factors such as subject complexity, ambiguity in the assistant's answer, or incompleteness of the response. When a follow-up is required, the module conditions on the seed, the current and prior sub-plans, the most recent $M$ turns, and summaries of earlier turns to generate the follow-up query using the prompt shown in Listing 39. The generated query is then passed back to the assistant LLM for resolution. As with the assistant-question loop, a strict threshold $\delta_2$ (which is set to two like $\delta_1$) limits the number of follow-up exchanges, preventing unbounded cycles.

Through the interaction of these two threshold-controlled modules, the system produces conversations that exhibit naturalistic bidirectional dynamics, rich contextual references, and realistic clarification behaviors characteristic of human–AI dialogues.

---

**Algorithm 1** Conversation plan generation.

---

**Input:** domain $c$, length budget $L$, title $\theta$, theme $\tau$, subtopics $\Sigma$, user profile $u$, user relationships $\rho$, timeline $\Gamma$, number of conversation sub-plans $N$, number of bullet-points in each conversation sub-plan $M$, generator $G$
**Output:** Conversation plan set $p$

1:  $S \leftarrow (c, \theta, \tau, \Sigma)$                                                                     ▷ Initialize seed
2:  **if** $L \in \{128K, 500K, 1M\}$ **then**
3:      $\Lambda \leftarrow G(S)$                                                               ▷ Generate narratives using Listing 23
4:      $P \leftarrow G(S, u, \rho, \Gamma, N, M, \Lambda)$             ▷ Generate a single conversation plan with Listing 24
5:  **else if** $L = 10M$ **then**
6:      $P \leftarrow \{\}$                                                                     ▷ Initialize set of plans
7:      **if** $\sigma =$ Sequential Expansion **then**
8:          $S' \leftarrow G_{\text{seeds}}(S, \Gamma)$                                  ▷ Generate sequential sub-seeds with Listing 28
9:          **for** each $s'_i \in S'$ **do**
10:             $\Lambda_i \leftarrow G(s'_i)$                                              ▷ Generate narratives for sub-seed
11:             $b \leftarrow \mathbf{1}[i = 0]$                                    ▷ Binary indicator: 1 if first plan, else 0
12:             $P_i \leftarrow G(s'_i, \Gamma_i, N, \Lambda_i, u, \rho, P_{i-1}, i, b)$        ▷ Generate plan with Listing 30
13:             $P \leftarrow P \cup \{P_i\}$
14:         **end for**
15:     **else if** $\sigma =$ Hierarchical Decomposition **then**
16:         $S' \leftarrow G_{\text{decompose}}(S, \Gamma)$                               ▷ Decompose seed with Listing 29
17:         **for** each $s'_i \in S'$ **do**
18:             $\Lambda_i \leftarrow G(s'_i)$                                              ▷ Generate narratives for sub-seed
19:             $b \leftarrow \mathbf{1}[i = 0]$
20:             $P_i \leftarrow G(S, S', s'_i, \Gamma_i, N, \Lambda_i, u, \rho, P_{i-1}, \overline{P_{0,\dots,i-1}}, i, b)$       ▷ Generate plan with
    Listing 31
21:             $P \leftarrow P \cup \{P_i\}$
22:         **end for**
23:     **end if**
24: **end if**
25: **return** $P$

---

**Algorithm 2** User questions generation.

---

**Input:** seed $S$, conversation plan $p$, number of questions per iteration $I$, generator $G$
**Output:** Question set $Q$

1:  $p \leftarrow \{p_1, \dots, p_N\}$                                               ▷ Conversation plan with $N$ sub-plans
2:  $Q \leftarrow \{\}$                                                            ▷ Initialize empty question set
3:  **for** each $p_i \in P$ **do**
4:      $p_i = \{p_{i1}, \dots, p_{iK}\}$
5:      **for** each $p_{ij} \in p_i$ **do**
6:          $Q_{ij} \leftarrow G(S, p_{ij}, \{p_{i1}, \dots, p_{i(j-1)}\}, \{p_1, \dots, p_{i-1}\}, I)$        ▷ Generate $I$ questions using
    Listing 32
7:          $Q \leftarrow Q \cup \{Q_{ij}\}$                                   ▷ Append generated questions to the question set
8:      **end for**
9:  **end for**
10: **return** $Q$

---

### B.3.5 ALGORITHMS

---

**Algorithm 3** Answer generation.

---

**Input:** question set $Q = \{Q_1, \ldots, Q_N\}$, seed $S$, conversation plan set $P$, thresholds $\delta_1, \delta_2$, assistant-question detector $\phi$, follow-up detector $\psi$, generator $G$
**Output:** conversation list $T$

1: $\mathcal{T} \leftarrow \{\}$ ▷ Initialize empty conversation list
2: **for** each $Q_i \in Q$ **do**
3:   $Q_i = \{q_1, \ldots, q_J\}$ ▷ Questions in sub-plan $i$
4:   **for** each $q_j \in Q_i$ **do**
5:    $t \leftarrow \{\}$ ▷ Initialize turn sequence
6:    $H_t^{(M)} \leftarrow$ recent-$M$ turn window at turn $t$
7:    $\overline{H}_t \leftarrow$ summary of turns prior to $H_t^{(M)}$
8:    $\overline{P}^{(<p)} \leftarrow$ summaries of conversation plans preceding $p$
9:    $a_{ij} \leftarrow G_{\text{assistant}}(S, p_{1:(i-1)}, H_t^{(M)}, \overline{H}_t, \overline{P}^{(<p)})$ ▷ Generate assistant response with Listing 37
10:    $t \leftarrow t \cup \{a_{ij}\}$ ▷ Add assistant's response to current dialogues turn
11:    $isQ \leftarrow \phi(a_{ij}, H_t^{(M)}, \overline{H}_t)$ ▷ Checks if assistant response contains question from user with Listing 35
12:    $count \leftarrow 0$
13:    **while** $isQ$ **and** $count < \delta_1$ **do**
14:     $u_{ij} \leftarrow G_{\text{user}}(S, p_i, p_{1:(i-1)}, \overline{P}^{(<p)}, H_t^{(M)}, \overline{H}_t, a_{ij})$ ▷ Generate user's response to assistant question with Listing 38
15:     $t \leftarrow t \cup \{u_{ij}\}$ ▷ Add user's response to current dialogues turn
16:     $a_{ij} \leftarrow G_{\text{assistant}}(S, p_{1:(i-1)}, H_t^{(M)}, \overline{H}_t, \overline{P}^{(<p)})$ ▷ Generate assistant's response
17:     $t \leftarrow t \cup \{a_{ij}\}$ ▷ Add assistant's response to current dialogues turn
18:     $count \leftarrow count + 1$
19:     $isQ \leftarrow \phi(a_{ij}, H_t^{(M)}, \overline{H}_t)$
20:    **end while**
21:    $needFU \leftarrow \psi(a_{ij}, H_t^{(M)}, \overline{H}_t, S)$ ▷ Checks if user need to ask followup question with Listing 36
22:    $fu\_count \leftarrow 0$
23:    **while** $needFU$ **and** $fu\_count < \delta_2$ **do**
24:     $u_{ij} \leftarrow G_{\text{user}}(S, p_i, p_{1:(i-1)}, \overline{P}^{(<p)}, H_t^{(M)}, \overline{H}_t, a_{ij})$ ▷ Generate user's followup question with Listing 39
25:     $t \leftarrow t \cup \{u_{ij}\}$
26:     $a_{ij} \leftarrow G_{\text{assistant}}(S, p_{1:(i-1)}, H_t^{(M)}, \overline{H}_t, \overline{P}^{(<p)})$ ▷ Generate assistant's response to user's followup question
27:     $t \leftarrow t \cup \{a_{ij}\}$
28:     $fu\_count \leftarrow fu\_count + 1$
29:     $needFU \leftarrow \psi(a_{ij}, H_t^{(M)}, \overline{H}_t, S)$
30:    **end while**
31:    $\mathcal{T} \leftarrow \mathcal{T} \cup \{t\}$
32:   **end for**
33: **end for**
34: **return** $T$

---

## B.4 USER UTTERANCE GENERATION HYPERPARAMETERS

Table 6: Batching configuration by chat size and domain category for user-turn question generation. NUM_SUBPLANS denotes the number of conversation sub-plans, $K$ the number of batches per sub-plan, and $I$ the number of questions generated per batch.

| Chat Size | Category | NUM_SUBPLANS | K | I |
|---|---|---|---|---|
| 128K | General | 5 | 10 | 2 |
| | Coding | 3 | 23 | 1 |
| | Math | 3 | 25 | 1 |
| 500K | General | 10 | 10 | 4 |
| | Coding | 10 | 10 | 3 |
| | Math | 10 | 10 | 4 |
| 1M | General | 10 | 10 | 9 |
| | Coding | 10 | 10 | 6 |
| | Math | 10 | 10 | 6 |
| 10M | General | 10 | 10 | 9 |
| | Coding | 10 | 10 | 6 |
| | Math | 10 | 10 | 6 |

## B.5 CREATED PROBING QUESTIONS DISTRIBUTION

We measure which parts of the dialogue contain the information required to answer the probing questions. To this end, each conversation is divided into ten equal segments, and we record the segment(s) where the supporting evidence for each probing question resides. The detailed methodology for aligning probing questions with dialogue segments is described in Section 2.3. The resulting distributions across conversation lengths are reported in Table 7.

Table 7: Percentage distribution of created probing questions across ten equal chat segments (deciles) for different chat sizes. Each row corresponds to a segment of the dialogue, moving from the beginning (Segment 1) to the end (Segment 10).

| Chat Segment (Decile) | 100K | 500K | 1M | 10M |
|---|---|---|---|---|
| 1 | 0.00% | 0.65% | 0.19% | 0.00% |
| 2 | 11.05% | 23.70% | 21.60% | 10.24% |
| 3 | 14.83% | 15.91% | 20.11% | 16.27% |
| 4 | 12.79% | 14.45% | 15.83% | 15.06% |
| 5 | 13.08% | 7.95% | 9.50% | 14.46% |
| 6 | 13.37% | 9.09% | 8.01% | 9.64% |
| 7 | 11.92% | 6.33% | 5.96% | 10.24% |
| 8 | 8.14% | 5.52% | 5.21% | 13.25% |
| 9 | 9.59% | 4.55% | 4.47% | 8.43% |
| 10 | 5.23% | 11.85% | 9.12% | 2.41% |

### B.6 MEMORY ABILITIES EXAMPLES

To illustrate how our benchmark evaluates different aspects of long-term conversational memory, we provide representative probing questions and their ideal answers for each of the ten memory abilities. These examples demonstrate how each ability is operationalized in practice.

1. **Abstention (withholding answers when information is missing)**

   **Probing Question:** What specific advice did Manuel give about property management companies during the March 5 Investors Meetup?
   **Ideal Answer:** Based on the provided chat, there is no information related to the specific advice Manuel gave about property management companies.

2. **Contradiction Resolution (detecting and reconciling inconsistencies)**

   **Probing Question:** Have I ever attended any real estate webinars or investor meetups?
   **Ideal Answer:** I notice you've mentioned contradictory information about this. You said you have never attended any real estate webinars or investor meetups, but you also mentioned attending a webinar about Turkey's rising demand for multi-family rentals. Which statement is correct?

3. **Event Ordering (reasoning about chronological sequence)**

   **Probing Question:** How did my focus on different aspects of property investment and management develop throughout our conversations in order? Mention ONLY and ONLY ten items.
   **Ideal Answer:** Your focus developed in this sequence: 1) Agent interaction and preparation, 2) Financial evaluation, 3) Financing and mortgage, 4) Contractor management, 5) Zoning and permits, 6) Renovation planning, 7) Viewing preparation, 8) Negotiation and cash flow, 9) Renovation completion, 10) Commercial property considerations.

4. **Information Extraction (recalling factual details)**

   **Probing Question:** What amount did I say I had set aside as my initial capital for investing?
   **Ideal Answer:** You said you had set aside $50,000 as your initial capital.

5. **Instruction Following (adhering to explicit user constraints)**

   **Probing Question:** How should I allocate my funds for the upcoming quarter?
   **Ideal Answer:** The response should include an itemized list of costs, category-by-category breakdown, and detailed cost analysis, reflecting the instruction to always provide detailed financial breakdowns when budgeting.

6. **Information Update (revising prior facts when updated)**

   **Probing Question:** How much initial capital have I allocated for real estate investing?
   **Ideal Answer:** $60,000

7. **Multi-hop Reasoning (integrating evidence across turns)**

   **Probing Question:** How many different banks did I consider for my mortgage options across my conversations?
   **Ideal Answer:** Two banks: Halkbank and Ziraat Bank.

8. **Preference Following (adapting to evolving user preferences)**

> **Probing Question:** I'm looking at several properties and want to understand how to approach comparing them effectively. What steps would you suggest I take?
> **Ideal Answer:** A structured, step-by-step process that emphasizes data collection and analysis, breaking comparisons into measurable factors, reflecting the user's preference for methodical, data-driven decisions.

9. **Summarization (abstraction and compression of content)**

> **Probing Question:** Can you give me a comprehensive summary of my journey and decision-making process around investing in rental properties, including how my budget, property choices, management considerations, and financing plans have developed over time?
> **Ideal Answer:** Your journey began with an initial capital of $50,000, followed by exploration of market conditions, renovation planning, property type tradeoffs, financing through Halkbank and Ziraat Bank, and a structured plan for purchase and management.

10. **Temporal Reasoning (reasoning about durations and timelines)**

> **Probing Question:** How many days are there between my first property viewing with Mehmet Yilmaz and the last one I scheduled?
> **Ideal Answer:** There are 2 days between the first property viewing on March 25 and the last one on March 27.

## C  DETAILED EXPERIMENTS

### C.1  ABLATION STUDY

In this section, we present the complete results of our ablation experiments. All ablations are conducted using Qwen2.5-32B-AWQ as the base model. We evaluate the contribution of individual components in our proposed module as shown in table 8.

Table 8: Ablation study showing the impact of removing key memory components (retrieval, scratchpad, working memory, and noise filtering) on performance across various conversation lengths (100K–10M).

| Length | Memory Ability | Base | w/o Retrieval from Index | w/o Scratchpad | w/o Working Memory | w/o Noise Filtering |
|---|---|---|---|---|---|---|
| 100K | Abstention | 0.475 | 0.725 | 0.600 | 0.575 | 0.700 |
| | Contradiction Resolution | 0.037 | 0.043 | 0.012 | 0.043 | 0.018 |
| | Event Ordering | 0.205 | 0.190 | 0.194 | 0.220 | 0.200 |
| | Information Extraction | 0.479 | 0.329 | 0.510 | 0.451 | 0.485 |
| | Instruction Following | 0.362 | 0.375 | 0.287 | 0.387 | 0.312 |
| | Knowledge Update | 0.362 | 0.237 | 0.350 | 0.362 | 0.312 |
| | Multi-Hop Reasoning | 0.281 | 0.201 | 0.248 | 0.303 | 0.181 |
| | Preference Following | 0.566 | 0.675 | 0.533 | 0.579 | 0.491 |
| | Summarization | 0.232 | 0.232 | 0.143 | 0.223 | 0.103 |
| | Temporal Reasoning | 0.112 | 0.075 | 0.125 | 0.125 | 0.087 |
| | Average | 0.311 | 0.311 | 0.300 | **0.327** | 0.289 |
| 500K | Abstention | 0.571 | 0.571 | 0.585 | 0.657 | 0.585 |
| | Contradiction Resolution | 0.017 | 0.007 | 0.014 | 0.017 | 0.014 |
| | Event Ordering | 0.244 | 0.222 | 0.266 | 0.262 | 0.229 |
| | Information Extraction | 0.506 | 0.254 | 0.466 | 0.485 | 0.464 |
| | Instruction Following | 0.295 | 0.307 | 0.316 | 0.334 | 0.286 |
| | Knowledge Update | 0.278 | 0.192 | 0.285 | 0.235 | 0.314 |
| | Multi-Hop Reasoning | 0.214 | 0.104 | 0.227 | 0.192 | 0.247 |
| | Preference Following | 0.571 | 0.553 | 0.450 | 0.547 | 0.465 |
| | Summarization | 0.344 | 0.312 | 0.225 | 0.353 | 0.203 |
| | Temporal Reasoning | 0.121 | 0.042 | 0.116 | 0.114 | 0.130 |
| | Average | 0.316 | 0.256 | 0.295 | **0.320** | 0.294 |
| 1M | Abstention | 0.500 | 0.664 | 0.600 | 0.557 | 0.507 |
| | Contradiction Resolution | 0.021 | 0.021 | 0.035 | 0.042 | 0.032 |
| | Event Ordering | 0.200 | 0.215 | 0.221 | 0.227 | 0.199 |
| | Information Extraction | 0.366 | 0.246 | 0.391 | 0.397 | 0.366 |
| | Instruction Following | 0.419 | 0.427 | 0.335 | 0.384 | 0.351 |
| | Knowledge Update | 0.357 | 0.185 | 0.321 | 0.400 | 0.285 |
| | Multi-Hop Reasoning | 0.209 | 0.129 | 0.227 | 0.221 | 0.169 |
| | Preference Following | 0.551 | 0.602 | 0.536 | 0.597 | 0.540 |
| | Summarization | 0.316 | 0.310 | 0.169 | 0.330 | 0.128 |
| | Temporal Reasoning | 0.154 | 0.050 | 0.111 | 0.121 | 0.111 |
| | Average | 0.309 | 0.285 | 0.295 | **0.328** | 0.269 |
| 10M | Abstention | 0.550 | 0.800 | 0.650 | 0.650 | 0.600 |
| | Contradiction Resolution | 0.012 | 0.000 | 0.012 | 0.000 | 0.000 |
| | Event Ordering | 0.197 | 0.199 | 0.199 | 0.209 | 0.181 |
| | Information Extraction | 0.350 | 0.000 | 0.200 | 0.150 | 0.200 |
| | Instruction Following | 0.350 | 0.175 | 0.175 | 0.175 | 0.050 |
| | Knowledge Update | 0.275 | 0.050 | 0.300 | 0.150 | 0.225 |
| | Multi-Hop Reasoning | 0.125 | 0.000 | 0.125 | 0.125 | 0.075 |
| | Preference Following | 0.308 | 0.191 | 0.241 | 0.200 | 0.175 |
| | Summarization | 0.220 | 0.119 | 0.068 | 0.008 | 0.050 |
| | Temporal Reasoning | 0.000 | 0.000 | 0.050 | 0.075 | 0.000 |
| | Average | **0.238** | 0.153 | 0.202 | 0.181 | 0.155 |

### C.2  RETRIEVAL BUDGET

We investigate the impact of the retrieval budget through two sets of experiments: (i) varying the retrieval depth by setting the number of retrieved documents $K \in \{5, 10, 15, 20\}$, and (ii) comparing a dense retriever against a sparse retriever (SPLADE).

The full results examining the effect of different retrieval depths (number of retrieved documents) are presented in Table 9.

Table 9: Effect of retrieval depth on performance across conversation lengths (100K–10M) and memory abilities. Results are shown for different numbers of retrieved documents ($K \in \{5, 10, 15, 20\}$).

| Length | Memory Ability | K=5 | K=10 | K=15 | K=20 |
|---|---|---|---|---|---|
| 100K | Abstention | 0.475 | 0.500 | 0.625 | 0.625 |
| | Contradiction Resolution | 0.037 | 0.025 | 0.025 | 0.031 |
| | Event Ordering | 0.205 | 0.191 | 0.218 | 0.210 |
| | Information Extraction | 0.479 | 0.450 | 0.412 | 0.391 |
| | Instruction Following | 0.362 | 0.362 | 0.475 | 0.462 |
| | Knowledge Update | 0.362 | 0.375 | 0.350 | 0.300 |
| | Multi-Hop Reasoning | 0.281 | 0.322 | 0.321 | 0.309 |
| | Preference Following | 0.566 | 0.591 | 0.562 | 0.575 |
| | Summarization | 0.232 | 0.231 | 0.218 | 0.213 |
| | Temporal Reasoning | 0.112 | 0.162 | 0.137 | 0.137 |
| | Average | 0.311 | 0.321 | **0.334** | 0.325 |
| 500K | Abstention | 0.571 | 0.514 | 0.614 | 0.642 |
| | Contradiction Resolution | 0.017 | 0.021 | 0.071 | 0.071 |
| | Event Ordering | 0.244 | 0.229 | 0.238 | 0.247 |
| | Information Extraction | 0.506 | 0.531 | 0.503 | 0.507 |
| | Instruction Following | 0.295 | 0.341 | 0.390 | 0.373 |
| | Knowledge Update | 0.278 | 0.307 | 0.326 | 0.326 |
| | Multi-Hop Reasoning | 0.214 | 0.188 | 0.234 | 0.213 |
| | Preference Following | 0.571 | 0.597 | 0.628 | 0.607 |
| | Summarization | 0.344 | 0.354 | 0.375 | 0.376 |
| | Temporal Reasoning | 0.121 | 0.128 | 0.121 | 0.135 |
| | Average | 0.316 | 0.321 | **0.350** | 0.350 |
| 1M | Abstention | 0.500 | 0.521 | 0.600 | 0.585 |
| | Contradiction Resolution | 0.021 | 0.021 | 0.057 | 0.053 |
| | Event Ordering | 0.200 | 0.224 | 0.240 | 0.242 |
| | Information Extraction | 0.366 | 0.398 | 0.377 | 0.391 |
| | Instruction Following | 0.419 | 0.476 | 0.439 | 0.446 |
| | Knowledge Update | 0.357 | 0.350 | 0.400 | 0.407 |
| | Multi-Hop Reasoning | 0.209 | 0.189 | 0.209 | 0.190 |
| | Preference Following | 0.551 | 0.596 | 0.535 | 0.514 |
| | Summarization | 0.316 | 0.317 | 0.325 | 0.351 |
| | Temporal Reasoning | 0.154 | 0.154 | 0.119 | 0.199 |
| | Average | 0.309 | 0.325 | **0.330** | 0.330 |
| 10M | Abstention | 0.550 | 0.600 | 0.650 | 0.600 |
| | Contradiction Resolution | 0.012 | 0.012 | 0.025 | 0.025 |
| | Event Ordering | 0.197 | 0.210 | 0.213 | 0.236 |
| | Information Extraction | 0.350 | 0.150 | 0.300 | 0.300 |
| | Instruction Following | 0.350 | 0.150 | 0.450 | 0.400 |
| | Knowledge Update | 0.275 | 0.200 | 0.300 | 0.300 |
| | Multi-Hop Reasoning | 0.125 | 0.100 | 0.125 | 0.150 |
| | Preference Following | 0.308 | 0.175 | 0.275 | 0.275 |
| | Summarization | 0.220 | 0.089 | 0.196 | 0.164 |
| | Temporal Reasoning | 0.000 | 0.025 | 0.000 | 0.000 |
| | Average | 0.238 | 0.171 | **0.253** | 0.245 |

In a complementary experiment, we analyzed the impact of retriever choice. Our base architecture employs a dense retriever, which we compare against the sparse SPLADE-V2 retriever (Formal et al., 2022). As shown in Figure 5 in Appendix C.2, SPLADE yields performance gains of 1.7% at 100K tokens, 0.7% at 500K, and 0.8% at 1M, but results in a slight performance drop of 0.7% at 10M. On average, the sparse retriever provides a modest improvement across conversation lengths. Complete results comparing the dense retriever with SPLADE are presented in Table 10.

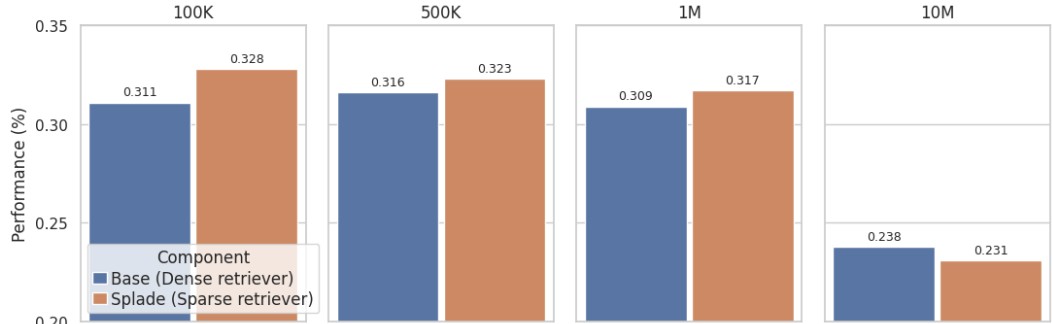

Figure 5: Performance comparison between dense retrieval and sparse retrieval (SPLADE) in LIGHT.

Table 10: Comparison of dense and sparse retrieval strategies across conversation lengths (100K–1M) and ten memory abilities. The table reports performance when using the default dense retriever versus a sparse retriever (SPLADE).

| Length | Memory Ability | Base (Dense retriever) | Sparse retriever (SPLADE) |
|---|---|---|---|
| 100K | Abstention | 0.475 | 0.525 |
| | Contradiction Resolution | 0.037 | 0.43 |
| | Event Ordering | 0.205 | 0.181 |
| | Information Extraction | 0.479 | 0.596 |
| | Instruction Following | 0.362 | 0.400 |
| | Knowledge Update | 0.362 | 0.350 |
| | Multi-Hop Reasoning | 0.281 | 0.267 |
| | Preference Following | 0.566 | 0.562 |
| | Summarization | 0.232 | 0.230 |
| | Temporal Reasoning | 0.112 | 0.125 |
| | Average | 0.311 | **0.328** |
| 500K | Abstention | 0.571 | 0.557 |
| | Contradiction Resolution | 0.017 | 0.025 |
| | Event Ordering | 0.244 | 0.226 |
| | Information Extraction | 0.506 | 0.559 |
| | Instruction Following | 0.295 | 0.345 |
| | Knowledge Update | 0.278 | 0.307 |
| | Multi-Hop Reasoning | 0.214 | 0.212 |
| | Preference Following | 0.571 | 0.565 |
| | Summarization | 0.344 | 0.330 |
| | Temporal Reasoning | 0.121 | 0.107 |
| | Average | 0.316 | **0.323** |
| 1M | Abstention | 0.500 | 0.564 |
| | Contradiction Resolution | 0.021 | 0.028 |
| | Event Ordering | 0.200 | 0.196 |
| | Information Extraction | 0.366 | 0.392 |
| | Instruction Following | 0.419 | 0.401 |
| | Knowledge Update | 0.357 | 0.371 |
| | Multi-Hop Reasoning | 0.209 | 0.193 |
| | Preference Following | 0.551 | 0.595 |
| | Summarization | 0.316 | 0.300 |
| | Temporal Reasoning | 0.154 | 0.133 |
| | Average | 0.309 | **0.317** |
| 10M | Abstention | 0.550 | 0.700 |
| | Contradiction Resolution | 0.012 | 0.000 |
| | Event Ordering | 0.197 | 0.202 |
| | Information Extraction | 0.350 | 0.350 |
| | Instruction Following | 0.350 | 0.250 |
| | Knowledge Update | 0.275 | 0.375 |
| | Multi-Hop Reasoning | 0.125 | 0.125 |
| | Preference Following | 0.308 | 0.200 |
| | Summarization | 320.220 | 0.090 |
| | Temporal Reasoning | 0.000 | 0.025 |
| | Average | **0.238** | 0.231 |

## C.3 EFFECT OF EMBEDDING MODEL CHOICE

We also examined how the choice of embedding model affects the performance of both the RAG baseline and the episodic memory component of LIGHT. In the primary experiments, we used the *BAAI/bge-small-en-v1.5* embedding model. To assess robustness, we re-ran all experiments using the larger *BAAI/bge-large-en-v1.5* model while keeping the LLM reader fixed to *GPT-4.1-nano*. As shown in Table 11, LIGHT consistently outperforms the RAG baseline under both embedding configurations. Moreover, LIGHT exhibits larger gains when switching to the higher-capacity embedding model, achieving an additional 2.08% improvement at the 1M-token setting and 16.37% at the 10M-token setting. These results indicate that LIGHT is robust to changes in embedding quality and can effectively leverage stronger embedding models to enhance long-term memory abilities.

Table 11: Effect of the embedding model on performance across conversation lengths (100K–10M) and ten memory abilities. Results are shown for the RAG baseline and LIGHT using two different embedding models.

| Length | Memory Ability | RAG (bge-small) | Ours (bge-small) | RAG (bge-large) | Ours (bge-large) |
|---|---|---|---|---|---|
| 100K | Abstention | 0.800 | 0.575 | 0.825 | 0.600 |
| | Contradiction Resolution | 0.018 | 0.031 | 0.012 | 0.031 |
| | Event Ordering | 0.169 | 0.177 | 0.185 | 0.171 |
| | Information Extraction | 0.362 | 0.538 | 0.404 | 0.562 |
| | Instruction Following | 0.350 | 0.400 | 0.337 | 0.462 |
| | Knowledge Update | 0.375 | 0.375 | 0.325 | 0.375 |
| | Multi-Hop Reasoning | 0.263 | 0.365 | 0.224 | 0.341 |
| | Preference Following | 0.550 | 0.625 | 0.537 | 0.562 |
| | Summarization | 0.083 | 0.202 | 0.089 | 0.148 |
| | Temporal Reasoning | 0.125 | 0.162 | 0.112 | 0.162 |
| | Average | 0.309 | **0.345** | 0.305 | 0.341 |
| 500K | Abstention | 0.828 | 0.600 | 0.814 | 0.571 |
| | Contradiction Resolution | 0.025 | 0.035 | 0.028 | 0.032 |
| | Event Ordering | 0.180 | 0.204 | 0.178 | 0.202 |
| | Information Extraction | 0.382 | 0.491 | 0.345 | 0.454 |
| | Instruction Following | 0.286 | 0.342 | 0.303 | 0.363 |
| | Knowledge Update | 0.288 | 0.240 | 0.380 | 0.321 |
| | Multi-Hop Reasoning | 0.233 | 0.266 | 0.272 | 0.282 |
| | Preference Following | 0.577 | 0.684 | 0.571 | 0.650 |
| | Summarization | 0.184 | 0.334 | 0.153 | 0.316 |
| | Temporal Reasoning | 0.161 | 0.154 | 0.126 | 0.126 |
| | Average | 0.314 | **0.335** | 0.317 | 0.331 |
| 1M | Abstention | 0.778 | 0.678 | 0.771 | 0.657 |
| | Contradiction Resolution | 0.028 | 0.021 | 0.021 | 0.025 |
| | Event Ordering | 0.179 | 0.211 | 0.194 | 0.211 |
| | Information Extraction | 0.399 | 0.410 | 0.360 | 0.439 |
| | Instruction Following | 0.271 | 0.394 | 0.269 | 0.421 |
| | Knowledge Update | 0.342 | 0.392 | 0.371 | 0.378 |
| | Multi-Hop Reasoning | 0.293 | 0.278 | 0.204 | 0.254 |
| | Preference Following | 0.513 | 0.576 | 0.497 | 0.598 |
| | Summarization | 0.152 | 0.290 | 0.116 | 0.296 |
| | Temporal Reasoning | 0.064 | 0.107 | 0.119 | 0.150 |
| | Average | 0.302 | 0.336 | 0.292 | **0.343** |
| 10M | Abstention | 0.650 | 0.400 | 0.800 | 0.550 |
| | Contradiction Resolution | 0.012 | 0.025 | 0.025 | 0.037 |
| | Event Ordering | 0.201 | 0.173 | 0.203 | 0.171 |
| | Information Extraction | 0.300 | 0.350 | 0.300 | 0.450 |
| | Instruction Following | 0.175 | 0.250 | 0.175 | 0.275 |
| | Knowledge Update | 0.325 | 0.300 | 0.325 | 0.300 |
| | Multi-Hop Reasoning | 0.091 | 0.135 | 0.066 | 0.075 |
| | Preference Following | 0.366 | 0.425 | 0.316 | 0.525 |
| | Summarization | 0.063 | 0.179 | 0.100 | 0.224 |
| | Temporal Reasoning | 0.000 | 0.025 | 0.000 | 0.025 |
| | Average | 0.218 | 0.226 | 0.231 | **0.263** |

## C.4 EFFECT OF INDEXING SETUP

We also investigated the effect of the vector database indexing setup on the performance of LIGHT. In the primary experiments, we used *IndexFlatIP*, and in the experiments below, we examined the

effect of switching the indexing setup to *IndexHNSWFlat*. For this experiment, the reader LLM was GPT-4.1-nano. The results are shown in Table 12.

Table 12: Effect of vector database indexing setup on performance across conversation lengths (100K–10M) and ten memory abilities. Results are reported for LIGHT under two different indexing configurations.

| Length | Memory Ability | Ours (IndexFlatIP) | Ours (IndexHNSWFlat) |
|---|---|---|---|
| | Abstention | 0.575 | 0.600 |
| | Contradiction Resolution | 0.031 | 0.031 |
| | Event Ordering | 0.177 | 0.173 |
| | Information Extraction | 0.538 | 0.565 |
| | Instruction Following | 0.400 | 0.375 |
| 100K | Knowledge Update | 0.375 | 0.400 |
| | Multi-Hop Reasoning | 0.365 | 0.285 |
| | Preference Following | 0.625 | 0.662 |
| | Summarization | 0.202 | 0.217 |
| | Temporal Reasoning | 0.162 | 0.162 |
| | Average | 0.345 | **0.347** |
| | Abstention | 0.600 | 0.528 |
| | Contradiction Resolution | 0.035 | 0.032 |
| | Event Ordering | 0.204 | 0.207 |
| | Information Extraction | 0.491 | 0.503 |
| | Instruction Following | 0.342 | 0.332 |
| 500K | Knowledge Update | 0.240 | 0.226 |
| | Multi-Hop Reasoning | 0.266 | 0.269 |
| | Preference Following | 0.684 | 0.666 |
| | Summarization | 0.334 | 0.317 |
| | Temporal Reasoning | 0.154 | 0.176 |
| | Average | **0.335** | 0.325 |
| | Abstention | 0.678 | 0.578 |
| | Contradiction Resolution | 0.021 | 0.025 |
| | Event Ordering | 0.211 | 0.211 |
| | Information Extraction | 0.410 | 0.420 |
| | Instruction Following | 0.394 | 0.386 |
| 1M | Knowledge Update | 0.392 | 0.385 |
| | Multi-Hop Reasoning | 0.278 | 0.278 |
| | Preference Following | 0.576 | 0.567 |
| | Summarization | 0.290 | 0.257 |
| | Temporal Reasoning | 0.107 | 0.128 |
| | Average | **0.336** | 0.324 |
| | Abstention | 0.400 | 0.600 |
| | Contradiction Resolution | 0.025 | 0.025 |
| | Event Ordering | 0.173 | 0.168 |
| | Information Extraction | 0.350 | 0.350 |
| | Instruction Following | 0.250 | 0.300 |
| 10M | Knowledge Update | 0.300 | 0.225 |
| | Multi-Hop Reasoning | 0.135 | 0.075 |
| | Preference Following | 0.425 | 0.433 |
| | Summarization | 0.179 | 0.194 |
| | Temporal Reasoning | 0.025 | 0.000 |
| | Average | 0.226 | **0.237** |

## C.5 SUPPLEMENTARY BASELINE EVALUATION

Alongside long-context LLMs and RAG, we also evaluated ReadAgent (Lee et al., 2024), another method designed to enhance long-term memory in LLMs, on BEAM and compared it with LIGHT. The results demonstrate that LIGHT consistently outperforms ReadAgent across all four conversation lengths (100K, 500K, 1M, and 10M). The full results are shown in Table 13.

Table 13: Comparing LIGHT with ReadAgent across conversation lengths (100K–10M) and ten memory abilities.

| Length | Memory Ability | ReadAgent | Ours (LIGHT) |
|---|---|---|---|
| | Abstention | 0.850 | 0.475 |
| | Contradiction Resolution | 0.000 | 0.037 |
| | Event Ordering | 0.200 | 0.205 |
| | Information Extraction | 0.066 | 0.479 |
| | Instruction Following | 0.237 | 0.362 |
| 100K | Knowledge Update | 0.150 | 0.362 |
| | Multi-Hop Reasoning | 0.095 | 0.281 |
| | Preference Following | 0.425 | 0.566 |
| | Summarization | 0.045 | 0.232 |
| | Temporal Reasoning | 0.000 | 0.112 |
| | Average | 0.206 | **0.311** |
| | Abstention | 0.928 | 0.571 |
| | Contradiction Resolution | 0.007 | 0.017 |
| | Event Ordering | 0.237 | 0.244 |
| | Information Extraction | 0.047 | 0.506 |
| | Instruction Following | 0.166 | 0.295 |
| 500K | Knowledge Update | 0.014 | 0.278 |
| | Multi-Hop Reasoning | 0.022 | 0.214 |
| | Preference Following | 0.386 | 0.571 |
| | Summarization | 0.069 | 0.344 |
| | Temporal Reasoning | 0.028 | 0.121 |
| | Average | 0.191 | **0.316** |
| | Abstention | 0.792 | 0.500 |
| | Contradiction Resolution | 0.003 | 0.021 |
| | Event Ordering | 0.211 | 0.200 |
| | Information Extraction | 0.106 | 0.366 |
| | Instruction Following | 0.166 | 0.419 |
| 1M | Knowledge Update | 0.014 | 0.357 |
| | Multi-Hop Reasoning | 0.105 | 0.209 |
| | Preference Following | 0.391 | 0.551 |
| | Summarization | 0.041 | 0.316 |
| | Temporal Reasoning | 0.033 | 0.154 |
| | Average | 0.186 | **0.309** |
| | Abstention | 0.750 | 0.550 |
| | Contradiction Resolution | 0.000 | 0.012 |
| | Event Ordering | 0.205 | 0.197 |
| | Information Extraction | 0.000 | 0.350 |
| | Instruction Following | 0.300 | 0.350 |
| 10M | Knowledge Update | 0.000 | 0.275 |
| | Multi-Hop Reasoning | 0.000 | 0.125 |
| | Preference Following | 0.166 | 0.308 |
| | Summarization | 0.061 | 0.220 |
| | Temporal Reasoning | 0.000 | 0.000 |
| | Average | 0.148 | **0.238** |

## D NUGGET DESIGN

In this section, we provide illustrative examples for each memory ability, demonstrating how nuggets are derived from the corresponding probing questions.

1. **Abstention**

   **Objective:** The correct behavior is to acknowledge that the requested information is not present in the provided conversation.
   **Rubric pattern:** Each atomic unit should be in this format: *States that, based on the provided chat, there is no information about* `<target topic>`
   **Example JSON:**

   ```
   {
       "question": "What specific advice did Manuel give about property management
           companies during the March 5 Investors Meetup?",
       "ideal_response": "Based on the provided chat, there is no information related
           to the specific advice Manuel gave about property management companies.",
       "source_chat_ids": {},
       "rubric": [
           "Based on the provided chat, there is no information related to the specific
               advice Manuel gave about property management companies."
       ]
   }
   ```

2. **Contradiction Resolution**

   **Objective:** Correct behavior is that the LLM should detect the contradiction and state both contradictory information while requesting clarification.
   **Rubric pattern:**
   - States there is contradictory information.
   - Mentions claim `<A>`
   - Mentions claim ``
   - Requests clarification about which statement is correct
   **Example JSON:**

   ```
   {
       "question": "Have I ever attended any real estate webinars or investor meetups?"
           ,
       "ideal_answer": "I notice you've mentioned contradictory information about this.
           You said you have never attended any real estate webinars or investor
           meetups, but you also mentioned attending a webinar about Turkey's rising
           demand for multi-family rentals. Which statement is correct?",
       "source_chat_ids": {
           "first_statement": [
           ],
           "second_statement": [
           ]
       },
       "rubric": [
           "LLM response should state: there is contradictory information",
           "LLM response should mention: You said you have never attended any real
               estate webinars or investor meetups",
           "LLM response should mention: you also mentioned attending a webinar about
               Turkey\u2019s rising demand for multi-family rentals",
           "LLM response should mention: which statement is correct?"
       ]
   }
   ```

3. **Event Ordering**

   **Objective:** Correct behavior is the model lists a sequence of events/topics in the correct chronological order.
   **Rubric pattern:**
   - LLM response should mention: `<event 1>`
   - . . .

> • LLM response should mention: `<event N>`
> **Example JSON:**
>
> ```
> {
>     "question": "How did my focus on different aspects of property investment and
>         management develop throughout our conversations in order? Mention ONLY and
>         ONLY ten items.",
>     "answer": "Your focus on property investment and management developed in this
>         sequence: 1) Initial engagement with the local agent and preparation for
>         property viewings, 2) Evaluation of property financials including ROI and
>         rental income potential, 3) Exploration of financing options and mortgage
>         concerns, 4) Handling contractor performance and repair negotiations, 5)
>         Understanding zoning regulations and permit requirements for property
>         conversions, 6) Planning and prioritizing renovations and investment risks
>         for multi-family properties, 7) Detailed preparation for property viewings
>         involving both agent and contractor, 8) Negotiation strategies and cash
>         flow implications related to repair costs, 9) Final renovation project
>         completion steps and portfolio diversification strategies, 10)
>         Consideration of commercial property types and location factors for long-
>         term investment.",
>     "ordering_tested": [
>         "1st: Agent interaction and viewing preparation",
>         "2nd: Property financial evaluation",
>         "3rd: Financing and mortgage concerns",
>         "4th: Contractor management",
>         "5th: Zoning and permits",
>         "6th: Renovation planning and investment risks",
>         "7th: Viewing preparation with agent and contractor",
>         "8th: Repair cost negotiation and cash flow",
>         "9th: Renovation completion and portfolio diversification",
>         "10th: Commercial property and location considerations"
>     ],
>     "source_chat_ids": [],
>     "rubric": [
>         "LLM response should mention: Agent interaction and viewing preparation",
>         "LLM response should mention: Property financial evaluation",
>         "LLM response should mention: Financing and mortgage concerns",
>         "LLM response should mention: Contractor management",
>         "LLM response should mention: Zoning and permits",
>         "LLM response should mention: Renovation planning and investment risks",
>         "LLM response should mention: Viewing preparation with agent and contractor"
>             ,
>         "LLM response should mention: Repair cost negotiation and cash flow",
>         "LLM response should mention: Renovation completion and portfolio
>             diversification",
>         "LLM response should mention: Commercial property and location
>             considerations",
>         "Presents the events in the correct chronological order"
>     ]
> }
> ```

4. **Information Extraction**

> **Objective:** LLM should answer the questioned facts correctly.
> **Rubric pattern:**
> • Instantiate one criterion per fact directly from the ideal answer, using the stem "LLM response should state/mention:"
> **Example JSON:**
>
> ```
> {
>     "question": "What amount did I say I had set aside as my initial capital for
>         investing?",
>     "ideal_answer": "You said you had set aside $50,000 as your initial capital.",
>     "source_chat_ids": [],
>     "rubric": [
>         "LLM response should state: $50,000"
>     ]
> }
> ```

5. **Instruction Following**

**Objective:** LLM should adhere to format and/or content priorities stated in the conversation.
**Rubric pattern:**
- Use `instruction_being_tested` (the explicit instruction) and decompose `expected_compliance` into atomic criteria

**Example JSON:**

```
{
    "question": "How should I allocate my funds for the upcoming quarter?",
    "instruction_being_tested": "Always provide detailed financial breakdowns when I
        ask about budgeting decisions.",
    "expected_compliance": "Response should include itemized costs, specific amounts
        for different categories, and detailed breakdown rather than just a total
        estimate",
    "source_chat_ids": [],
    "rubric": [
        "LLM response should contain: itemized list of costs",
        "LLM response should contain: category-by-category breakdown",
        "LLM response should contain: detailed cost analysis"
    ]
}
```

6. **Knowledge Update**

**Objective:** LLM must reflect updated values when prior values have changed over time.
**Rubric pattern:**
- Derive criteria from the ideal answer, using the stem "LLM response should state/mention:" for the updated value(s).

**Example JSON:**

```
{
    "question": "How much initial capital have I allocated for real estate investing
        ?",
    "answer": "$60,000",
    "source_chat_ids": {
        "original_info": [

        ],
        "updated_info": [
        ]
    },
    "rubric": [
        "LLM response should state: $60,000"
    ]
}
```

7. **Multi-hop Reasoning**

**Objective:** LLM must aggregate or compare information spanning multiple sessions.
**Rubric pattern:**
- Instantiate criteria from the ideal answer for each required intermediate or aggregated fact.

**Example JSON:**

```
{
    "question": "How many different banks did I consider for my mortgage options
        across my conversations?",
    "answer": "Two banks: Halkbank and Ziraat Bank.",
    "source_chat_ids": [],
    "rubric": [
        "LLM response should state: Two banks",
        "LLM response should state: Halkbank",
        "LLM response should state: Ziraat Bank"
    ]
}
```

8. **Preference Following**

**Objective:** LLM must generate content consistent with user-specified preferences.
**Rubric pattern:**
- Use `preference_being_tested` (the user's stated preference) and decompose `expected_compliance` into atomic criteria.

**Example JSON:**

```
{
    "question": "I'm looking at several properties and want to understand how to
        approach comparing them effectively. What steps would you suggest I take?",
    "preference_being_tested": "I prefer methodical, data-driven decisions over
        impulsive purchases, reflecting my analytical nature.",
    "expected_compliance": "Response should outline a structured, step-by-step
        approach that involves gathering and analyzing relevant data before making
        a choice, rather than suggesting quick or impulsive actions.",
    "source_chat_ids": [],
    "rubric": [
        "LLM response should contain: provides a clear, logical process",
        "LLM response should contain: emphasizes data collection and analysis",
        "LLM response should contain: breaks down comparison into measurable factors
            "
    ]
}
```

9. **Summarization**

**Objective:** LLM must provide a comprehensive summary covering required content elements.
**Rubric pattern:**
- Decompose `ideal_summary` into atomic content units; use the stem "LLM response should contain:".

**Example JSON:**

```
{
    "question": "Can you give me a comprehensive summary of my journey and decision-
        making process around investing in rental properties, including how my
        budget, property choices, management considerations, and financing plans
        have developed over time?",
    "ideal_summary": "Your journey toward investing in rental properties began with
        an initial capital of $50,000, which you questioned as potentially
        insufficient for purchasing a property within 12 months. Early discussions
        highlighted the need to research local market conditions, down payment
        requirements, and additional costs like closing fees and renovations,
        revealing that typical investments might exceed your initial capital. You
        explored identifying good fixer-upper properties by learning to recognize
        signs such as structural issues and outdated features, emphasizing the
        importance of cost-benefit analysis for renovations. As your plans
        progressed, you weighed the pros and cons of investing close to your
        location versus elsewhere, balancing ease of management against market
        diversity and growth potential. You also considered the choice between
        single-family homes and multi-family units, analyzing factors like rental
        yield, management complexity, and investment scale, with examples showing
        similar yields but differing capital needs. Financing options were
        carefully compared, particularly between Halkbank and Ziraat Bank mortgages
        , focusing on interest rates, fees, and service quality to optimize costs.
        Throughout, you developed a step-by-step plan for purchasing your first
        rental property, including market research, budgeting, inspections,
        financing, and tenant management, with timelines to reduce anxiety and
        ensure readiness. This comprehensive process reflects a thoughtful
        evolution from initial capital concerns to detailed investment strategies,
        property evaluation, financing decisions, and management planning, all
        aimed at making informed, balanced real estate investment choices.",
    "source_chat_ids": [],
    "rubric": [
        "LLM response should contain: investing in rental properties began with an
            initial capital of $50,000",
        "LLM response should contain: Early discussions highlighted the need to
            research local market conditions, down payment requirements, and
            additional costs like closing fees",
        "LLM response should contain: You explored identifying good fixer-upper
            properties by learning to recognize signs such as structural issues and
             outdated features",
```

```
        "LLM response should contain: you weighed the pros and cons of investing
            close to your location versus elsewhere, balancing ease of management
            against market diversity and growth potential",
        "LLM response should contain: You also considered the choice between single-
            family homes and multi-family units, analyzing factors like rental
            yield, management complexity, and investment scale",
        "LLM response should contain: Financing options were carefully compared,
            particularly between Halkbank and Ziraat Bank mortgages, focusing on
            interest rates, fees, and service quality to optimize costs",
        "LLM response should contain: you developed a step-by-step plan for
            purchasing your first rental property, including market research,
            budgeting, inspections, financing, and tenant management"
    ]
}
```

10. **Temporal Reasoning**

**Objective:** LLM must compute or restate durations and timeline relations correctly.
**Rubric pattern:**
   - Derive criteria from the ideal answer, using the stem "LLM response should state:".
**Example JSON:**

```
{
    "question": "How many days are there between my first property viewing with
        Mehmet Yilmaz and the last one I scheduled?",
    "answer": "There are 2 days between the first property viewing on March 25 and
        the last one on March 27.",
    "calculation_required": "March 27 - March 25 = 2 days",
    "source_chat_ids": {
        "first_event": [],
        "second_event": []
    },
    "rubric": [
        "LLM response should state: 2 days",
        "LLM response should state: from March 25, 2024 till March 27, 2024"
    ]
}
```

# E    EXAMPLES FROM DIFFERENT COMPONENTS OF BEAM

In this section, we provide illustrative examples of generating a chat in the *coding* domain. Specifically, we include a representative *chat seed* with its domain, title, theme, and subtopics, followed by the corresponding *narratives*, where only a truncated set is shown for brevity. We then present the *user profile* and the user's social *relationships*. Next, we provide excerpts from the *conversation plans*, showing only a subset of bullet points from each sub-plan while preserving their full descriptions to maintain clarity. Finally, we provide samples of the *generated chat*, highlighting exchanges where the user shares or requests code, and including follow-up turns to demonstrate the naturalistic back-and-forth flow. Together, these examples illustrate how different components of BEAM interact to form coherent, long-context dialogues.

---

**Chat Seed**

**Domain:** Coding
**Title:** Automating Social Media Posts with Python
**Theme:** Scheduling and posting content across multiple platforms
**Subtopics:**

- Twitter API integration    ·    Facebook Graph API usage    ·    Instagram automation tools
- Scheduling with cron jobs / APScheduler
- Image and caption management; hashtag generation
- Error handling for failed posts; tracking engagement metrics

---

**Narratives (Truncated)**

**Technical Problem-Solving:** Debugging Twitter OAuth/403/429; fixing hashtag validation; profiling scheduler bottlenecks.
**Learning & Knowledge:** API docs comprehension (Twitter v2, Facebook Graph v12–15); best practices for Instagram automation; mastering cron/APScheduler.
**Progress & Development:** Setting up Twitter/Facebook integrations; building Instagram tools; designing scheduling algorithms.
**Implementation:** Feature implementation and refactoring for efficiency; async migration; retry and backoff strategies.
**Framework & Technology:** Python libraries (Tweepy, facebook-sdk, requests); APScheduler/cron; Redis; asyncio.
**Testing & QA:** Unit/integration/E2E tests (pytest, Selenium); TDD for schedulers and hashtag rules.
**DevOps & Deployment:** CI/CD (GitHub Actions), containerization (Docker), EC2 deployment, blue–green releases.
**Data:** PostgreSQL schemas, indices, ETL for engagement metrics, Redis caching.
**Integration & APIs:** Webhooks, message queues (RabbitMQ), API Gateway, SNS/Lambda.
**Performance:** Caching, load balancing (HAProxy), CPU/memory targets, throughput goals.
**Security/Compliance:** OAuth, token rotation, TLS, GDPR.
**PM & Workflow:** Sprints, reviews, documentation standards.

---

**User Profile**

**Name:** John Brooks    **Age:** 52    **Gender:** Male
**Location:** Port Charles, Luxembourg    **Profession:** Secretary/Administrator

**Personality:** He is a pillar of his community, always ready to lend a helping hand and offer guidance when needed. With a strong sense of tradition and order, he values honesty and dedication, often taking on a mentorship role to help others. His diligent and efficient approach to planning and organization makes him a reliable asset to those around him. He has a warm and welcoming demeanor, always willing to open his heart and home to friends, loved ones, and neighbors. Despite his strong convictions, he believes in the power of hospitality and good manners, often going out of his way to make others feel supported and cared for.

With a dry sense of humor and a quick wit, he can be entertaining to be around, but he's not afraid to speak his mind and challenge the status quo when necessary. His practical and responsible nature makes him a respected member of his community, and his ability to stay grounded and logical in stressful situations is a valuable asset to those around him.

### Relationships

**Parents:** Elizabeth (74), Robert (76)
**Partner:** Shannon (48)
**Close Friends:** Taylor (51), Teresa (62), Thomas (44), Charles (56), Patricia (46)
**Acquaintances/Colleagues:** Wesley (26), Jason (59), Claudia (15), Janice (13), Dana (55)

### Conversation Plan (Only a few representative bullets from each sub-plan)

**Subplan 1 — March 1, 2024**
- **Project Initialization:** I'm setting up a Python 3.10 environment with Tweepy v4.10.1 and Facebook SDK v3.1.0 for API integrations.
- **Security & Compliance Labels: Authentication for Twitter API Integration:** Implemented OAuth 1.0a with environment variables `TWITTER_API_KEY` and `TWITTER_API_SECRET` securely stored.
- **Database & Data Management Labels: Database Design for Social Media Posting:** Designed PostgreSQL 14 schema with tables for posts, platforms, and scheduling metadata.
- **User Instruction:** Always include exact API version numbers when I ask about integration details.
- **Logical Contradiction:** I have never registered a Twitter Developer account or created any Twitter app.

**Subplan 2 — March 20, 2024**
- **Technical Problem-Solving Labels: Debugging Twitter API Integration:** Fixed "403 Forbidden" error caused by missing media upload step before tweet creation.
- **Implementation & Development Labels: Code Refactoring for Performance:** Refactored `twitter_post.py` to async functions using `asyncio`, improved throughput by 30%.
- **Security & Compliance Labels: Authorization for Facebook Graph API:** Implemented OAuth 2.0 flow with refresh tokens stored encrypted using Fernet symmetric encryption.
- **Information Update:** The Instagram automation prototype sprint deadline was adjusted to April 5, 2024, to allow additional testing of media upload features.

**Subplan 3 — April 5, 2024**
- **Implementation & Development Labels: Implementing Error Handling:** Added retry logic with exponential backoff for Instagram API 429 Too Many Requests errors.
- **Performance & Optimization Labels: Caching Strategies for Image and Caption Management:** Implemented Redis caching for resized images, reducing image processing time from 800ms to 200ms.
- **Debugging & Troubleshooting Labels: Incident Response for Social Media Automation:** Responded to March 30, 2024, outage caused by expired Instagram tokens, implemented alerting via Slack webhook.

**Subplan 4 — April 20, 2024**
- **Implementation & Development Labels: Algorithm Optimization for Scheduling:** Rewrote scheduling algorithm to use async priority queues, reducing average job dispatch latency from 500ms to 150ms.

- **Framework & Technology Labels: Integrating Twitter API with Python:** Upgraded Tweepy from v4.10.1 to v4.12.1 to leverage new media upload endpoints.

- **Security & Compliance Labels: Authentication for Twitter API Integration:** Rotated Twitter API keys on April 15, 2024, updated environment variables `TWITTER_API_KEY` and `TWITTER_API_SECRET`.

**Subplan 5 — May 5, 2024**

- **Progress & Development Labels: Building Hashtag Generation Tools:** Developed hashtag generator supporting dynamic keyword extraction using spaCy v3.5.0 NLP library.

- **Database & Data Management Labels: Data Warehousing for Engagement Metrics:** Designed PostgreSQL 14 schema for engagement_metrics with partitioning by month for scalability.

- **Debugging & Troubleshooting Labels: Log Analysis for Facebook Graph API:** Detected "OAuthException: Error validating access token" on May 1, 2024, resolved by token refresh automation.

**Subplan 6 — May 20, 2024**

- **Implementation & Development Labels: Implementing Error Handling:** Added centralized error handler middleware in posting API, logging errors with Sentry v1.12.0.

- **Debugging & Troubleshooting Labels: Error Diagnosis for Twitter API Integration:** Fixed intermittent "ConnectionResetError" during media upload by adding retry with jitter.

- **DevOps & Deployment Labels: Containerization for Instagram Automation:** Updated Dockerfile to use multi-stage builds, reduced image size from 120MB to 85MB.

**Subplan 7 — June 5, 2024**

- **DevOps & Deployment Labels: Deploying Social Media Automation Tools:** Deployed v1.0.0 release on AWS EC2 t3.medium with 99.9% uptime SLA.

- **Integration & API Labels: Event-Driven Architecture for Social Media Automation:** Implemented AWS SNS topics for post status updates, integrated with Lambda v3.2.1 functions.

- **User Experience & Interface Labels: Mobile App Design for Social Media Automation:** Released beta version of React Native app on Android with basic scheduling and metrics display.

**Subplan 8 — June 20, 2024**

- **Progress & Development Labels: Developing Instagram Automation Tools:** Implemented batch media uploads for Instagram, supporting up to 10 images per carousel post.

- **User Experience & Interface Labels: Responsive Design for Scheduling:** Enhanced React 18.2 dashboard for scheduling with drag-and-drop post reordering, tested on Chrome and Safari.

- **Security & Compliance Labels: Encryption for Instagram Automation:** Migrated token storage to AWS KMS encrypted S3 bucket, access controlled via IAM roles.

**Subplan 9 — July 5, 2024**

- **Security & Compliance Labels: Authentication for Twitter API Integration:** Implemented OAuth 2.0 PKCE flow for enhanced security, tested with Postman v10.15.0.

- **Performance & Optimization Labels: Code Optimization for Twitter API Integration:** Reduced memory usage by 20% by optimizing data serialization and connection pooling.

- **Debugging & Troubleshooting Labels: Error Diagnosis for Twitter API Integration:** Fixed "TimeoutError" during high load by increasing HTTP client timeout from 10s to 30s.

**Subplan 10 — July 20, 2024**

- **DevOps & Deployment Labels: Deploying Social Media Automation Tools:** Completed production rollout on July 18, 2024, with zero downtime and 99.95% uptime in first 48 hours.

- **Integration & API Labels: Event-Driven Architecture for Social Media Automation:** Fully implemented AWS SNS event bus with Lambda v3.2.1 for asynchronous post status updates.

- **Testing & Quality Assurance Labels: End-to-End Testing for Instagram Automation:** Verified Instagram automation stability with 1000+ posts over 7 days, no failures reported.

### Generated Dialogues (Exemplars, Truncated)

Turn 1 (Mar 1, 2024)

**User:** I'm trying to set up a Python 3.10 environment for automating social media posts, and I've chosen Tweepy v4.10.1 and Facebook SDK v3.1.0 for API integrations, but I'm not sure how to structure my project for better outreach. Can you help me plan this out in terms of scheduling and deadlines?

**Assistant:**
Certainly! Setting up an automated posting system with Tweepy and Facebook SDK is a great start.
Project Structure:
1. Project Setup (Mar 1–7) [Truncated ...]
2. Content Creation (Mar 8–14) [Truncated ...]
3. Automation Development (Mar 15–21) [Truncated ...]
4. Deployment and Monitoring (Mar 22–28) [Truncated ...]

**User (Follow-up)**
That sounds like a great plan! One thing though—could we add a step for testing posts on a smaller scale before full deployment? Just to make sure everything looks good and functions as expected.

**Assistant**
Absolutely, adding a step for testing the posts on a smaller scale is a great idea to ensure everything looks good and functions as expected before full deployment. Here's how you can incorporate that into the existing plan:
### Updated Project Structure
#### 1. **Project Setup [Trunicated ...]
#### 4. **Testing Phase (March 22 - March 24, 2024)**
- **Days 22-23:** Conduct a small-scale test.
- Post a few test messages to a limited audience or a test account.
- Verify that the posts look good and function as expected.
- **Day 24:** Review the test results.
- Address any issues found during the test phase.
- Make final adjustments to the automation script and content.

[Trunicated ...]

---

**User (Mid-turn) –> Code Sharing**
I'm trying to implement a feature to collect engagement metrics for Instagram posts, but I'm not sure how to integrate it with the existing PostgreSQL schema. I've designed a table for instagram_posts with foreign key to posts table, but I'm not sure how to handle the data migration. Can you help me with that? Here's my code:

```
import psycopg2
conn = psycopg2.connect(dbname=\"engagement_metrics\",    user=\"username\",
```

```
password=\"password\",
host=\"localhost\",
port=\"5432\")
cur = conn.cursor()
cur.execute(\"\"\"
CREATE TABLE instagram_posts (id SERIAL PRIMARY KEY,        post_id INTEGER REFERENCES
    posts(id),        engagement_metrics JSONB);
\"\"\")\n\nconn.commit()
cur.close()
conn.close()
```

### Assistant –> Code Review
Your approach is good! Consider adding `NOT NULL`, timestamps, and indexes for performance . . .

```
CREATE TABLE IF NOT EXISTS instagram_posts (
  id SERIAL PRIMARY KEY,
  post_id INTEGER NOT NULL REFERENCES posts(id),
  engagement_metrics JSONB,
  created_at TIMESTAMP NOT NULL DEFAULT CURRENT_TIMESTAMP
);
CREATE INDEX IF NOT EXISTS idx_instagram_posts_post_id
  ON instagram_posts(post_id);
```

[Truncated ...] plus safe migration with INSERT SELECT to backfill existing posts.

---

### User –> Deployment
I'm trying to deploy my updated posting services to an AWS EC2 t3.medium instance with 2 vCPUs and 4GB RAM, but I'm running into some issues with the deployment process. Can you help me figure out how to properly deploy my application to this instance, considering the DevOps and deployment aspects? I've been using Docker to containerize my application, and I'm not sure what changes I need to make to ensure a smooth deployment. Here's a snippet of my current Dockerfile:

```
FROM python:3.10-slim
WORKDIR /app
COPY requirements.txt .
RUN pip install -r requirements.txt
COPY . .
CMD ["python", "app.py"]
```

### Assistant –> Deployment Guidance
Install Docker on EC2, push image to registry, pull with pinned tag, and run with `--restart=always` [Truncated ...] add healthchecks and a minimal `systemd` unit [Truncated ...]

### User (Follow-up)
hmm, what's the best way to automate the pulling and running of the Docker container?

### Assistant –> CI/CD Automation
Use GitHub Actions with SSH deploy step [Truncated ...]

```
name: Deploy
on: { push: { branches: [main] } }
jobs:
  deploy:
    runs-on: ubuntu-latest
    steps:
      - uses: actions/checkout@v4
      - run: |
          ssh -i id_rsa ec2-user@EC2_DNS \
          "docker stop socauto || true && docker rm socauto || true &&
           docker pull username/your-app:sha-${GITHUB_SHA} &&
           docker run -d --restart=always -p 80:80 username/your-app:sha-${GITHUB_SHA}"
```

### User (Follow-up)
hmm, what's the best way to handle environment variables for different stages like dev and prod?

> **Assistant –> Env Vars**
> Use separate `.env.dev` / `.env.prod` files locally. In production, prefer AWS SSM Parameter St46ore or Secrets Manager [Truncated ...] inject via `--env-file` or `systemd` environment exports.

## F CASE STUDY

The scratchpad serves as a persistent, iteratively compressed semantic layer that (i) consolidates scattered entities and facts (improving *Information Extraction*), (ii) preserves user-level meta-instructions alongside task context (improving *Instruction Following*), (iii) records state changes and overrides (improving *Information Update*), and (iv) normalizes timeline cues into explicit anchors (improving *Temporal Reasoning*). Out of the ten memory abilities in our benchmark, we present case studies from these four; however, the scratchpad provides similar benefits for the remaining six abilities as well. Below we contrast "LIGHT with all components" versus "LIGHT w/o scratchpad" on representative probes, and show the exact scratchpad cue that enabled the correct answer.

---

**Information Extraction**

**(1) Question:** Which versions of the tools did I say I was using to start my project?
**Correct Answer:** Python 3.10, spaCy v3.5, Flask 2.2, PyMuPDF 1.22.0
**LIGHT with all components:** Python 3.10, spaCy v3.5, Flask 2.2, PyMuPDF 1.22.0
**LIGHT wo scratchpad:** Based solely on the provided context, there is no direct answer to this question
**Scratchpad cue (excerpt):**
\*\*KEY ENTITIES & RELATIONSHIPS:\*\*
...... [trunecated]
- Technologies: Python 3.10, spaCy v3.5, Flask 2.2/2.2.3, PyMuPDF 1.22.0, scikit-learn v1.2.2, Redis, RabbitMQ, Docker
...... [trunecated]

---

**(2) Question:** Can you remind me of the city and country I said I live in, along with my age?
**Correct Answer:** New Gary, Saint Helena; 26
**LIGHT with all components:** New Gary, Saint Helena; 26
**LIGHT wo scratchpad:** No information provided about your city, country, or age in the given context
**Scratchpad cue (excerpt):**
\*\*KEY ENTITIES & RELATIONSHIPS:\*\*
- \*\*Primary individuals:\*\* User and partner Wendy (both turning 26 on March 3, 2024)
- \*\*Friends and collaborators:\*\* Joe (writer, 35), Stacy ...... [trunecated]
- \*\*Locations:\*\* New Gary, Saint Helena; Island Art Gallery; Saint Helena Book Fair; Saint Helena Community Center; GreenLeaf Nursery; Island Finds thrift store; Saint Helena Charity; The Lighthouse Bistro

---

**Instruction Following**

**Question:** Can you tell me about the mentoring activities I've been involved in recently?
**User instruction:** Always provide the number of team members involved when I ask about mentoring activities.
**Correct Answer:** Mentions number of team members
**LIGHT with all components:** Mentoring 3 junior editors
**LIGHT wo scratchpad:** No recent mentoring activities noted in the provided context
**Scratchpad cue (excerpt):**
...... [trunecated]
The user requested to always include the number of team members when discussing mentoring activities.
...... [trunecated]
\*Mentoring & Crisis Management:\*

- Mentoring 3 junior editors; balancing mentoring with crisis management.
...... [trunecated]

---

### Knowledge Update

**(1) Question:** What is the typical memory usage per session for batch resume processing?
**Correct Answer:** Around 140MB per session
**LIGHT with all components:** ˜140MB
**LIGHT wo scratchpad:** The typical memory usage per session for batch resume processing can vary widely depending on the complexity of the resumes and the system's available memory, but optimizing with batch sizes and efficient data handling can keep it under 100MB per resume.
**Scratchpad cue (excerpt):**
...... [trunecated]
- Memory optimization:
- Reduce peak memory usage from 250MB to 140MB per batch.
...... [trunecated]

---

**(2) Question:** What is the deadline for completing the Instagram automation prototype sprint?
**Correct Answer:** April 5, 2024
**LIGHT with all components:** April 5, 2024
**LIGHT wo scratchpad:** April 1, 2024
**Scratchpad cue (excerpt):**
**PROCESSES & WORKFLOWS:**
1. **Project Planning & Sprint Management:**
...... [trunecated]
- Instagram automation sprint extended to April 5, 2024.

---

### Temporal Reasoning

**(1) Question:** How many days passed between when I met my mom at home and when I completed my first successful batch of Miche's sourdough?
**Correct Answer:** Jan 6 – Jan 1 = 5 days
**LIGHT with all components:** 5 days
**LIGHT wo scratchpad:** 37 days
**Scratchpad cue (excerpt):**
**IMPORTANT DATES:**
- **January 1, 2024:** Met mother Brittney at home.
- **January 5-6, 2024:** Completed first batch of Michele's sourdough.
...... [trunecated]

---

**(2) Question:** How many days do I have between finalizing my survey plans at Café Soleil and attending the Raymondburgh Startup Meetup to prepare effectively?
**Correct Answer:** Mar 28 – Mar 10 = 18 days
**LIGHT with all components:** 18 days
**LIGHT wo scratchpad:** 28 days
**Scratchpad cue (excerpt):**
**IMPORTANT DATES:**
- **March 10, 2024**: Paper-based customer survey at Cafe Soleil.
- **March 28, 2024**: Raymondburgh Startup Meetup.
...... [trunecated]

**Takeaways.** Across abilities, removing the scratchpad consistently causes failures that the full model avoids. In *Information Extraction*, the scratchpad aggregates dispersed entity/version mentions so the model can recover exact tool versions and bios (city/age). For *Instruction Following*, it retains user meta-preferences (e.g., "always include team count"), ensuring style/format compliance even many turns later. For *Knowledge Update*, it encodes overrides (e.g., extended deadline; reduced memory), preventing stale answers. For *Temporal Reasoning*, it surfaces normalized date anchors, enabling simple, correct day-difference calculations. These examples show that the scratchpad provides a high-utility semantic scaffold that complements working (recency) and episodic (retrieval) memory, yielding robust long-context behavior.

# G  QUALITATIVE ERROR ANALYSIS

We conduct a qualitative analysis of failure cases across the ten memory abilities in our benchmark to better characterize the limitations of LIGHT and identify systematic patterns. For each ability, we manually inspected probing questions that LIGHT answered incorrectly and analyzed the underlying reasons. Below, we summarize the dominant error modes observed for each ability.

**Abstention**   In this ability, the LLM should abstain from answering because the answer to the probing question is not present in the conversation. Therefore, the context that LIGHT provides to the LLM does not contain the required information. One failure mode occurs when the context contains nothing relevant to the question, yet the LLM hallucinates and generates an answer. This is because these LLMs are usually trained to always provide an answer, regardless of actually having this knowledge (Kalai et al., 2025). Another hallucination pattern occurs when the LLM produces an answer entirely unrelated to the question, which stems from the long-context nature of the task and the inability of the LLM to understand the context correctly. The main failure mode, however, arises when the context contains information about entities, dates, or concepts that are similar to, but not the same as, the information requested by the question. In these cases, the LLM uses these similar details and generates an answer instead of abstaining. This pattern is the primary failure mode for abstention.

**Contradiction Resolution**   For this ability, the LLM should identify the contradiction, state both sides, and request clarification. One common failure mode occurs when the context contains only one side of the contradiction, leading the LLM to answer based solely on that information. Since the model does not have access to the other side of the story, it cannot detect the conflict. Another common failure occurs when both sides of the contradiction are present in the context, but the LLM still overweighs one side of the contradiction due to position and frequency bias for that side in the context.

**Event Ordering**   In this ability, the LLM should recognize and reconstruct the sequence of evolving information in the conversation. A common failure occurs when the context contains items from the sequence but the LLM does not include them in the response. Another failure mode occurs when the model includes the items but presents them in the wrong order. This happens because the retrieval model retrieves based on similarity, which does not necessarily preserve temporal order, leaving the LLM without clues about the correct sequence of events. Also, in many cases, the retriever does not retrieve all the events related to the question.

**Information Extraction**   In this ability, one failure mode arises when the context does not contain the answer to the question; thus, the LLM cannot extract the answer from the retrieved context. Another occurs when the answer is present, but the LLM produces an incorrect answer because it becomes confused by details in the context that are similar to the answer. A third failure happens when the answer is present but the LLM provides an incomplete answer.

**Instruction Following**   For this ability, the LLM should adhere to user-specified instructions. Failures occur either when the user instruction is present in the context, but the LLM does not follow it, or when the instruction is missing from the context, and the LLM cannot answer the question without it.

**Knowledge Update**   Here, the LLM should answer the question using the updated version of the facts. One failure mode occurs when the context contains only the old value and does not include the updated value, causing the LLM to respond using outdated information, because the retrieval model retrieves based on similarity, which does not necessarily preserve temporal order. A more common failure mode occurs when the context contains both the old and updated values, but the LLM still bases its answer on the old value, because the retrieved documents are not necessarily presented in the correct temporal order, again due to the retriever.

**Multi-Hop Reasoning**   For this ability, the LLM often fails when the context contains the necessary pieces of information but the model does not use them to answer the question. Another failure

occurs when the context is missing some components required for the multi-hop reasoning chain, making it impossible for the LLM to answer correctly.

**Preference Following**   In this ability, the LLM should incorporate user-stated preferences into its answer. Failures occur when the context does not contain the user's preference and the model therefore answers without considering it, or when the preference is present but the LLM does not use it when generating the response.

**Summarization**   For this ability, one failure mode occurs when the context contains some components of the correct answer but the LLM fails to include them in the summary. Another failure arises when the context is missing some parts of the answer, which leads the LLM to omit those details as well.

**Temporal Reasoning**   In this ability, the LLM should reason about explicit and implicit temporal relations. A common failure mode occurs when the context contains the required dates, but the LLM becomes confused by another date (or dates) in the context and answers incorrectly. Another failure occurs when the context contains both dates but the LLM performs the arithmetic incorrectly, producing an answer with a numerical error. A third failure mode occurs when the context does not contain one of the necessary dates, causing the LLM to incorrectly substitute another date when answering.

## H PROMPTS

Here we provide the prompts used in different stages of our framework.

```
This is a plan that contains detailed bullet points about a topic. This plan is used to generate realistic
      chat conversations between a user and an AI assistant, which are then used to evaluate the long-term
      memory capabilities of LLMs.
Your task is to analyze this plan and select bullet points that would be most effective for testing
      information extraction abilities when incorporated into chat conversations.
Analyze this plan and identify bullet points that contain specific factual information ideal for testing
      precise recall and information extraction capabilities.

## INPUT DATA
- **PLAN**: <plan>

## CRITICAL REQUIREMENT: EARLY BATCH PRIORITIZATION
**SELECTION PRIORITY ORDER:**
1. **Batch 1-3 (HIGHEST PRIORITY)**: Select 70-80% of your choices from these early batches
2. **Batch 4-6 (MEDIUM PRIORITY)**: Select 10-20% of your choices from these middle batches
3. **Batch 7+ (LOW PRIORITY)**: Select only 5-10% of your choices from later batches

Focus on bullet points with:
- **Specific numbers, quantities, measurements, prices, percentages**
- **Proper names of people, organizations, brands, locations**
- **Exact dates, times, schedules, or deadlines**
- **Contact details such as addresses, phone numbers, email IDs**
- **Technical or detailed descriptions (model names, product codes, ratings, specifications)**
- **Distinctive events, awards, or milestones**
- **Direct quotes, messages, or instructions with exact wording**
- **Precise parameters, formulas, or datasets in technical and academic contexts**
- **Mathematical expressions, theorems, proofs**

Prioritize information that:
- Appears early in the timeline
- Contains multiple distinct factual details in one bullet point
- Includes uncommon names, technical terms, or culturally specific references
- Has precise numerical values or measurements that could be easily confused
- Could be misremembered if details are swapped, rounded, or reworded
- Requires high accuracy to preserve meaning (e.g., formulas, addresses, step-by-step processes)

Return your analysis in this exact JSON format:
[{"capability": "information_extraction", "batch_numbers": 1, "bullet_numbers": 3,
      "bullet_points": "• **Personal Introduction:** I am Sherry Rodriguez, 34, licensed conveyancer in
            Hollyborough, Bahrain, earning approximately $68,000 annually."}
]

Important formatting notes:
- The "batch_numbers" and "bullet_numbers" correspond to each other positionally
- "1" and "3" means: Batch 1 Bullet 3

Select ONLY <bullet_number> bullet points total that would generate the highest quality information extraction
      questions.

NOTE: Only output the list without any explanation before or after the list.
```

Listing 1: Candidate selection information extraction prompt

```
This is a plan that contains detailed bullet points about a topic. This plan is used to generate realistic
      chat conversations between a user and an AI assistant, which are then used to evaluate the long-term
      memory capabilities of LLMs.
Your task is to analyze this plan and select GROUPS of related bullet points that would be most effective for
      testing multi-session reasoning abilities when incorporated into chat conversations.
Analyze this project plan and identify GROUPS of bullet points that enable testing of aggregation, comparison,
       and synthesis across multiple batches/sessions. Each group should contain 2-6 related bullet points
      that together enable complex multi-hop reasoning questions.

## INPUT DATA
- **PLAN**: <plan>

## CRITICAL REQUIREMENT: EARLY BATCH PRIORITIZATION
**SELECTION PRIORITY ORDER:**
1. **Groups starting in Batches 1-3 (HIGHEST PRIORITY)**: Select 70-80% of your groups with primary content
      from early batches
2. **Groups spanning early to middle batches (MEDIUM PRIORITY)**: Select 10-20% of groups that bridge early-to
      -middle timeline
3. **Groups from later batches only (LOW PRIORITY)**: Select only 5-10% from purely later batches

Focus on bullet point groups that involve:
- **Aggregation opportunities**: Multiple costs that need to be summed, events that need counting,
      measurements to be totaled
- **Comparison patterns**: Same categories appearing in different batches (budget changes, progress updates,
      relationship interactions)
- **Evolution tracking**: How preferences, decisions, or situations change over time across multiple bullet
      points
- **Cross-reference relationships**: Information that connects between different people, events, or decisions
      across batches

Prioritize bullet point groups that:
- Are part of recurring themes across multiple batches (budget tracking, progress monitoring, relationship
      dynamics)
```

```
- Enable mathematical aggregation across multiple entries (total costs, time durations, quantity counting)
- Allow before/after comparisons of the same entities across different batches
- Require synthesis of information from 3+ different batches
- Create opportunities for complex multi-hop reasoning questions

Return your analysis in this exact JSON format where each object contains multiple related bullet points:
[{"capability": "multi_session_reasoning", "batch_numbers": "1, 2, 3, 5", "bullet_numbers": "10, 4, 7, 7",
        "bullet_points": "Financial & Budget:Cost Estimation: Initial budget set at $12,500, including
                materials and labor for decoupled framing and MLV. | Financial & Budget:Expense Tracking: Paid
                $2,200 deposit to QuietFlow Bahrain for HVAC silencing on March 14 via bank transfer. |
                Financial & Budget:Expense Tracking: Total spent $9,300 by April 1 on materials, labor, and
                consultant fees. | Financial & Budget:Expense Tracking: Total project spending $12,000 as of May
                 1, within original $12,500 budget."}
]

Important formatting notes:
- The "batch_numbers" and "bullet_numbers" correspond to each other positionally
- "1, 2, 3, 5" and "10, 4, 7, 7" means: Batch 1 Bullet 10, Batch 2 Bullet 4, Batch 3 Bullet 7, Batch 5 Bullet
        7
- Use comma-separated values for batch_number and bullet_number
- Separate multiple bullet_point entries with " | "
- Each group should contain 2-6 related bullet points
- Focus on groups that enable the most sophisticated multi-hop aggregation and comparison questions

Select 8-12 groups of bullet points that would enable the most sophisticated multi-session reasoning questions
        .

NOTE: Only output the list without any explanation before or after the list.
```

Listing 2: Candidate selection multi-hop reasoning prompt

```
This is a plan that contains detailed bullet points about a topic. This plan is used to generate realistic
        chat conversations between a user and an AI assistant, which are then used to evaluate the long-term
        memory capabilities of LLMs.
Your task is to analyze this plan and select PAIRS of related bullet points that would be most effective for
        testing knowledge update abilities when incorporated into chat conversations.
Analyze this plan and identify bullet points labeled as "Information Update" and match them with their
        corresponding original facts from earlier in the plan.

## INPUT DATA
- **PLAN**: <plan>

## CRITICAL REQUIREMENT: SPECIAL UPDATE BULLETS WITH ORIGINAL FACTS
Focus on bullet points that:
- **Are labeled "Information Update"**: Look specifically for bullet points with this exact label
- **Have corresponding original facts**: Find the earlier bullet point that contains the original information
        being updated
- **Show clear before/after relationships**: Original information paired with its explicit update or
        correction

For each "Information Update" bullet point you find:
1. **Locate the original fact** in an earlier bullet point that this update refers to
2. **Create a pair** with the original bullet point first, then the "Information Update" bullet point second
3. **Ensure clear connection** between the original fact and its update

Look specifically for "Information Update" bullet points that contain:
- Clear update language ("updated," "changed," "revised," "rescheduled," "increased," "decreased")
- References to modifications of previously mentioned information
- Corrections or adjustments to earlier facts
- Timeline or specification changes
Then match each update with its original fact from earlier bullet points.

Return your analysis in this exact JSON format where each object contains exactly TWO related bullet points (
        original + Information Update):
[{"capability": "knowledge_update", "batch_numbers": "1, 3", "bullet_numbers": "9, 31",
        "bullet_points": "• **Financial & Budget:Cost Estimation:** Initial budget set at $12,500, including
                materials and labor for decoupled framing and MLV. | Information Update: The initial framing
                materials purchase included an additional 10% surplus to accommodate unexpected cuts and errors
                ."}
]

Important formatting notes:
- The "batch_numbers" and "bullet_numbers" correspond to each other positionally
- "1, 3" and "9, 31" means: Batch 1 Bullet 9 (original), Batch 3 Bullet 31 (Information Update)
- Each object must contain exactly 2 bullet points separated by " | "
- Use comma-separated values for batch_number and bullet_number
- First bullet point should represent the original information
- Second bullet point should be the "Information Update" labeled bullet
- Focus on pairs that enable questions like "How did the original plan change when you got the update?"

Select all "Information Update" bullet points and pair them with their corresponding original facts (
        approximately 10 pairs total).

NOTE: Only output the list without any explanation before or after the list.
```

Listing 3: Candidate selection knowledge update prompt

```
This is a plan that contains detailed bullet points about a topic. This plan is used to generate realistic
        chat conversations between a user and an AI assistant, which are then used to evaluate the long-term
        memory capabilities of LLMs.
```

```
Your task is to analyze this plan and select PAIRS of related bullet points that would be most effective for
    testing temporal reasoning abilities when incorporated into chat conversations.
Analyze this project plan and identify PAIRS of bullet points that enable testing duration calculations and
    sequence understanding between two events. Each pair should enable questions about time duration,
    sequence, or temporal relationships between two events.

## INPUT DATA
- **PLAN**: <plan>

## CRITICAL REQUIREMENT: BALANCED BATCH DISTRIBUTION
**SELECTION PRIORITY ORDER:**
1. **Pairs starting in Batches 1-3 (MEDIUM-HIGH PRIORITY)**: Select 40-50% of your pairs with at least one
    bullet from early batches
2. **Far-distance pairs (HIGH PRIORITY)**: Select 30-40% of pairs that span large batch distances (e.g., Batch
    1 & Batch 6, Batch 2 & Batch 8, Batch 1 & Batch 7, etc.) to test long-term temporal reasoning
3. **Pairs spanning early to middle batches (MEDIUM PRIORITY)**: Select 10-15% of pairs that bridge early-to-
    middle timeline
4. **Pairs from later batches only (LOW PRIORITY)**: Select only 5-10% from purely later batches

Focus on bullet point pairs that: - Enable duration calculations between two time points - Show sequence
    relationships between events
- Allow comparison of timing across different batches - Demonstrate temporal progression or changes over time
    - Include scheduling, deadlines, or milestone comparisons

## EXPLICIT TIME MENTION REQUIREMENTS
**ONLY absolute dates count as explicit time mentions:**
[Examples]

**THESE DO NOT COUNT as explicit time mentions:**
- Specific times - Calendar references - Specific weekdays - Relative durations - Time periods - Vague
    references - Duration spans

## IMPORTANT TIME ANCHOR RULES:
1. If BOTH bullet points contain explicit absolute dates, use them as-is
2. If ONE bullet point lacks explicit absolute dates, prepend that bullet point with its batch's Time Anchor
3. If BOTH bullet points lack explicit absolute dates, prepend both with their respective Time Anchors

FORMAT EXAMPLES:
Case 1 - Both have time mentions (no Time Anchor needed):
[Example]
Case 2 - Second bullet point lacks explicit absolute dates (add Time Anchor to second):
[Example]
Case 3 - Both bullet points lack explicit absolute dates (add Time Anchors to both):
[Example]

Return your analysis in this exact JSON format where each object contains exactly TWO related bullet points:
[{"capability": "temporal_reasoning", "batch_numbers": "1, 2", "bullet_numbers": "17, 9",
        "bullet_points": "Bullet Description: ... | Bullet Description: ..."}
]

Important formatting notes:
- The "batch_numbers" and "bullet_numbers" correspond to each other positionally
- "1, 2" and "17, 9" means: Batch 1 Bullet 17, Batch 2 Bullet 9
- Each object must contain exactly 2 bullet points separated by " | "
- Use comma-separated values for batch_number and bullet_number
- Add Time Anchors before bullet points that lack explicit time mentions
- Focus on pairs that enable duration calculation questions like "How many days between X and Y?"

Select 8-10 pairs of bullet points that would enable the most sophisticated temporal reasoning and duration
    calculation questions.

NOTE: Only output the list without any explanation before or after the list.
```

Listing 4: Candidate selection temporal reasoning prompt

```
This is a plan that contains detailed bullet points about a topic. This plan is used to generate realistic
    chat conversations between a user and an AI assistant, which are then used to evaluate the long-term
    memory capabilities of LLMs.
Your task is to analyze this plan and select bullet points that would be most effective for testing preference
    following abilities when incorporated into chat conversations.
Analyze this plan and identify bullet points labeled as "Preference Statement" and select all.

## INPUT DATA
- **PLAN**: <plan>

Focus on bullet points with:
- **Explicit preference statements**: "I prefer", "I like", "I choose", "I favor"
- **Decision choices**: Selections between options with stated reasoning
- **Personal preferences**: Style, approach, method, or format preferences
- **Avoidance statements**: "I don't like", "I avoid", "I prefer not to"
- **Priority preferences**: What user values most or considers important

Prioritize preferences that:
- Are clearly stated with specific reasoning
- Involve choices between multiple options
- Contain detailed preference explanations
- Include comparative preferences (X over Y)
- Express strong preferences or dislikes
- Relate to recurring decisions or situations
NOTE: ONLY CONSIDER ''PREFERENCE'' NOT INSTRUCTION.
```

```
Return your analysis in this exact JSON format:
[{"capability": "preference_following", "batch_numbers": 1,"bullet_numbers": 17,
        "bullet_points": "**Preference Statement:** I prefer materials that balance cost and performance;
            chose 3.5 lb/ft$^2$ MLV despite 20% higher price."}
]

Important formatting notes:
- The "batch_numbers" and "bullet_numbers" correspond to each other positionally
- "1" and "17" means: Batch 1 Bullet 17

Select ONLY <bullet_number> bullet points total that would generate the highest quality preference following
     questions.

NOTE: Only output the list without any explanation before or after the list.
```

Listing 5: Candidate selection preference following prompt

```
This is a plan that contains detailed bullet points about a topic. This plan is used to generate realistic
        chat conversations between a user and an AI assistant, which are then used to evaluate the long-term
        memory capabilities of LLMs.
Your task is to analyze this plan and select GROUPS of related bullet points that would be most effective for
        testing event ordering abilities when incorporated into chat conversations.
Event ordering tests whether the LLM can recall the chronological order in which events or topics were
        MENTIONED in the conversation, regardless of when the actual events occurred in real life.
Analyze this plan and identify GROUPS of 8-12 or more related bullet points that represent the same topic/
        theme mentioned across different batches, enabling testing of mention-order recall and conversation
        sequence understanding.

## INPUT DATA
- **PLAN**: <plan>

## CRITICAL REQUIREMENT: EARLY BATCH PRIORITIZATION
**SELECTION PRIORITY ORDER:**
1. **Groups starting in Batches 1-3 (HIGHEST PRIORITY)**: Select 70-80% of your groups with first mention in
        early batches
2. **Groups spanning early to middle batches (MEDIUM PRIORITY)**: Select 10-20% of groups that bridge early-to
        -middle timeline
3. **Groups from later batches only (LOW PRIORITY)**: Select only 5-10% from purely later batches

Focus on bullet point groups that show:
- **Same person mentioned multiple times**: Different interactions or mentions of the same person across
        batches
- **Same component/process discussed repeatedly**: Multiple mentions of the same equipment, material, or
        process
- **Same location/venue referenced**: Multiple mentions of the same place or address
- **Same decision/topic revisited**: The same subject brought up in different conversation sessions
- **Same problem/solution mentioned**: Multiple references to the same issue across different times
- **Same financial item tracked**: Multiple mentions of the same cost, budget item, or expense

Prioritize bullet point groups that:
- Contain 8-12 mentions of the same topic across different batches
- Enable questions about "In what order events X,Y,Z,... happen?" Or ...
- Allow testing of conversation chronology rather than real-world event chronology
- Test recall of mention sequence: "Which did I talk about first, second, third?"
- Focus on the order topics appeared in conversation, not when events actually happened
- Create opportunities to test conversational memory rather than factual timeline memory

Return your analysis in this exact JSON format where each object contains 3+ bullet points about the same
        topic across different batches:
[{"capability": "event_ordering", "batch_numbers": "1, 3, 5, 7", "bullet_numbers": "22, 18, 11, 27",
        "bullet_points": "- **Character & Relationship:Acoustic Consultant:** Met Rami Al-Hassan at Bahrain
            Acoustic Expo on Feb 20; he recommended HVAC silencing at $2,200. | - **Character & Relationship
            :Acoustic Consultant:** Rami conducted mid-project site visit April 3, advised on bass trap
            repositioning to improve 5 dB absorption. | - **Character & Relationship:Acoustic Consultant:**
            Rami praised progress in May 1 email; suggested minor EQ tweaks. | - **Character & Relationship:
            Acoustic Consultant:** Rami praised final results August 20; recommended ongoing maintenance and
             periodic EQ checks."}
]

Important formatting notes:
- The "batch_numbers" and "bullet_numbers" correspond to each other positionally
- "1, 2, 3, 5, 7" and "22, 18, 11, 27" means: Batch 1 Bullet 22, Batch 3 Bullet 18, Batch 5 Bullet 11, Batch 7
        Bullet 27
- Each object must contain 8-12 related bullet points separated by " | "
- Use comma-separated values for batch_number and bullet_number
- All bullet points must reference the same topic/person/component/theme
- Focus on groups that enable mention-order questions
- Test conversational chronology, not real-world event chronology

Select 8-10 groups of bullet points that would enable the most sophisticated mention-order and conversation
     sequence questions.

CRITICAL NOTE: DO NOT consider bulletpoint names for selecting the bullets. ONLY consider the bullets contents
     .
NOTE: Only output the list without any explanation before or after the list.
```

Listing 6: Candidate selection event ordering prompt

```
This is a plan that contains detailed bullet points about a topic. This plan is used to generate realistic
      chat conversations between a user and an AI assistant, which are then used to evaluate the long-term
      memory capabilities of LLMs.
Your task is to analyze this plan and select PAIRS of bullet points that would be most effective for testing
      contradiction resolution abilities when incorporated into chat conversations.
Contradiction resolution tests whether the LLM can detect and appropriately handle impossible contradictions -
       statements that logically cannot both be true simultaneously.
Analyze this project plan and identify PAIRS of bullet points where one completely contradicts the other with
      impossible contradictions. Each pair should contain statements that are logically incompatible and
      cannot both be true.

## INPUT DATA
- **PLAN**: <plan>

Focus on bullet point pairs that show:
- **Never-Statement Violations**: One bullet says "never" did something, another shows they did it
- **Always-Statement Violations**: One bullet claims "always" pattern, another breaks that pattern
- **Only-Statement Conflicts**: One bullet claims exclusivity ("only"), another contradicts it
- **Impossible Reversals**: Age going backward, timeline impossibilities, logical reversals
- **Dead-Alive Contradictions**: References to deceased people being active
- **Mutually Exclusive States**: Being in two places simultaneously, having contradictory capabilities
- **Absolute Negations**: Claiming something is impossible then showing it happened

**Types of Impossible Contradictions to look for:**
1. **Never-Statement Violations**: "Never attended X" vs "Attended X event"
2. **Always-Statement Violations**: "Always lived in Y" vs "Moved from Z to Y"
3. **Only-Statement Conflicts**: "Only child" vs "Has siblings"
4. **Timeline Impossibilities**: Events happening in wrong chronological order
5. **Capability Contradictions**: "Cannot do X" vs "Successfully did X"
6. **Location Impossibilities**: Being in two places at once
7. **Relationship Contradictions**: "Never met person" vs "Long friendship with person"

Prioritize bullet point pairs that:
- Contain completely impossible contradictions that cannot be resolved or explained
- Use absolute language ("never," "always," "only," "impossible," "cannot")
- Create clear logical impossibilities rather than simple inconsistencies
- Enable questions about detecting fundamental contradictions
- Test whether the AI can identify when statements are mutually exclusive
- Focus on contradictions that are objectively impossible, not subjective differences

Return your analysis in this exact JSON format where each object contains exactly TWO contradicting bullet
      points:
[{"capability": "contradiction_resolution", "batch_numbers": "1, 8", "bullet_numbers": "30, 29",
        "bullet_points": "• **Logical Contradiction:** Jeremiah has never attended any Bahrain Jazz Festival
                events. | • **Character & Relationship:Close Friend:** Jeremiah, 37, met at Bahrain Jazz
                Festival 2015, recommended an acoustic consultant."}
]

Important formatting notes:
- The "batch_numbers" and "bullet_numbers" correspond to each other positionally
- "1, 8" and "30, 29" means: Batch 1 Bullet 30, Batch 8 Bullet 29
- Each object must contain exactly 2 bullet points separated by " | "
- Use comma-separated values for batch_number and bullet_number
- First bullet point can be the contradiction marker or the contradicted statement
- Second bullet point should directly contradict the first with impossible logic
- Focus on pairs that test detection of fundamental logical impossibilities

Select <bullet_number> pairs of bullet points that demonstrate the clearest impossible contradictions for
      testing contradiction resolution abilities.

NOTE: Only output the list without any explanation before or after the list.
```

Listing 7: Candidate selection contradiction resolution prompt

```
This is a plan that contains detailed bullet points about a topic. This plan is used to generate realistic
      chat conversations between a user and an AI assistant, which are then used to evaluate the long-term
      memory capabilities of LLMs.
Your task is to analyze this plan and select GROUPS of bullet points that would be most effective for testing
      summarization abilities when incorporated into chat conversations.
Summarization tests whether the LLM can synthesize and condense information from across multiple conversation
      sessions into coherent, comprehensive summaries.
Analyze this plan and identify GROUPS of 8-12 related bullet points that represent topics suitable for
      summarization testing. Groups can vary in size depending on the richness and complexity of the topic.

## INPUT DATA
- **PLAN**: <plan>

## CRITICAL REQUIREMENT: EARLY BATCH PRIORITIZATION
**SELECTION PRIORITY ORDER:**
1. **Groups starting in Batches 1-3 (HIGHEST PRIORITY)**: Select 60-70% of your groups with foundational
      content from early batches
2. **Groups spanning early to middle batches (MEDIUM PRIORITY)**: Select 20-30% of groups that bridge early-to
      -middle timeline
3. **Groups from later batches only (LOW PRIORITY)**: Select only 10-20% from purely later batches

## CRITICAL REQUIREMENT: CONTENT-BASED ANALYSIS
**ANALYZE BULLET CONTENT, NOT CATEGORY NAMES:**
- Read the actual bullet point text to identify mentions of entities (people, places, items, topics, amounts,
      processes)
- The same entity might appear in different category types - include mentions across all categories
- The same process might appear in different category types - include mentions across all categories
```

```
- The same project might appear in different category types - include mentions across all categories

**SEARCH METHODOLOGY:**
1. **Identify key entities** in bullet content: names, places, amounts, equipment, topics, processes
2. **Search ALL batches** for any mention of these entities in ANY category
3. **Group by content similarity**, not category similarity
4. **Include 8-12 mentions** regardless of how they're categorized

Focus on complete topic clusters that enable summarization of:
- **Entity or Relationship Histories**: interactions, developments, or changes related to a specific person,
      organization, group, or other identifiable entity across the entire plan (complete relationship or
      entity arc)
- **End-to-End Processes**: steps, stages, or phases of a specific process, workflow, or methodology from
      initiation to conclusion across the entire plan (no missing steps)
- **Resource or Asset Lifecycles**: mentions of acquisition, allocation, usage, modification, and outcomes for
       a specific resource, asset, or material across the entire plan
- **Decision and Strategy Journeys**: details of decision-making, planning, and strategy development from
      problem identification through implementation across the entire plan
- **Problem/Challenge Resolution Narratives**: instances of identifying, analyzing, addressing, and resolving
      a particular issue or challenge across the entire plan
- **Timeline-Driven Developments**: events and updates showing chronological evolution of a specific project,
      initiative, or topic across the entire plan
- **Knowledge or Skill Development Sequences**: progress updates, milestones, and learning activities related
      to acquiring or improving a specific skill or knowledge area across the entire plan
- **Discussion and Agreement Processes**: discussions, debates, negotiations, and agreements related to a
      specific matter across the entire plan

Prioritize bullet point groups that:
- Contain rich, interconnected information suitable for synthesis
- Enable questions like "Can you summarize my interactions with X?" or "Summarize the [X] process"
- Include both factual/quantitative details and qualitative/narrative elements for well-rounded summaries
- Have varying complexity levels (simple single-topic vs. complex multi-faceted stories)
- Allow testing of information condensation across multiple conversation sessions
- Include both factual details and narrative elements for comprehensive summarization
- Create opportunities to test synthesis of scattered information into coherent narratives

Return your analysis in this exact JSON format where each object contains 8-12 related bullet points:
[{"capability": "summarization", "batch_numbers": "1, 1, 2, 3, 4, 5", "bullet_numbers": "5, 22, 18, 19, 23,
      31",
          "bullet_points": "• **Character & Relationship:Close Friend:** Jeremiah, 37, met at festival,
                recommended consultant. | • **Conflict & Resolution:Relationship Boundaries:** Jeremiah
                requested exclusive studio access; agreed to 3 hours only. | • **Character & Relationship:Close
                Friend:** Jeremiah helped install MLV, bringing snacks from bakery. | • **Character &
                Relationship:Close Friend:** Jeremiah invited me to music club to test prototype. | • **
                Character & Relationship:Close Friend:** Jeremiah brought dinner during late work session. | •
                **Goals & Progress:Milestone Celebration:** Hosted listening party with Jeremiah, Jon, and Tonya
                ."}
]

Important formatting notes:
- The "batch_numbers" and "bullet_numbers" correspond to each other positionally
- "1, 1, 2, 3, 4, 5" and "5, 22, 18, 19, 23, 31" means: Batch 1 Bullet 5, Batch 1 Bullet 22, Batch 2 Bullet
      18, Batch 3 Bullet 19, Batch 4 Bullet 23, Batch 5 Bullet 31
- Each bullet point separated by " | "
- Use comma-separated values for batch_number and bullet_number
- Vary group sizes based on topic complexity and richness
- Focus on groups that enable comprehensive summarization questions
- Include both simple single-topic and complex multi-topic groups
- **NO LIMIT on number of bullet points** - include as many as needed for complete coverage

CRITICAL NOTES:
- **ANALYZE BULLET CONTENT, NOT CATEGORY NAMES**: Search for mentions of entities in the actual text
- **IGNORE CATEGORY LABELS**: The same entity mentioned in different category types should all be grouped
      together
- **COMPREHENSIVE ENTITY SEARCH**: For each entity/topic/process, scan ALL batches and ALL categories for any
      mention
- Include 8-12 mentions regardless of bullet point category if they reference the same entity/topic in the
      content

Select 7-9 groups of bullet points with COMPLETE mention coverage that would enable the most sophisticated and
      comprehensive summarization questions across different complexity levels.

NOTE: Only output the list without any explanation before or after the list.
```

Listing 8: Candidate selection summarization prompt

```
This is a plan that contains detailed bullet points about a topic. This plan is used to generate realistic
      chat conversations between a user and an AI assistant, which are then used to evaluate the long-term
      memory capabilities of LLMs.
Your task is to analyze this plan and select bullet points that would be most effective for testing
      instruction following abilities when incorporated into chat conversations.
Analyze this plan and identify bullet points that contain user instructions ideal for testing whether the LLM
      remembers and follows user-given instructions.

## INPUT DATA
- **PLAN**: <plan>

Focus on bullet points with:
- **User Instruction** category/label
- **Explicit instruction statements**: "Always", "Never", "When I ask about X, do Y"
- **Behavioral directives**: How the AI should respond or behave
```

```
- **Format instructions**: Specific response formats or structures requested
- **Content instructions**: What to include or exclude in responses
- **Process instructions**: How to handle specific types of requests

Look specifically for bullet points labeled as "User Instruction" that contain:
- Clear directive language ("Always provide", "Never include", "When I ask")
- Specific behavioral expectations for the AI assistant
- Conditional instructions ("When I ask about X, do Y")
- Response formatting requirements
- Content inclusion/exclusion rules

Return your analysis in this exact JSON format:
[{"capability": "instruction_following", "batch_numbers": 1,"bullet_numbers": 32,
        "bullet_points": "User Instruction: Always provide detailed cost breakdowns when I ask about budget
                estimates."}
]

Important formatting notes:
- The "batch_numbers" and "bullet_numbers" correspond to each other positionally
- "1" and "32" means: Batch 1 Bullet 32
- Include the full bullet point text as it appears in the plan
- Focus specifically on "User Instruction" labeled bullet points

Select all bullet points labeled as "User Instruction" from each batch (approximately 10 total).

NOTE: Only output the list without any explanation before or after the list.
```

Listing 9: Candidate selection instruction following prompt

```
You are tasked with generating a probing question to test information extraction capabilities of LLMs. You
    will be given a bullet point and the corresponding multi-turn dialog between a user and assistant that
    incorporates this bullet point information.

Your task is to create ONE question that tests whether an LLM can precisely extract and recall specific
    factual details from the conversation through indirect questioning that requires synthesizing multiple
    details from different parts of the conversation.

## INPUT DATA
- **BULLET POINT**: <bullet_point>
- **CONVERSATION TURNS**: <conversation_turns>

## CRITICAL REQUIREMENT: INFORMATION EXTRACTION
- The question MUST NOT directly ask for the information being tested
- Ask about related topics/contexts that require the LLM to synthesize multiple details from different
    conversation parts
- Force the LLM to extract and combine information scattered across different conversation turns
- Make the LLM demonstrate knowledge of facts without being directly asked for them
- Require connecting and integrating information from multiple different conversation elements
- Remove ALL specific details from the question that would give away the answer

## FORBIDDEN QUESTION ELEMENTS
- Do NOT repeat specific names, numbers, or details being tested
- Do NOT mention key characteristics or attributes being extracted
- Do NOT include descriptive words that hint at the answer
- Do NOT reference specific categories or types being tested
- Do NOT use qualifying details that narrow down the answer

## QUESTION LANGUAGE REQUIREMENTS
- Write questions as if the USER is asking them naturally
- Questions MUST ONLY BE from USER language not ASSISTANT
- **If testing information from USER messages**: Use first person ("I", "my", "me") in question -> Answer uses
    ("you", "your")
  - Example: "How did I decide on the location?" -> "You decided on the location because..."
- **If testing information from ASSISTANT messages**: Use second person ("you", "your") in question -> Answer
    uses ("I", "my")
  - Example: "What steps did you suggest for handling this?" -> "I suggested doing..."
- Avoid phrases like "according to the conversation", "based on what was discussed", "from our chat history"
- Make questions sound conversational and natural
- Questions should flow naturally as if continuing the conversation
- Ask about context, or relationships rather than direct facts

## INDIRECT QUESTIONING STRATEGIES
### 1. **Context-Based Recall** Ask about the surrounding circumstances instead of the exact fact
[Example]
### 2. **Comparison Questions** Encourage differentiation between similar elements
[Example]
### 3. **Timeline Integration** Link facts to their sequence in time
[Example]
### 4. **Problem-Solution Context** Frame questions around issues and how they were addressed:
[Example]
### 5. **Discovery and Learning Process** Focus on the origin of knowledge or awareness
[Example]
### 6. **Relationship and Connection Context** Test understanding of associations
[Example]

## FORBIDDEN DIRECT QUESTIONS
[Examples]

## CHAT ID TRACKING REQUIREMENT
- You MUST identify which specific chat_id(s) contain the information being tested - List ALL chat_ids where
    the answer appears
```

```
- NOTE: If the answer is spread out between multiple chat_ids, group them in one list - NOTE: DO NOT INCLUDE
      chat_ids in the answer
- If answer spans multiple chats, include all relevant chat_ids - Use the exact chat_id numbers from the
      conversation turns

## DIFFICULTY LEVEL: HARD
- **Hard**: Requires synthesizing multiple details from different parts of conversation
- Force integration of information scattered across multiple conversation turns
- Test ability to connect related facts from different conversation contexts
- Require deep understanding and synthesis rather than simple recall

## OUTPUT FORMAT
Return your analysis in this exact JSON format:
{
    "question": [], "answer": [], "difficulty": "hard", "question_type": # one of: [] "conversation_reference
        ": "", "key_facts_tested": "",
    "extraction_challenge": "", "source_chat_ids": [X, Y, ...]
}

## IMPORTANT REQUIREMENTS
1. **Indirect questioning**: Ask about context rather than direct facts
2. **Question source flexibility**: Questions can be based on information from EITHER user messages OR
      assistant messages
3. **Perspective matching**: Question perspective must match the source of information:
   - **User info** -> "I/my/me" question -> "you/your" answer
   - **Assistant info** -> "you/your" question -> "I/my" answer
4. **Assistant information questions**: When testing assistant advice/suggestions, use "What did you suggest/
      recommend/advise..." format
5. **Multi-detail synthesis**: Question should require combining information from different conversation parts
6. **Cross-turn integration**: Force LLM to connect scattered information across multiple turns
7. **Complex reasoning**: Require understanding of relationships and synthesis of multiple elements
8. **Challenging extraction**: Force LLM to demonstrate knowledge through indirect demonstration

Generate ONE high-quality indirect information extraction question that tests recall of specific factual
      details through contextual questioning requiring synthesis of multiple details from different parts of
      the conversation.

NOTE: Only output the JSON object without any explanation before or after.
```

Listing 10: Information extraction probing question generation prompt

```
You are tasked with generating a probing question to test multi-session reasoning capabilities of LLMs. You
      will be given multiple related bullet points and the corresponding multi-turn dialogs between a user and
       assistant that incorporate this information across different conversation sessions.

Your task is to create ONE question that tests whether an LLM can perform complex multi-hop reasoning,
      synthesis, and analysis across 4+ conversation sessions.

## INPUT DATA
- **BULLET POINTS**: <bullet_points>
- **CONVERSATION TURNS**: <conversation_turns>

## CRITICAL REQUIREMENT: HARD MULTI-SESSION REASONING
- The question MUST NOT include any explicit number, dates, times, duration, or temporal references
- Focus on complex synthesis requiring multi-hop reasoning across 4+ sessions
- Test sophisticated analysis that requires connecting multiple data points
- Ask for complex calculations, patterns, or insights that need advanced reasoning

## QUESTION GENERATION GUIDELINES
Focus on creating questions that require: - **Complex Aggregation** - **Advanced Synthesis** - **Multi-hop
      Reasoning**
- **Pattern Recognition** - **Performance Evaluation** - **Comparative Analysis** - **Predictive Reasoning**

## QUESTION TYPES TO GENERATE (HARD LEVEL)
1. **Complex Multi-hop Calculation** 2. **Performance Evaluation** 3. **Multi-variable Comparison** 4. **
      Complex Evolution Analysis**

## REASONING COMPLEXITY LEVEL: HARD
- **Hard**: Requires complex multi-hop reasoning, synthesis, and analysis across 4+ sessions
- Focus on sophisticated calculations or insights requiring advanced reasoning
- Test ability to identify complex patterns, correlations, or relationships
- Include deep analytical thinking and synthesis of multiple data points

## QUESTION LANGUAGE REQUIREMENTS
- Write questions as if the USER is asking them naturally
- **If testing information from USER messages**: Use first person ("I", "my", "me") in question -> Answer uses
       ("you", "your")
  - Example: "How did I decide on the location?" -> "You decided on the location because..."
- **If testing information from ASSISTANT messages**: Use second person ("you", "your") in question -> Answer
      uses ("I", "my")
  - Example: "What steps did you suggest for handling this?" -> "I suggested doing..."
- Avoid phrases like "according to the conversation", "based on what was discussed", "from our chat history"
- Make questions sound conversational and natural
- Questions should flow naturally as if continuing the conversation

## CHAT ID TRACKING REQUIREMENT
- You MUST identify which specific chat_id(s) contain the information needed for reasoning
- List ALL chat_ids where relevant information appears across the reasoning chain
- NOTE: If the answer is spread out between multiple chat_ids, group them in one list
- NOTE: DO NOT INCLUDE chat_ids in the answer
- If reasoning spans multiple chats, include all relevant chat_ids
```

```
- Use the exact chat_id numbers from the conversation turns

## OUTPUT FORMAT
You will output exactly ONE JSON object matching this schema:
{
  "question": string, "answer": string, "difficulty": "hard", "reasoning_type": [Some categories]
  "sessions_required": integer, "conversation_references": [string,...], "reasoning_steps": [string,...], "
      source_chat_ids": [integer,...]
}

## IMPORTANT REQUIREMENTS
1. **Complex multi-session dependency**: Question must require sophisticated information synthesis from 4+
      conversation sessions
2. **Question source**: The question and answer to it MUST BE based on information from USER messages in
      CONVERSATION TURNS. You MUST NOT generate questions about assistant responses or suggestions.
3. **User information only**: Only create questions that test details the user provided, not assistant advice
      or recommendations.
5. **Advanced reasoning path**: Provide complex reasoning steps that require multi-hop thinking
6. **Precise answer**: Give exact answer that demonstrates complex analysis
7. **Session references**: Note which sessions contain relevant information
8. **High complexity**: Ensure question requires advanced multi-session reasoning and sophisticated synthesis

Generate ONE high-quality hard multi-session reasoning question.

NOTE: Only output the JSON object without any explanation before or after.
```

Listing 11: Multi-hop reasoning probing question generation prompt

```
You are tasked with generating a probing question to test knowledge update capabilities of LLMs. You will be
      given two related bullet points (original information and updated information) and the corresponding
      multi-turn dialogs between a user and assistant that incorporate both pieces of information across
      different conversation sessions.

Your task is to create ONE question that asks about the current/updated state of information, testing whether
      the LLM correctly recalls the most recent version rather than outdated information.

## INPUT DATA
- **BULLET POINTS**: <bullet_points>
- **CONVERSATION TURNS**: <conversation_turns>

## CRITICAL REQUIREMENT: NO SPECIFIC CONTEXT HINTS
- MUST NOT GIVE ANY INFORMATION RELATED TO OLD AND UPDATED INFORMATION/FACTS OR ANY HINTS THAT THERE IS UPDATE
      AT ALL
_ DO NOT use words like: currently, now, ... that shows update of information
- The question MUST NOT include specific dates, times, locations, or detailed circumstances
- Do NOT reference specific events, phases, or instances that would hint at which version to recall
- Ask about the general current state, not specific occurrences

## FACTUAL UPDATE IDENTIFICATION
Before creating the question:
1. Identify the EXACT fact that was updated in the "Information Update" bullet
2. Determine what the original fact was vs. the updated fact
3. Create a question that tests recall of the updated fact specifically
4. Ensure question asks for the factual detail, not procedures or implications

## QUESTION GENERATION GUIDELINES
Focus on creating questions that:
- **Ask about current state**: Question the most recent/updated version of information
- **Test update retention**: Whether LLM remembers the latest information, not the original
- **Avoid mentioning changes**: Don't explicitly ask "how did X change" - just ask about current state
- **Target updated facts**: Focus on information that was specifically updated/changed

## QUESTION TYPES TO GENERATE
1. **Current State Query** 2. **Latest Status** 3. **Updated Decision** 4. **Final Information** 5. **Recent
      Details**

## QUESTION LANGUAGE REQUIREMENTS
- Write questions as if the USER is asking them naturally
- **If testing information from USER messages**: Use first person ("I", "my", "me") in question -> Answer uses
      ("you", "your")
  - Example: "How did I decide on the location?" -> "You decided on the location because..."
- **If testing information from ASSISTANT messages**: Use second person ("you", "your") in question -> Answer
      uses ("I", "my")
  - Example: "What steps did you suggest for handling this?" -> "I suggested doing..."
- Avoid phrases like "according to the conversation", "based on what was discussed", "from our chat history"
- Make questions sound conversational and natural
- Questions should flow naturally as if continuing the conversation

## CHAT ID TRACKING REQUIREMENT
- You MUST identify which specific chat_id(s) contain the original and updated information
- List the chat_id with the original information and the chat_id with the updated information
- NOTE: If the answer is spread out between multiple chat_ids, group them in one list
- NOTE: DO NOT INCLUDE chat_ids in the answer
- Use the exact chat_id numbers from the conversation turns

## OUTPUT FORMAT
Return your analysis in this exact JSON format:
{
    "question": "", "answer": "", "difficulty": "moderate", "update_type": "", "tests_retention_of": "", "
          conversation_references": "",
    "potential_confusion": "", "source_chat_ids": {"original_info": [, ], "updated_info": [, ]}
```

```
}

## IMPORTANT REQUIREMENTS
1. Do not mention how or when the value changed.Question text must not contain words like 'after,''negotiated
    ,''updated,''revised,' or any mention of a change process.
2. **Current state focus**: Question must ask about the updated/current information only
3. **No change language**: Avoid words like "changed," "updated," "revised" in the question
4. **Updated answer**: Answer must reflect the most recent version of the information
5. **Confusion potential**: Note what outdated information the LLM might incorrectly recall
6. **Natural phrasing**: Question should sound like asking for current facts, not testing memory updates
7. 'Include at least **two** entries in 'conversation_references': one for the original fact session and one
    for the updated fact session.'

Generate ONE knowledge update question that tests whether the LLM correctly recalls the updated information
    rather than the original outdated version.

CRITICAL NOTE: Do not mention how or when the value changed.Question text must not contain words like 'after,'
    'negotiated,''updated,''revised,' or any mention of a change process.

NOTE: Only output the JSON object without any explanation before or after.
```

Listing 12: Knowledge update probing question generation prompt

```
You are tasked with generating a probing question to test temporal reasoning capabilities of LLMs. You will be
    given two related bullet points with temporal information and the corresponding multi-turn dialogs
    between a user and assistant that incorporate both time points across different conversation sessions.

Your task is to create ONE question that tests whether an LLM can perform complex multi-step temporal
    reasoning, advanced calculations, pattern analysis, or synthesis of multiple temporal relationships.

## INPUT DATA
- **BULLET POINTS**: <bullet_points>
- **CONVERSATION TURNS**: <conversation_turns>

## CRITICAL REQUIREMENTS: CHALLENGING TEMPORAL REASONING
- The question MUST NOT include any explicit dates, times, or temporal references
- Use only event descriptions that require the LLM to recall temporal information
- Create questions that require complex temporal reasoning, not simple lookups
- Test sophisticated temporal understanding across multiple conversation sessions

## ADVANCED QUESTION GENERATION GUIDELINES
Focus on creating questions that test:
- **Complex duration calculations** - **Relative temporal positioning** - **Cross-session temporal synthesis**
    - **Temporal pattern recognition**
- **Conditional temporal logic** - **Temporal inference**

## SOPHISTICATED QUESTION TYPES
### **Duration & Calculation Questions** 1. **Multi-hop Duration** [Other examples]

### **Sequence & Ordering Questions** 6. **Complex Sequencing** [Other examples]

### **Comparative & Analytical Questions** 10. **Timeline Comparison** [Other examples]

### **Inferential & Complex Questions** 15. **Causal Temporal** [Other examples]

### **Between-Time Information Extraction**:  21. "What/Who/Where/How much/When [specific query] between [
    starting point] and [ending point]?"

## FORBIDDEN QUESTION ELEMENTS
- Do NOT mention specific dates, times, or numbers in the question
- Do NOT use phrases like "on [specific date]" or "after [X] days/weeks/months"
[Other examples]

## GOOD VS BAD EXAMPLES
[Examples]

## TEMPORAL COMPLEXITY LEVEL: HARD
- **Hard**: Requires multi-step temporal reasoning across 3+ conversation sessions, complex calculations,
    pattern analysis, temporal inference, or synthesis of multiple temporal relationships

## QUESTION LANGUAGE REQUIREMENTS
- Write questions as if the USER is asking them naturally
- **If testing information from USER messages**: Use first person ("I", "my", "me") in question -> Answer uses
    ("you", "your")
  - Example: "How did I decide on the location?" -> "You decided on the location because..."
- **If testing information from ASSISTANT messages**: Use second person ("you", "your") in question -> Answer
    uses ("I", "my")
  - Example: "What steps did you suggest for handling this?" -> "I suggested doing..."
- Avoid phrases like "according to the conversation", "based on what was discussed"
- Make questions sound conversational and natural
- Questions should require deep temporal reasoning to answer

## CHAT ID TRACKING REQUIREMENT
- You MUST identify which specific chat_id(s) contain the temporal information for both events
- List the chat_id for the first temporal event and the chat_id for the second temporal event
- NOTE: If the answer is spread out between multiple chat_ids, group them in one list
- NOTE: DO NOT INCLUDE chat_ids in the answer
- Use the exact chat_id numbers from the conversation turns

## OUTPUT FORMAT
Return your analysis in this exact JSON format:
```

```
{
    "question": "", "answer": "", "difficulty": "hard", "temporal_type": "", "time_points": "", "
        conversation_references": "",
    "calculation_required": "", "complexity_factors": "", "source_chat_ids": { "first_event": [12, 15], "
        second_event": [28, 31]}
}

## IMPORTANT REQUIREMENTS
1. **No temporal hints**: Question must NOT include dates, times, or temporal references
2. **Complex reasoning**: Require multi-step temporal analysis, not simple lookup
3. **Cross-session dependency**: Events should span multiple conversation sessions
4. **Advanced difficulty**: Always use "hard" difficulty level
5. **Sophisticated types**: Use advanced question types that test complex temporal reasoning
6. **Pattern recognition**: Include questions that test temporal patterns and relationships

Generate ONE high-quality, challenging temporal reasoning question that requires sophisticated temporal
    analysis between the provided bullet points.

NOTE: Only output the JSON object without any explanation before or after.
```

Listing 13: Temporal reasoning probing question generation prompt

```
You are tasked with generating multiple probing questions to test abstention capabilities of LLMs. You will be
    given a plan containing bullet points that will be used to generate conversations between a user and AI
    assistant.

Your task is to create questions that test whether an LLM can appropriately respond with "I don't know" or "
    Based on the provided chat, there is no information related to [topic]" when asked about information
    that is missing or unavailable from the plan.

## INPUT DATA
- **PLAN**: <plan>

## CRITICAL REQUIREMENT: EARLY BATCH PRIORITIZATION
**SELECTION PRIORITY ORDER:**
1. **Batch 1-3 (HIGHEST PRIORITY)**: Select 70-80% of your questions from these early batches
2. **Batch 4-6 (MEDIUM PRIORITY)**: Select 10-20% of your questions from these middle batches
3. **Batch 7+ (LOW PRIORITY)**: Select only 5-10% of your questions from later batches

## CRITICAL REQUIREMENT: CONTEXT-BASED HARD QUESTIONS
- Questions MUST be related to the plan's context and domain
- Ask about plausible information that SEEMS like it should be available but isn't
- Use names, places, events, and topics mentioned in the plan
- Ask for specific details about mentioned entities that aren't actually provided
- Create questions that would be natural follow-ups but can't be answered

## QUESTION GENERATION GUIDELINES
Focus on creating questions that test appropriate abstention for:
### 1. Missing Details About Mentioned People/Entities [Exmaples]
### 2. Unavailable Specifics About Mentioned Events/Activities [Examples]
### 3. Missing Information About Referenced Sources/Materials [Examples]
### 4. Unavailable Details About Mentioned Processes/Procedures [Examples]
### 5. Missing Context About Mentioned Decisions/Choices [Examples]
### 6. Unavailable Quantitative/Measurement Details [Examples]
### 7. Missing Emotional/Subjective Information [Examples]
### 8. Unavailable Technical/Specialized Details [Examples]
### 9. Unavailable Future/Predictive Information [Examples]

## ABSTENTION QUESTION TYPES
Only generate these two types: 1. **Unavailable Information Questions**: Ask about topics, people, events, or
    details that are completely absent from the plan
2. **Missing Detail Questions**: Ask for specific details about topics that may be mentioned generally but
    lack the requested specifics

## DIFFICULTY LEVELS
Generate questions of varying abstention difficulty:
- **Easy**: Ask for details about mentioned entities that seem like they should be available
- **Medium**: Ask for specific information about mentioned processes/events that weren't detailed
- **Hard**: Ask for nuanced details that are plausible but not provided

## QUESTION LANGUAGE REQUIREMENTS
- Write questions as if the USER is asking them naturally
- **If testing information from USER messages**: Use first person ("I", "my", "me") in question -> Answer uses
    ("you", "your")
  - Example: "How did I decide on the location?" -> "You decided on the location because..."
- **If testing information from ASSISTANT messages**: Use second person ("you", "your") in question -> Answer
    uses ("I", "my")
  - Example: "What steps did you suggest for handling this?" -> "I suggested doing..."
- Avoid phrases like "according to the conversation", "based on what was discussed", "from our chat history"
- Make questions sound conversational and natural
- Questions should flow naturally as if continuing the conversation

## OUTPUT FORMAT
Return your analysis in this exact JSON format:
[{"question": "", "ideal_response": "", "difficulty": "easy", "abstention_type": "", "why_unanswerable": "",
    "plan_reference": ""},
]

## IMPORTANT REQUIREMENTS
1. **Context-based questions**: Questions must relate to plan content and use mentioned entities
2. **Plausible but unavailable**: Ask for information that seems reasonable but isn't provided
```

```
3. **Natural follow-ups**: Questions should sound like logical next questions about mentioned topics
4. **Challenging abstention**: Make it harder for LLM to recognize information is missing
5. **Domain consistency**: Questions should fit naturally within the plan's subject matter

Generate 8-15 abstention questions that test the LLM's ability to appropriately say "Based on the provided
     chat, there is no information related to [topic]" when information is missing or unavailable in the plan
     .

NOTE: Only output the JSON array without any explanation before or after the list.
```

Listing 14: Abstention probing question generation prompt

```
You are tasked with generating a probing question to test preference following capabilities of LLMs. You will
     be given a bullet point containing a user preference and corresponding conversation turns where this
     preference was mentioned.

Your task is to create ONE question that tests whether an LLM remembers and respects the user's stated
     preference when making recommendations or providing advice.

## INPUT DATA
- **BULLET POINT**: <bullet_point>
- **CONVERSATION TURNS**: <conversation_turns>

## CRITICAL REQUIREMENT: COMPLETELY NEUTRAL QUESTION
- The question MUST BE COMPLETELY NEUTRAL with NO hints about any preference
- The question MUST NOT contain ANY words that suggest evaluation criteria
- The question should be so neutral it could test ANY type of preference in that domain
- The LLM should only know about the preference from previous conversation history

## MANDATORY PREFERENCE ANALYSIS STEP
BEFORE writing the question, you MUST:
1. **Extract ALL preference-related words** from the bullet point
2. **List ALL forbidden terms** including synonyms and related concepts
3. **Verify your question contains NONE of these terms**

## FORBIDDEN QUESTION ELEMENTS
[Examples]

## QUESTION LANGUAGE REQUIREMENTS
- Write questions as if the USER is asking them naturally
- Use first person ("I", "my", "me") when referring to the user
- Use second person ("you") when addressing the assistant
- Avoid phrases like "according to the conversation", "based on what was discussed", "from our chat history"
- Make questions sound conversational and natural
- Questions should flow naturally as if continuing the conversation
- NEVER mention the preference, decision criteria, or reasoning from the bullet point

## CHAT ID TRACKING REQUIREMENT
- You MUST identify which specific chat_id(s) contain the preference information
- List ALL chat_ids where the preference was mentioned or demonstrated
- NOTE: If the answer is spread out between multiple chat_ids, group them in one list
- NOTE: DO NOT INCLUDE chat_ids in the answer
- Use the exact chat_id numbers from the conversation turns

## OUTPUT FORMAT
Return your analysis in this exact JSON format:
{
    "question": "", "preference_being_tested": "", "expected_compliance": "", "compliance_indicators": [], "
        non_compliance_signs": [],
    "difficulty": "medium", "preference_type": "","source_chat_ids": []
}

## IMPORTANT REQUIREMENTS
1. **Preference-triggering question**: Question must create a situation where the stated preference should
     guide the response
2. **Clear compliance expectations**: Define what respecting the preference looks like
3. **Measurable indicators**: Provide specific signs of following vs. ignoring the preference
4. **Natural question phrasing**: Question should sound realistic and conversational
5. **Preference relevance**: Question must relate to the same domain/context as the stated preference

Generate ONE preference following question that tests whether the LLM remembers and applies the user's stated
     preference when providing recommendations or advice.

NOTE: Only output the JSON object without any explanation before or after.
```

Listing 15: Preference following probing question generation prompt

```
You are tasked with generating a probing question to test event ordering capabilities of LLMs. You will be
     given multiple related bullet points about the same topic/theme and the corresponding multi-turn dialogs
     between a user and assistant that incorporate these mentions across different conversation sessions.

Your task is to create ONE question that tests whether an LLM can recall the chronological order in which
     topics were MENTIONED in the conversation, regardless of when the actual events occurred in real life.

## INPUT DATA
- **BULLET POINTS**: <bullet_points>
- **CONVERSATION TURNS**: <conversation_turns>

## CRITICAL REQUIREMENTS: NO SPOILERS OR TIME HINTS
- The question MUST NOT list, mention, or hint at the specific events/mentions being tested
```

```
- The question should only specify the general topic/theme, not the individual events
- Do NOT include any time references, dates, or temporal hints in the question
- The LLM must recall and order the mentions entirely from memory without any hints

## ADVANCED QUESTION TYPES FOR EVENT ORDERING
### **Sequential Ordering Questions** 1. **General Mention Order** [Other types]
### **Comparative Ordering Questions** 4. **Priority Sequencing** [Other types]
### **Pattern Recognition Questions** 7. **Mention Pattern**: [Other types]
### **Analytical Ordering Questions** 10. **Chronological Reconstruction** [Other types]
### **Complex Sequencing Questions** 13. **Multi-faceted Ordering** [Other types]

## FORBIDDEN QUESTION ELEMENTS
- Do NOT list specific events like "including X, Y, and Z"
- Do NOT mention specific details, dates, times, or temporal references
- Do NOT provide hints about what mentions to look for
- Do NOT reference specific timeframes (e.g., "in February", "during spring", "early in project")
- Do NOT use temporal words like "first", "then", "after", "before" in the question

## GOOD VS BAD EXAMPLES
[Examples]

## ORDERING COMPLEXITY LEVEL: HARD
- **Hard**: Either 8-10 mentions requiring chronological reconstruction or 8-10 mentions with complex
      conversational patterns or 8+ mentions requiring sophisticated sequence analysis
- Focus on advanced sequence reconstruction with sophisticated analysis
- Test ability to track complex mention patterns across multiple sessions
- Include scenarios requiring expert-level sequence analysis and pattern recognition

## QUESTION LANGUAGE REQUIREMENTS
- Write questions as if the USER is asking them naturally
- Use first person ("I", "my", "me") when referring to the user
- Use second person ("you") when addressing the assistant
- Avoid phrases like "according to the conversation", "based on what was discussed", "from our chat history"
- Make questions sound conversational and natural
- Questions should flow naturally as if continuing the conversation
- NEVER include temporal references or time-related words

## MANDATORY QUESTION ENDING REQUIREMENT
- ALL questions MUST end with the phrase "in order" if previously didn't mention the order
- The order should mention ONLY ONCE
- This signals to the LLM that a sequential, ordered response is expected

## CHAT ID TRACKING REQUIREMENT [ONLY for source_chat_ids filed in JSON object]
- You MUST identify which specific chat_id(s) contain each mention in the ordering sequence
- List ALL chat_ids for each mention in chronological order in source_chat_ids field
- NOTE: If the answer is spread out between multiple chat_ids, group them in one list
- NOTE: DO NOT INCLUDE chat_ids in the answer
- Use the exact chat_id numbers from the conversation turns
- Map each mention to its source chat_id

## OUTPUT FORMAT
Return your analysis in this exact JSON format:
{
    "question": "", "answer": "", "difficulty": "hard", "ordering_type": "", "total_mentions": , "
        conversation_references": [],
    "ordering_tested": [], "complexity_factors": [], "source_chat_ids": []
}

## IMPORTANT REQUIREMENTS
1. **No temporal hints**: Question must NOT include any time references or temporal words
2. **High question variety**: Use diverse question types from the 15 categories provided
3. **Complex reasoning**: Require sophisticated mention-order analysis
4. **Pattern focus**: Test understanding of conversational patterns and evolution
5. **Advanced difficulty**: Always use "hard" difficulty level
6. **General topic focus**: Ask about broad themes, not specific events
7. **Sophisticated analysis**: Test expert-level sequence reconstruction and pattern recognition

Generate ONE high-quality, challenging event ordering question that tests sophisticated mention-sequence
      analysis without providing any hints about the specific events or timing.

NOTE: Only output the JSON object without any explanation before or after.
```

Listing 16: Event ordering probing question generation prompt

```
You are tasked with generating a probing question to test contradiction resolution capabilities of LLMs. You
      will be given two contradictory bullet points and the corresponding multi-turn dialogs between a user
      and assistant that incorporate both contradictory statements across different conversation sessions.

Your task is to create ONE question that asks about the contradicted topic as if seeking factual information,
      testing whether the LLM will detect the contradiction and ask for clarification rather than giving a
      definitive answer.

## INPUT DATA
- **BULLET POINTS**: <bullet_points>
- **CONVERSATION TURNS**: <conversation_turns>

## QUESTION GENERATION GUIDELINES
Focus on creating questions that:
- **Ask about the contradicted fact**: Question the topic that has contradictory information
- **Sound like normal fact-seeking**: Don't hint that there's a contradiction
- **Test contradiction detection**: Whether LLM notices conflicting information
```

```
- **Expect clarification request**: LLM should ask which contradictory statement is correct
- **Avoid revealing the contradiction**: Question should seem like routine information request

## CRITICAL REQUIREMENT:
- The question MUST be general and broad about the contradicted topic
- Do NOT include specific dates, times, locations, or detailed circumstances
- Do NOT reference specific events or instances that would narrow down the contradiction
- Ask about the general pattern/behavior, not specific occurrences
- This tests whether the LLM can detect contradictions across all related mentions

## QUESTION LANGUAGE REQUIREMENTS
- Write questions as if the USER is asking them naturally
- Use first person ("I", "my", "me") when referring to the user
- Use second person ("you") when addressing the assistant
- Avoid phrases like "according to the conversation", "based on what was discussed", "from our chat history"
- Make questions sound conversational and natural
- Questions should flow naturally as if continuing the conversation

## QUESTION FORMAT Generate questions that ask about the contradicted topic directly: [Examples]

## EXPECTED LLM BEHAVIOR The ideal LLM response should: [Example]

## CHAT ID TRACKING REQUIREMENT
- You MUST identify which specific chat_id(s) contain each contradictory statement
- List the chat_id for the first contradictory statement and the chat_id for the second contradictory
      statement
- NOTE: If the answer is spread out between multiple chat_ids, group them in one list
- NOTE: DO NOT INCLUDE chat_ids in the answer
- Use the exact chat_id numbers from the conversation turns

## OUTPUT FORMAT
Return your analysis in this exact JSON format:
{
    "question": "", "ideal_answer": "", "difficulty": "", "contradiction_type": "", "topic_questioned": "", "
        conversation_references": [],
    "tests_for": "", "source_chat_ids": {"first_statement": [, ], "second_statement": [, ]}
}

## IMPORTANT REQUIREMENTS
1. **Natural question phrasing**: Question should sound like normal fact-seeking, not contradiction testing
2. **Topic focus**: Ask directly about the contradicted subject
3. **Contradiction detection expectation**: LLM should notice and ask for clarification
4. **No hint giving**: Don't reveal that there's a contradiction in the question
5. **Clarification seeking**: Ideal response should ask which statement is correct

Generate ONE contradiction resolution question that tests whether the LLM will detect the contradiction and
      appropriately request clarification when asked about the contradicted topic.

CRITICAL NOTE: Do NOT include specific dates, times, locations, or detailed circumstances in the question that
        make the question easy.
NOTE: Only output the JSON object without any explanation before or after.
```

Listing 17: Contradiction resolution probing question generation prompt

```
You are tasked with generating a probing question to test advanced summarization capabilities of LLMs. You
      will be given 6-8 related bullet points about the same topic/theme and the corresponding multi-turn
      dialogs between a user and assistant that incorporate this information across different conversation
      sessions.

Your task is to create ONE question that tests whether an LLM can synthesize and condense complex, multi-
      faceted information from across 4+ conversation sessions into sophisticated, comprehensive summaries.

## INPUT DATA
- **BULLET POINTS**: <bullet_points>
- **CONVERSATION TURNS**: <conversation_turns>

## CRITICAL REQUIREMENT: HARD SUMMARIZATION
- Focus on complex information synthesis from 8-10 bullet points
- Test comprehensive synthesis requiring sophisticated analysis
- Require advanced narrative construction with multiple threads
- Ask for summaries that demonstrate deep understanding and integration

## CRITICAL REQUIREMENT: NEUTRAL SUMMARIZATION TESTING
- The question MUST NOT reveal what should be included in the summary
- The question MUST mention *only* the overarching topic-no specific bullet-point details, subtopics, phases,
      or technical terms may appear.
- The question MUST NOT hint at the structure or content of the expected answer
- The question should be maximally generic, forcing the LLM to identify and synthesize all relevant
      information independently

## QUESTION GENERATION GUIDELINES
Focus on creating questions that test: - **Complex information synthesis** - **Advanced cross-session
      condensation** - **Comprehensive overview**
- **Sophisticated narrative coherence** - **Strategic detail prioritization**

## QUESTION TYPES TO GENERATE (HARD LEVEL)
1. **Complex Relationship & Interaction Summary** 2. **Complete Sequence & Event Analysis** 3. **Resource,
      Effort, & Timeline Evolution**
4. **Multi-factor Decision Process Review** 5. **Problem-to-Resolution Journey** 6. **Chronological
      Development Overview**
7. **Knowledge & Insight Integration Summary** 8. **Complex Negotiation or Agreement Path**
```

```
## SUMMARIZATION COMPLEXITY LEVEL: HARD
- **Hard**: 8-10 bullet points requiring comprehensive synthesis with detailed progression
- Focus on sophisticated analysis requiring understanding of complex relationships and patterns
- Test ability to synthesize multiple narrative threads and extensive information
- Include advanced narrative elements with multi-layered connections and sophisticated causation

## QUESTION LANGUAGE REQUIREMENTS
- Write questions as if the USER is asking them naturally
- Use first person ("I", "my", "me") when referring to the user
- Use second person ("you") when addressing the assistant
- Avoid phrases like "according to the conversation", "based on what was discussed", "from our chat history"
- Make questions sound conversational and natural
- Questions should flow naturally as if continuing the conversation

## CHAT ID TRACKING REQUIREMENT
- You MUST identify which specific chat_id(s) contain the information needed for the summary
- List ALL chat_ids where relevant summary information appears
- NOTE: If the answer is spread out between multiple chat_ids, group them in one list
- NOTE: DO NOT INCLUDE chat_ids in the answer
- Use the exact chat_id numbers from the conversation turns

## OUTPUT FORMAT
Return your analysis in this exact JSON format:
{
    "question": "", "ideal_summary": "", "difficulty": "hard", "summarization_type": "",
    "bullet_points_covered": , "conversation_sessions": , "key_elements_tested": [], "synthesis_required": "",
        "source_chat_ids": [, , ...]
}

## IMPORTANT REQUIREMENTS
1. **Comprehensive coverage**: Summary should integrate all key information from 8-10 bullet points
2. **Sophisticated coherence**: Create complex narrative with multiple threads and advanced logical structure
3. **Advanced multi-session synthesis**: Combine information from 4+ conversation sessions
4. **Strategic condensation**: Include extensive important details while maintaining sophisticated narrative
        structure
5. **Complex question phrasing**: Question should request comprehensive, sophisticated summaries

Generate ONE advanced summarization question that tests the LLM's ability to synthesize 8-10 bullet points
        into a sophisticated, comprehensive summary.

NOTE: Only output the JSON object without any explanation before or after.
```

Listing 18: Summarization probing question generation prompt

```
This is a plan that contains detailed bullet points about a topic. This plan is used to generate realistic
        chat conversations between a user and an AI assistant, which are then used to evaluate the long-term
        memory capabilities of LLMs.
Your task is to analyze this plan and select bullet points that would be most effective for testing
        instruction following abilities when incorporated into chat conversations.
Analyze this plan and identify bullet points that contain user instructions ideal for testing whether the LLM
        remembers and follows user-given instructions.

## INPUT DATA
- **PLAN**: <plan>

Focus on bullet points with:
- **User Instruction** category/label
- **Explicit instruction statements**: "Always", "Never", "When I ask about X, do Y"
- **Behavioral directives**: How the AI should respond or behave
- **Format instructions**: Specific response formats or structures requested
- **Content instructions**: What to include or exclude in responses
- **Process instructions**: How to handle specific types of requests

Look specifically for bullet points labeled as "User Instruction" that contain:
- Clear directive language ("Always provide", "Never include", "When I ask")
- Specific behavioral expectations for the AI assistant
- Conditional instructions ("When I ask about X, do Y")
- Response formatting requirements
- Content inclusion/exclusion rules

Return your analysis in this exact JSON format:
[{"capability": "instruction_following", "batch_numbers": 1,"bullet_numbers": 32,
        "bullet_points": "User Instruction: Always provide detailed cost breakdowns when I ask about budget
                estimates."}
]

Important formatting notes:
- The "batch_numbers" and "bullet_numbers" correspond to each other positionally
- "1" and "32" means: Batch 1 Bullet 32
- Include the full bullet point text as it appears in the plan
- Focus specifically on "User Instruction" labeled bullet points

Select all bullet points labeled as "User Instruction" from each batch (approximately 10 total).

NOTE: Only output the list without any explanation before or after the list.
```

Listing 19: Instruction following probing question generation prompt

```
You are an expert evaluator tasked with judging whether the LLM's response demonstrates compliance with the
        specified RUBRIC CRITERION.

## EVALUATION INPUTS
- QUESTION (what the user asked): <question>
- RUBRIC CRITERION (what to check): <rubric_item>
- RESPONSE TO EVALUATE: <llm_response>

## EVALUATION RUBRIC:
The rubric defines a specific requirement, constraint, or expected behavior that the LLM response should
        demonstrate.

**IMPORTANT**: Pay careful attention to whether the rubric specifies:
- **Positive requirements** (things the response SHOULD include/do)
- **Negative constraints** (things the response SHOULD NOT include/do, often indicated by "no", "not", "avoid
        ", "absent")

## RESPONSIVENESS REQUIREMENT (anchored to the QUESTION)
A compliant response must be **on-topic with respect to the QUESTION** and attempt to answer it.
- If the response does not address the QUESTION, score **0.0** and stop.
- For negative constraints, both must hold: (a) the response is responsive to the QUESTION, and (b) the
        prohibited element is absent.

## SEMANTIC TOLERANCE RULES:
Judge by meaning, not exact wording.
- Accept **paraphrases** and **synonyms** that preserve intent.
- **Case/punctuation/whitespace** differences must be ignored.
- **Numbers/currencies/dates** may appear in equivalent forms (e.g., '$68,000', '68K', '68,000 USD', or 'sixty
        -eight thousand dollars'). Treat them as equal when numerically equivalent.
- If the rubric expects a number or duration, prefer **normalized comparison** (extract and compare values)
        over string matching.

## STYLE NEUTRALITY (prevents style contamination):
Ignore tone, politeness, length, and flourish unless the rubric explicitly requires a format/structure (e.g.,
        'itemized list', 'no citations', 'one sentence').
- Do **not** penalize hedging, voice, or verbosity if content satisfies the rubric.
- Only evaluate format when the rubric **explicitly** mandates it.

## SCORING SCALE:
- **1.0 (Complete Compliance)**: Fully complies with the rubric criterion.
  - Positive: required element present, accurate, properly executed (allowing semantic equivalents).
  - Negative: prohibited element **absent** AND response is **responsive**.

- **0.5 (Partial Compliance)**: Partially complies.
  - Positive: element present but minor inaccuracies/incomplete execution.
  - Negative: generally responsive and mostly avoids the prohibited element but with minor/edge violations.

- **0.0 (No Compliance)**: Fails to comply.
  - Positive: required element missing or incorrect.
  - Negative: prohibited element present **or** response is non-responsive/evasive even if the element is
        absent.

## EVALUATION INSTRUCTIONS:
1. **Understand the Requirement**: Determine if the rubric is asking for something to be present (positive) or
        absent (negative/constraint).

2. **Parse Compound Statements**: If the rubric contains multiple elements connected by "and" or commas,
        evaluate whether:
  - **All elements** must be present for full compliance (1.0)
  - **Some elements** present indicates partial compliance (0.5)
  - **No elements** present indicates no compliance (0.0)

3. **Check Compliance**:
  - For positive requirements: Look for the presence and quality of the required element
  - For negative constraints: Look for the absence of the prohibited element

4. **Assign Score**: Based on compliance with the specific rubric criterion according to the scoring scale
        above.

5. **Provide Reasoning**: Explain whether the rubric criterion was satisfied and justify the score.

## OUTPUT FORMAT:
Return your evaluation in JSON format with two fields:

{
    "score": [your score: 1.0, 0.5, or 0.0],
    "reason": "[detailed explanation of whether the rubric criterion was satisfied and why this justified the
        assigned score]"
}

NOTE: ONLY output the json object, without any explanation before or after that
```

Listing 20: Rubric scoring for nugget satisfaction prompt

```
You are a binary classifier.
If the TWO snippets describe the SAME event/fact, reply **YES**
Otherwise reply **NO**. No extra words.
DO NOT provide any explanation.

First snippet: {first_paragraph}
```

```
Second snippet: {second_paragraph}
```

Listing 21: Fact equivalence detection prompt

```
I want {chat_number} chat titles, themes, and subtopics for the category {chat_category}, in the format below:

{"id": 1,
"category": "Trip Planning",
"title": "Designing a Year-Long Round-the-World Itinerary on a Shoestring",
"theme": "Sequencing flights, overland legs, and visas across five continents in 12 months",
"subtopics": ["Round-the-world tickets", "Back-to-back visa rules", "Seasonal climate mapping", "Open-jaw
    routing", "Long-term travel insurance", "Budget forecasting", "Digital-nomad logistics"]
}

NOTE: Generate the most common ones.
```

Listing 22: Chat titles generation prompt

```
You are a conversation framework specialist tasked with identifying the most relevant section label categories
      for a specific chat scenario.

INPUT PARAMETERS:
DOMAIN: {}
TITLE: {}
THEME: {}
SUBTOPICS: {}

CORE OBJECTIVE:
Analyze the given DOMAIN, TITLE, THEME, SUBTOPICS to determine which section label categories would be most
    relevant and natural for this specific scenario.
Generate 15-20 section label categories that best fit this particular context.

LABEL CATEGORIES SELECTION:
Here are some examples:

**UNIVERSAL CATEGORIES (Always Include 2-3):**
- **Character & Relationship Labels** (relationships are always relevant)
- **Personal & Emotional Labels** (human element always present)
- **Decision & Change Labels** (conversations involve decisions)

**TOPIC-SPECIFIC CATEGORIES (Select 9-13 based on relevance):**

**Planning & Logistics Labels** - Use when scenario involves:
- Travel, events, projects, moves, construction, organizing
- Resource management, scheduling, coordination
- Physical planning or systematic approaches

[More examples]

ELECTION CRITERIA:
**Must Include If Relevant:**
- Categories directly related to the main topic
- Categories that would naturally generate diverse conversations
- Categories that allow for progression and development over time
- Categories that create authentic human concerns and interests

**Avoid Including:**
- Categories that don't naturally fit the scenario
- Too many similar categories that would overlap
- Categories that wouldn't generate meaningful conversation
- Generic categories that don't add specific value

OUTPUT FORMAT:
Generate exactly 15-20 section label categories in this format:

**[Category Name] Labels:**
- Brief explanation of why this category is relevant to the DOMAIN/TITLE/THEME/SUBTOPICS
- 3-5 specific label examples that would be used within this category

Example:
**Planning & Logistics Labels:**
- Essential for travel scenarios involving coordination, scheduling, and resource management
- Budget Planning, Transportation Strategy, Accommodation Research, Itinerary Adjustment, Packing Organization

QUALITY STANDARDS:
Each selected category must:
- Be directly relevant to the specific DOMAIN, TITLE and THEME
- Generate natural, varied conversation opportunities
- Allow for progression and development across multiple batches
- Feel authentic to what a real person would discuss in this scenario
- Provide enough depth for 15-20 bullet points across the conversation
- Just ouput the labels, without explanation at the first

Focus on categories that would create the most natural, engaging, and realistic chat conversations for this
    specific scenario.
```

Listing 23: Narrative generation prompt

```
You are a long-form narrative planning specialist creating a COHERENT STORY PLANSET for natural conversational
        flow.
Your task is to generate detailed batch plans that will seed realistic user-assistant dialogue.

## INPUT DATA
- **DOMAIN:** <domain>
- **TITLE:** <title>
- **THEME:** <theme>
- **SUBTOPICS:** <subtopics>
- **TIMELINE:** <timeline>
- **NUM_BATCHES:** <num_batches> batches
- **LABELS:** <provided_labels>
- **USER PROFILE:** <user_profile>
- **USER RELATIONSHIPS:** <user_relationships>

=============== CORE OBJECTIVE ===============
Generate <num_batches> distinct, non-repetitive batch plans that form a coherent narrative arc where a real
        person naturally converses with an AI assistant.
Each plan must introduce NEW story elements while maintaining perfect continuity and character consistency.

=============== CRITICAL DETAIL REQUIREMENTS ===============
**MANDATORY SPECIFIC DETAILS:**
Every batch MUST include numerous concrete details that enable factual answers:

**Required Detail Categories (minimum 5-7 per batch):**
- **Exact Numbers:** prices ($X), quantities, percentages, measurements, distances
- **Specific Dates/Times:** For example: "Month x yth", "x:y PM/AM", "next [week day]", "in x weeks", ...
- **Named Locations:** restaurants, stores, streets, buildings, parks, venues
- **Brand/Product Names:** specific items, services, companies, tools, software
- **Yes/No Situations:** decisions made, preferences stated, conflicts resolved
- **Event Outcomes:** what happened, who won/lost, what was chosen/rejected
- **Specific Preferences:** favorite foods, colors, activities, music, books
- **Quantifiable Results:** test scores, rankings, ratings, completion times

**Detail Distribution Rules:**
- Each bullet must contain AT LEAST one verifiable detail
- Avoid vague statements like "discussed options" - specify WHAT options
- Instead of "considering choices" use "choosing between X, Y, and Z"

=============== STRUCTURE REQUIREMENTS ===============
**1. OUTPUT FORMAT:**
- Generate exactly <num_batches> plans
- Format: `BATCH X PLAN` headers
- Each plan contains exactly <num_bullets> bullets
- Each bullet: "• **[LABEL CATEGORY]:[LABEL DESCRIPTION]:** [content]" (<=25 words)
- NOTE: Each label consists of category and description. Use both for each bullet point.
- Use only the provided LABELS - no custom categories
- CRITICAL: Add one time anchor bulletpoint at the begining of each batch with this format: Month Day, Year.
**NOTE**: Time anchor must correlated with other dates in the batch and the time anchors among batches should
        be increasing and time anchor in each batch should be before the dates mentioned in the batches.

**2. STORY PROGRESSION ARCHITECTURE:**
**BATCH 1 (Story Foundation):**
- First bullet MUST be: "• **Time Anchor:**"
- Second bullet MUST be: "• **Personal Introduction:**" [Must be from user language (I ...)]
- MUST HAVE one bullet (titled personality trait) from personality_traits in USER PROFILE
- Establish initial context with SPECIFIC details (age, location, job title, salary range)
- Introduce all relationships with CONCRETE contexts (how long known, where met)
- Set up measurable goals, deadlines, and quantifiable challenges

**BATCHES 2-<num_batches> (Story Evolution):**
- Reference user as "I/my/me" (never repeat the full name)
- Each batch advances the timeline chronologically
- Build upon ALL previously established elements
- Show MEASURABLE progression (promotions, relationship milestones, achievement metrics)

**3. RELATIONSHIP CONTINUITY SYSTEM:** [Other details if domain is Coding or Math]
**Relationship Evolution Mandate:**
Every relationship mention MUST include specific interaction details:

**Evolution Stages with Required Details:**
- **Introduction:** "Met [Name] at [specific place] on [date/time]"
- **Development:** "[Name] suggested [specific action] which resulted in [outcome]"
- **Deepening:** "[Name] revealed [specific information] during [specific event]"
- **Maturation:** "After [X months/years], [Name] and I [specific change]"

**Interaction Variety (rotate - never repeat within batches):**
- Collaborative, Supportive, Conflictual, Social, Professional, Personal, Transactional, Serendipitous

**Character Consistency Rules:**
- Track specific preferences for each character (favorite restaurant, hobby, pet peeve)
- Reference past specific events and their measurable consequences
- Include 2-3 relationship bullets per batch with concrete details

=============== CONFLICT & RESOLUTION TRACKING ===============
**Conflict Elements:**
Each batch must include situations with:
- **Binary Decisions:**
- **Measurable Outcomes:**
- **Specific Disagreements:**

**Conflict Types to Rotate:**
```

```
- Financial decisions with specific amounts
- Time management with exact deadlines
- Relationship boundaries with specific incidents
- Professional choices with concrete options
- Personal values with specific scenarios

=============== ANTI-REPETITION VERIFICATION ===============
**Before writing ANY bullet, verify:**
- Have I included at least one specific, verifiable detail?
- Can this generate a question with a one-word answer?
- Does this show MEASURABLE progression from previous mentions?
- Are the numbers, dates, names, and locations specific?
- Is this a NEW piece of information with NEW details?

=============== CONTENT DISTRIBUTION STRATEGY ===============
**Per Batch Requirements:**
- 2-3 bullets: Relationship developments with specific incidents
- 2-4 bullets: Current situation with measurable metrics
- 1 bullet: Exact temporal anchor (specific date/time)
- 5-7 bullets: Events with verifiable outcomes
- 3-4 bullets: Decisions/preferences with specific choices
- 1 bullet: **Preference Statement**: implicitly showing user preferences
- Rest: Using remaining labels with concrete details

=============== SPECIAL BULLET REQUIREMENTS ===============
**1. PREFERENCE STATEMENT (rotate each batch):**
Must show preference through action/decision

**Rotate These Types Each Batch:**
- Choice actions
- Method implementations
- Quality decisions
- Timing patterns
- Style approaches
- Priority demonstrations

**Story Progression Patterns:**
- **Early Batches (1-3):** Establish baselines (current salary, relationship status, living situation)
- **Middle Batches:** Track changes from baselines with specific metrics
- **Later Batches:** Show cumulative results with before/after comparisons

=============== QUALITY STANDARDS ===============
**Specificity Checklist:**
- Every person has a full name and defined relationship
- Every event has a date, time, or specific temporal reference
- Every location has a name or address
- Every decision has concrete options with specific details
- Every outcome is measurable or verifiable

**Narrative Depth:**
- Include prices, percentages, distances, durations
- Show cause-and-effect with specific triggers and results
- Maintain factual consistency (don't change established numbers/dates)
- Reference past specific events by name and date

=============== EXECUTION NOTES ===============
- Prioritize concrete details over abstract descriptions
- Every bullet should enable at least 2-3 factual questions
- Include cultural, financial, and geographic specificity
- Ensure details are realistic and internally consistent
- End immediately after `BATCH <num_batches> PLAN`

Begin generation now.
```

Listing 24: General domain conversation plan generation prompt

```
=========== CRITICAL DETAIL REQUIREMENTS ===========
**MANDATORY SPECIFIC DETAILS:**
Every batch MUST include numerous concrete, verifiable technical details that enable single-word or short
     factual answers:

**Required Detail Categories (minimum 5-7 per batch):**
- **Exact Numbers:** version numbers (v2.3.1), port numbers (3000), response times (250ms), file sizes (2.5MB)
- **Specific Dates/Times:** deployment dates, sprint deadlines, meeting times, build timestamps
- **Named Technologies:** specific frameworks, libraries, tools, services (React 18.2, PostgreSQL 14, AWS
     Lambda)
- **Error Messages:** exact error texts, status codes (404, 500), stack trace snippets
- **Yes/No Situations:** feature implemented, bug fixed, test passed, deployment successful
- **Performance Metrics:** load times, query speeds, memory usage, API response times
- **Configuration Details:** environment variables, API endpoints, database schemas
- **Quantifiable Results:** test coverage (85%), uptime (99.9%), user count (1,000+), bug count

**Detail Distribution Rules:**
- Each bullet must contain AT LEAST one verifiable technical detail
- Avoid vague statements like "worked on feature" - specify WHAT feature and HOW
- Replace "had a bug" with "encountered 'undefined is not a function' error in UserAuth.js line 42"
- Instead of "improved performance" use "reduced API response time from 800ms to 200ms"

=========== STRUCTURE REQUIREMENTS ===========
TECHNICAL CONTINUITY SYSTEM:**
**Development Phase Evolution:**
```

```
Every technical element MUST show progression from previous batches:

**Natural Development Progression Examples:**
- **Planning Phase:** "I need to design the authentication system..."
[More examples]

**Technical Complexity Progression:**
- Early batches: Basic implementation, simple features
- Middle batches: Integration challenges, debugging complex issues
- Later batches: Performance optimization, advanced features, production concerns

=========== CONFLICT & RESOLUTION TRACKING ===========
**Mandatory Technical Conflict Elements:**
Each batch must include at least 2-3 technical challenges with:
- **Clear Stakes:** what's at risk (deployment deadline, performance SLA, budget constraint)
- **Binary Decisions:** chose Framework A over B, implemented Solution X vs Y, fixed vs workaround
- **Measurable Outcomes:** reduced latency by Xms, saved $X in hosting, improved performance by X%
- **Specific Trade-offs:** what was sacrificed for what gain (memory for speed, complexity for features)

**Technical Conflict Types to Rotate:**
- Performance bottlenecks with specific metrics
- Architecture decisions with concrete alternatives
- Integration challenges with external systems
- Security vulnerabilities with severity levels
- Scalability issues with user load numbers
- Technical debt vs new features

=========== CONTENT DISTRIBUTION STRATEGY ===========
**Per Batch Requirements:**
- 2-3 bullets: Technical implementation details with specific code elements
- 2-4 bullets: Current development status with measurable metrics
- 1 bullet: Exact temporal anchor (specific date/time)
- 5-7 bullets: Development activities with verifiable outcomes
- 3-4 bullets: Technical decisions with specific alternatives considered
- 1 bullet: **Preference Statement**: implicitly showing developer preferences
- Rest: Using remaining labels with concrete technical details

**Adaptive Batch Planning:**
Each batch should organically focus on what makes sense for that development phase:

**Implementation-Heavy Batch:**
- Multiple implementation requests
- Architecture decisions
- Code structure planning
- Framework/library selection

[More examples]

=========== NATURAL CODING CONVERSATION FLOW ===========
Each bullet should represent realistic developer-AI interactions:

**Implementation Requests:**
[Examples]

**Debugging Help:**
[Examples]

**Code Review/Optimization:**
[Examples]

=========== EXECUTION NOTES ===========
- Use plain, technical language throughout
- Include realistic technical specificity: version numbers, error messages, configuration details
- Make every bullet contribute to the overarching development story
- Ensure uniform technical detail quality across ALL batches
- Vary batch focus organically based on development phase (implementation vs debugging vs optimization)
- Prioritize concrete technical details over abstract descriptions
- Every bullet should enable at least 2-3 factual technical questions
- End immediately after `BATCH <num_batches> PLAN`
```

Listing 25: Coding domain conversation plan generation prompt

```
=========== CRITICAL DETAIL REQUIREMENTS ===========
**MANDATORY SPECIFIC DETAILS:**
Every batch MUST include numerous concrete, verifiable mathematical details that enable single-word or short
    factual answers:

**Required Detail Categories (minimum 5-7 per batch):**
- **Exact Numbers:** specific values (x = 3.14), coefficients ($2x^2 + 5x - 3$), dimensions ($5\times7$ matrix
    )
- **Specific Problems:** complete equations ($x^2 - 4x + 3 = 0$), specific integrals ($\int_0^5 (2x+1)\,dx$)
- **Named Concepts:** theorem names (Pythagorean Theorem), method names (Gaussian elimination), formulas (
    quadratic formula)
- **Calculation Results:** exact answers ($x = 4$), decimal results ($\pi \approx 3.14159$), fractions ($\
    tfrac{3}{4}$)
- **Yes/No Situations:** problem solved correctly, method applicable, theorem satisfied, solution exists
- **Score/Grade Metrics:** test scores (85%), homework grades (18/20), quiz results (9/10 correct)
- **Time/Duration:** study hours (3 hours), problem completion time (15 minutes), exam duration (2 hours)
- **Mathematical Properties:** function characteristics (continuous, differentiable), matrix properties (
    invertible, symmetric)
```

```
**Detail Distribution Rules:**
- Each bullet must contain AT LEAST one verifiable mathematical detail
- Avoid vague statements like "worked on problems" - specify WHICH problems and results
- Replace "studied math" with "completed 5 quadratic equation problems, solved 4 correctly"
- Instead of "improved understanding" use "increased quiz score from 70% to 85%"

=============== STRUCTURE REQUIREMENTS ===============
**3. MATHEMATICAL CONTINUITY SYSTEM:**
**Learning Phase Evolution:**
Every mathematical element MUST show progression from previous batches:

**Natural Learning Progression Examples:**
- **Conceptual Phase:** "I need to understand what derivatives mean..."
[More examples]

=============== CONFLICT & RESOLUTION TRACKING ===============
**Mandatory Mathematical Conflict Elements:**
Each batch must include at least 2-3 mathematical challenges with:
- **Clear Stakes:** what's at risk (exam grade, assignment deadline, course prerequisite)
- **Binary Decisions:** chose Method A over B, applied Theorem X vs Y, used algebraic vs geometric approach
- **Measurable Outcomes:** improved accuracy by X%, reduced solution time by Y minutes, raised grade from B to
      A
- **Specific Struggles:** which step caused confusion, what concept was misunderstood, where calculation went
      wrong

**Mathematical Conflict Types to Rotate:**
- Conceptual misunderstandings with specific confusion points
- Calculation errors with exact mistake locations
- Method selection dilemmas with pros/cons
- Time pressure challenges with specific deadlines
- Prerequisite knowledge gaps with missing concepts
- Application difficulties with real-world connections

=============== CONTENT DISTRIBUTION STRATEGY ===============
**Per Batch Requirements:**
- 2-3 bullets: Problem-solving activities with specific equations/solutions
- 1-2 bullets: Current learning status with measurable metrics
- 1 bullet: Exact temporal anchor (specific date/time)
- 4-6 bullets: Mathematical activities with verifiable outcomes
- 2-3 bullets: Learning decisions with specific alternatives considered
- 1 bullet: **Preference Statement**: implicitly showing learning preferences
- Rest: Using remaining labels with concrete mathematical details

**Adaptive Batch Planning:**
Each batch should organically focus on what makes sense for that learning phase:

[More examples]

=============== NATURAL MATH CONVERSATION FLOW ===============
Each bullet should represent realistic user-AI interactions:

**Problem-Solving Requests:**
[examples]

**Concept Clarification:**
[examples]

**Solution Verification:**
[examples]

**Method Explanation:**
[examples]

=============== QUALITY STANDARDS ===============

**Chronological Consistency:**
- Batch 1 = learning beginning/foundation phase
- Batch <num_batches> = evolved understanding with clear mathematical progression
- Each batch logically follows the previous learning timeline

**Mathematical Authenticity:**
- Include specific mathematical details: equation types, theorem names, calculation methods
[More examples]

**User Authenticity:**
- Keep user personality consistent with provided profile
[More examples]

**Learning Realism:**
- Follow realistic mathematical learning patterns
[More examples]

**Specificity Checklist:**
- Every equation has specific coefficients and variables
[More examples]

=============== EXECUTION NOTES ===============

- Use plain, mathematical language throughout
- Include realistic mathematical specificity: complete equations, exact values, specific theorems
- Make every bullet contribute to the overarching mathematical story
- Ensure uniform mathematical detail quality across ALL batches
```

```
- Vary batch focus organically based on learning phase (understanding vs solving vs applying)
- Prioritize concrete mathematical details over abstract descriptions
- Every bullet should enable at least 2-3 factual mathematical questions
- End immediately after `BATCH <num_batches> PLAN`

Begin generation now.
```

Listing 26: Math domain conversation plan generation prompt

```
You are a specialized editor that adds three specific test bullets to existing batch plans for synthetic
    conversation generation.

## INPUT & TASK
- **PLAN:** <plan>
- For EACH batch: Keep ALL <num_bullets> original bullets unchanged, ADD exactly 3 bullets at positions <
    num_bullets>+1, <num_bullets>+2, <num_bullets>+3

## MANDATORY OUTPUT STRUCTURE
Each batch MUST have EXACTLY <num_bullets>+3 bullets:
- Bullets 1-<num_bullets>: Original bullets (unchanged)
- Bullet <num_bullets>+1: Information Update
- Bullet <num_bullets>+2: User Instruction
- Bullet <num_bullets>+3: Logical Contradiction

## THE THREE SPECIAL BULLETS
### 1. INFORMATION UPDATE (Bullet <num_bullets>+1)
**Format:** `• **Information Update:** [Natural narrative containing update]`

**MANDATORY PRE-CHECK:**
1. SCAN bullets 1-<num_bullets>-2 for EXPLICIT numerical/measurable data
2. IDENTIFY exact number, time, date, or measurement
3. VERIFY value is clearly stated in original text
4. ONLY THEN create update changing that exact value

EXAMPLES:
[Some examples]

**CRITICAL:** Make update IMPLICIT - embed new value in natural narrative, don't state "X is now Y"

**Update Categories (rotate through all):**
1. Numerical shifts (prices, quantities, measurements)
2. Status changes (employment, relationships, health)
3. Location changes (addresses, venues, destinations)
4. Relationship progressions (social connections)
[Some other categories]

**VERIFICATION:** STOP if no matching fact exists in bullets 1-<num_bullets>-2.

### 2. USER INSTRUCTION (Bullet <num_bullets>+2)
**Format:** `• **User Instruction:** Always [action] when I ask about [condition]`

**MANDATORY FORMAT:** Must include "when I ask about" - this makes it testable.

**Instruction Types (rotate through all):**
1. Output formatting rules
2. Content restrictions
3. Personal preferences
4. Conditional responses
5. Time-based rules
[Some other categories]

**EXAMPLES:**
[Some examples]

### 3. LOGICAL CONTRADICTION (Bullet <num_bullets>+3)
**Format:** `• **Logical Contradiction:** [Contradicting fact only]`

**CRITICAL PRE-CHECK:**
1. SCAN bullets 1-<num_bullets>-2 for COMPLETED ACTIONS or PERMANENT STATES:
   - Past tense actions: "visited", "ate", "traveled", "lived"
   - Permanent conditions: "born in", "raised as", "died in"
   - Absolute statements: "never did X", "always was Y"
2. FIND exactly ONE target fact to contradict
3. IF NO COMPLETED ACTIONS EXIST: CREATE setup bullet with completed action, insert between bullets 5-<
    num_bullets>-2, THEN contradict in bullet <num_bullets>+4

FORBIDDEN WORDS/PHRASES (NEVER USE):
"Before this batch", "In this batch", [Other examples]

**RULE:** Only contradict COMPLETED ACTIONS, never plans/intentions.

TEMPORAL QUALIFIER PROBLEM:
[Some examples]

FORBIDDEN WORDS/PHRASES (NEVER USE):
"Before this batch", "In this batch", "Previously",
[Other examples]

**CRITICAL:** Contradiction must make the **same original event/fact IMPOSSIBLE**, not describe a different
    event with different outcomes.
```

```
**Contradiction Types (use variety):**
1. Age/Time Reversal: Age going backward
2. Death Resurrection: Dead people doing activities
3. Never-Statement Violations: Contradicting "never" claims
4. Location Impossibilities: Being in two places simultaneously
5. Only-Statement Conflicts: Contradicting exclusivity

**VERIFICATION:**
Original is COMPLETED ACTION (past tense)?
Contradiction makes original IMPOSSIBLE, not just different?
Reads like normal, natural statement with NO hint words?
Avoided ALL forbidden words that suggest conflict?

## CRITICAL VERIFICATION STEPS
**Information Update:**
- Can you point to EXACT number/time/date from bullets 1-<num_bullets>-2?
- Changing ONLY that specific value?

**Logical Contradiction:**
- Can you point to EXACT bullet (1-<num_bullets>-2) being contradicted?
- Original fact is COMPLETED ACTION, not plan?
- Contradiction is IMPOSSIBLE, not just different?
- NOT using FORBIDDEN WORDS/PHRASES mentioned in LOGICAL CONTRADICTION section

## COMMON ERRORS TO AVOID
Don't place special bullets anywhere except <num_bullets>+1, <num_bullets>+2, <num_bullets>+3
Don't modify original <num_bullets> bullets
Don't skip any of the three special bullets
Don't use other labels for special bullets

## OUTPUT FORMAT
Return COMPLETE plan where each batch has ALL original bullets unchanged + exactly 3 additional bullets at the
    end. When setup fact is created, insert between bullets 5-<num_bullets>-2, renumber, and add special
    bullets as <num_bullets>+2, <num_bullets>+3, <num_bullets>+4.

Begin processing the plan now.
```

Listing 27: Adding special bulletpoints to conversation plan prompt

```
You are a narrative coherence specialist creating CHRONOLOGICALLY SEQUENCED topic clusters for realistic
    conversational AI dataset generation. Your task is to generate 10 interconnected topics that form a
    natural life progression.

## INPUT DATA
- **SEED TOPIC**: <seed_topic>
- **SEED THEME**: <seed_theme>
- **SEED SUBTOPICS**: <seed_subtopics>
- **USER PROFILE**: <user_profile>
- **TIMELINE**: <timeline>

## CORE OBJECTIVE
Generate a JSON object containing 10 topics (including the provided seed topic as Topic 0) that form a
    CHRONOLOGICALLY COHERENT narrative where each topic naturally follows the previous one in realistic time
     progression.

## CRITICAL REQUIREMENTS
### 1. CHRONOLOGICAL COHERENCE & TOPIC INDEPENDENCE
**TOPIC PROGRESSION RULES:**
- **Topic 0**: Use the provided seed topic EXACTLY as given
- **Topics 1-9**: Each must be a COMPLETELY DIFFERENT life domain/category
- **NO EXTENDED NARRATIVES**: Topics 1-9 should NOT continue the seed topic's story
- **LIFE PROGRESSION**: Each topic represents what naturally happens AFTER completing the previous life
    experience

**TOPIC INDEPENDENCE MANDATE:**
- Each topic must address a DIFFERENT life area (career, relationships, health, education, finances, etc.)
- Topics should show how one life experience leads to growth in OTHER areas
- NO topic should be "Part 2" of a previous topic

### 2. NATURAL LIFE FLOW REQUIREMENTS
**CAUSAL RELATIONSHIPS WITHOUT CONTINUATION:**
- Topic N+1 is INFLUENCED BY Topic N but addresses a DIFFERENT life domain
- Show how growth in one area catalyzes change in another area
- Example: Travel experience (Topic 0) -> Career reassessment (Topic 1) -> Relationship priorities (Topic 2)
[Examples]

### 3. USER PROFILE ALIGNMENT
- **Demographic Consistency**: All topics must align with user's age, education, career level, and life stage
- **Financial Realism**: Topics must reflect user's actual financial capacity and constraints
- **Geographic Logic**: Topics must consider user's location and mobility constraints
- **Value Alignment**: Topics must reflect user's stated priorities, interests, and life goals

### 4. TOPIC BREADTH REQUIREMENTS
Each topic must include a realistic timeline that:
- **Sequential Timing**: Topics must not overlap and should follow logical temporal progression
- **Duration Realism**: Each topic should span 1-2 months for authentic decision-making and implementation
- **Natural Gaps**: Include realistic time gaps between major life transitions
- **Seasonal Considerations**: Account for natural timing (job searches, moving seasons, academic calendars)
- **Timeline Format**: Use "Month X, Year Y - Month X', Year Y'" format
Each topic must be sufficiently BROAD to generate 2000+ authentic conversations by including:
```

```
- **Multiple Decision Points**: 15-20 major decisions per topic
- **Complex Subtopics**: 9-10 substantial subtopics that each require extensive discussion
- **Ongoing Processes**: Topics involving multi-month planning, execution, and adjustment phases
- **Cross-Domain Impact**: Topics affecting multiple life areas

### 5. NARRATIVE REALISM
- **Natural Timing**: Realistic time gaps between major life decisions
- **Emotional Progression**: Topics should reflect natural emotional and psychological development
- **Practical Constraints**: Topics must acknowledge real-world limitations (money, time, responsibilities)

## OUTPUT FORMAT REQUIREMENTS
Generate a single JSON object with this EXACT structure:
```json
{"topics": [{"id": 0,"category": "[Provided Category]","title": "[Provided Title]", "theme": "[Provided Theme
    ]","subtopics": [/* Provided Subtopics Array */],"timeline": "[Start Month, Year - End Month, Year]"},
  {"id": 1,"category": "[New Category]","title": "[Descriptive Title]","theme": "[Character-focused theme
      describing the challenge/opportunity]", "subtopics": ["[Subtopic 1]", "[Subtopic 2]", "[Subtopic
      3]", "[Subtopic 4]", "[Subtopic 5]", "[Subtopic 6]", "[Subtopic 7]", "[Subtopic 8]", "[Subtopic 9]",
      "[Subtopic 10]"],"timeline": "[Start Month, Year - End Month, Year]"},
  // ... topics 2-9 following same structure]}
```

## CRITICAL TIMELINE REQUIREMENTS
### MANDATORY NON-OVERLAPPING TIMELINE RULES
**ABSOLUTE RULE**: Each topic's timeline MUST start AT LEAST one month AFTER the previous topic ends.
**TIMELINE CALCULATION PROTOCOL:**
1. Topic 0: Uses provided timeline exactly
2. Topic N+1 start = Topic N end + AT LEAST 1 month gap
3. NO overlapping months between any topics
4. Each topic duration: 1-2 months
[Examples]

**TIMELINE VERIFICATION STEPS:**
Before finalizing each topic:
1. Identify previous topic's END month
2. Add AT LEAST 1 month to get earliest possible START
3. Verify NO month appears in multiple topics
4. Confirm realistic gaps for life transitions

**GAP JUSTIFICATION:**
The 1+ month gaps represent:
- Processing and integration time after major experiences
- Natural life rhythms and decision-making periods
- Realistic pacing of significant life changes
- Time for consequences of previous decisions to manifest

## TOPIC PROGRESSION GUIDELINES
### Phase 1: Post-Seed Topic Reality (Topics 1-2)
- **Topic 1**: How the seed topic experience changes perspective on ANOTHER life area
- **Topic 2**: Ripple effects creating needs in YET ANOTHER domain

### Phase 2: Multi-Domain Growth (Topics 3-5)
- **Topics 3-5**: Leveraging cumulative growth to address diverse life challenges

### Phase 3: Integration Across Life (Topics 6-8)
- **Topics 6-8**: Synthesizing learnings to optimize different life areas

### Phase 4: Holistic Vision (Topic 9)
- **Topic 9**: Long-term life design incorporating all previous growth

**CRITICAL QUESTION FOR EACH TOPIC:**
"After completing [previous topic], what DIFFERENT area of life would this person naturally need to address
    next?"

## QUALITY VALIDATION CHECKLIST
[Examples]

## EXAMPLE PROGRESSION LOGIC
[Example]

## FORBIDDEN ELEMENTS
- **Non-sequential topics**: Topics that could happen in any order
- **Profile contradictions**: Topics that contradict user's established circumstances
- **Unrealistic jumps**: Major life changes without proper foundation/motivation
- **Narrow topics**: Topics that couldn't generate extensive conversation
- **Template responses**: Generic topics that don't reflect unique user circumstances

## EXECUTION NOTES
- Generate all 10 topics in a single coherent response
- Ensure seamless narrative flow from Topic 0 through Topic 9
- Prioritize realism and character consistency over dramatic storylines
- Focus on authentic life progressions that real people experience
- End output immediately after closing the JSON structure

**CRITICAL**: Output your response in JSON format only. Do not include any explanatory text, markdown
    formatting, or additional commentary. Provide only the raw JSON object.

Generate the complete topic cluster now.
```

Listing 28: Ten million sequential seed generation prompt

```
You are creating a CHRONOLOGICAL SUBTOPIC FRAMEWORK that breaks down a main topic into 10 diverse, non-
    repetitive phases with strict timeline boundaries.

## INPUT DATA
- **MAIN TOPIC:** <main_topic>
- **MAIN THEME:** <main_theme>
- **MAIN SUBTOPICS:** <main_subtopics>
- **USER PROFILE:** <user_profile>
- **TOTAL TIMELINE:** <total_timeline>

## TIMELINE EXTRACTION (MANDATORY FIRST)
1. **Extract Core Action**:
    - Duration from main topic (e.g., "x-day doing Y" = x days)
    - Action type (trip/project/course/challenge/etc.)
    - Total timeline span in months

2. **Calculate Key Dates**:
    - MAIN_ACTION_START: When core action begins
    - MAIN_ACTION_END: When core action ends
    - Allocate realistic prep/integration time around core action

## SUBTOPIC GENERATION RULES
### Phase Distribution (10 Topics)
- **Topics 0-2**: PREPARATION (before action starts)
- **Topics 3-6**: CORE ACTION (during main action period)
- **Topics 7-9**: INTEGRATION (after action ends)

### MANDATORY DIVERSITY REQUIREMENTS
**EACH SUBTOPIC MUST BE UNIQUE:**
- No recycling of themes between subtopics
- Each explores DIFFERENT aspects/challenges
- Progressive complexity within each phase
- Distinct focus areas that don't overlap

**PREPARATION DIVERSITY (Topics 0-2):**
- Topic 0: Discovery/Research/Initial Planning
- Topic 1: Decision-Making/Resource Gathering/Skill Building
- Topic 2: Final Preparations/Confirmations/Pre-Launch

**CORE ACTION DIVERSITY (Topics 3-6):**
- Topic 3: Launch/Beginning/Initial Experiences
- Topic 4: Early Challenges/Adaptations/Progress
- Topic 5: Peak Performance/Deep Engagement/Mastery
- Topic 6: Final Push/Completion/Transition

**INTEGRATION DIVERSITY (Topics 7-9):**
- Topic 7: Immediate Reflection/Initial Processing
- Topic 8: Application/Transformation/Sharing
- Topic 9: Long-term Impact/Future Planning/Legacy

### Required Structure Per Subtopic
```json {"id": [0-9],"category": "[Phase name]","title": "[Unique descriptive title - NO REPETITION]","theme":
    "[Distinct challenge/opportunity - MUST BE DIFFERENT]","subtopics": ["10 DIVERSE sub-elements - NO
    OVERLAP with other topics]","timeline": "[Date range]","phase_type": "[preparation/core_action/
    integration]",
  "action_dates": {"main_action_type": "[type]","main_action_starts": "[date]","main_action_ends": "[date]","
    main_action_duration": "[duration]","current_phase_relation": "[before/during/after]"},
  "phase_boundaries": {"can_mention": ["Allowed activities"],"cannot_mention": ["Forbidden activities"],"
    tense_for_main_action": "[future/present/past]"},
  "key_milestones": ["3 unique milestones"],"future_references": ["Setup for continuity"],"continuity_hooks":
    ["Links to next topic"]}

DIVERSITY ENFORCEMENT CHECKLIST
[Examples]

MAIN SUBTOPIC DISTRIBUTION
Spread the provided main_subtopics across topics you generate strategically:

Show evolution: basic -> intermediate -> advanced -> mastery
Different angles in each phase (planning vs doing vs reflecting)

OUTPUT FORMAT
Generate ONLY this JSON structure:
json{"main_topic": "[Input]","main_theme": "[Input]","main_subtopics": ["Input array"],"total_timeline": "[
    Input]",
  "master_timeline": {"timeline_start": "[Date]","timeline_end": "[Date]","main_action_starts": "[Date]","
    main_action_ends": "[Date]","main_action_duration": "[Duration]",
    "preparation_phase": {"start": "[Date]","end": "[Date]","topics": [0, 1, 2]},
    "core_action_phase": {"start": "[Date]","end": "[Date]","duration": "[Duration]","topics": [3, 4, 5, 6]},
    "integration_phase": {"start": "[Date]","end": "[Date]","topics": [7, 8, 9]},
  "subtopics": [/* 10 UNIQUE subtopic objects */]}

CRITICAL:
Each subtopic explores DIFFERENT aspects
NO thematic repetition across topics
Progressive narrative arc
Diverse conversation opportunities

Output ONLY the JSON. No explanations
```

Listing 29: Ten million hierarchical seed generation prompt

```
You are a long-form narrative planning specialist creating a COHERENT STORY PLANSET for natural conversational
        flow. Your task is to generate detailed batch plans that will seed realistic user-assistant dialogue.

## INPUT DATA
- **DOMAIN:** <domain>
- **TITLE:** <title>
- **THEME:** <theme>
- **SUBTOPICS:** <subtopics>
- **TIMELINE:** <timeline>
- **NUM_BATCHES:** <num_batches> batches
- **LABELS:** <provided_labels>
- **USER PROFILE:** <user_profile>
- **CORE RELATIONSHIPS:** <core_relationships>
- **NEW RELATIONSHIPS:** <new_relationships>
- **PREVIOUS PLAN:** <previous_plan>
- **INCLUDE_INTRODUCTION:** <YES/NO>

=============== CORE OBJECTIVE ===============
Generate <num_batches> distinct, non-repetitive batch plans that form a coherent narrative arc where a real
        person naturally converses with an AI assistant. Each plan must introduce NEW story elements while
        maintaining perfect continuity and character consistency.

=============== CRITICAL NARRATIVE PERSPECTIVE ===============
**MANDATORY FIRST-PERSON PERSPECTIVE:** - ALL content must be written from the USER's perspective (first-
        person)
- Use first-person perspective throughout but VARY sentence structures - Natural narrative flow - avoid
        starting every bullet with "I"
- Mix active and passive voice while maintaining first-person perspective

=============== CONTINUITY REQUIREMENTS ===============
**CRITICAL**: If PREVIOUS PLAN is provided, you MUST:
- **Reference Previous Events** **Maintain Core Character Consistency** **Integrate New Relationships** **Show
        Temporal Progression**
**Build Upon Previous Decisions** **Preserve Established Facts** **Continue Relationship Arcs**

=============== STRICT TIMELINE ENFORCEMENT ===============
**CRITICAL TIMELINE PARSING (MANDATORY FIRST STEP):**
Before generating ANY content, you MUST internally calculate timeline boundaries.

**STEP 1: Extract and Write Timeline Boundaries**
**CALCULATE YOUR PARSED DATES:**

**STEP 2: Create Batch Date Assignments**
Divide timeline into <num_batches> segments: Days per batch = TOTAL DAYS / <num_batches>

**ABSOLUTE TIMELINE RULES:** 1. **EVERY date mentioned MUST be between START and END dates** 2. **NO future
        references beyond TIMELINE END** (no "next month" if timeline ends this month)
3. **NO past references before TIMELINE START** 4. **Temporal anchors MUST progress chronologically within
        boundaries** 5. **Final batch MUST conclude naturally before or on END DATE**

**TEMPORAL ANCHOR REQUIREMENTS:**
- First bullet of EACH batch MUST be temporal anchor
- Format: "• **Temporal Anchor:** [Month] [Day], [year], [event description]"
- Each temporal anchor date MUST be within that batch's assigned date range
- Dates must progress: Batch 2's date > Batch 1's date, etc.

**TIMELINE VIOLATION EXAMPLES (FORBIDDEN):**
[Examples]
**PRE-GENERATION CHECKLIST:**
[Examples]

=============== CRITICAL DETAIL REQUIREMENTS ===============
**MANDATORY SPECIFIC DETAILS:**
Every batch MUST include numerous concrete, verifiable details that enable single-word or short factual
        answers:

**Required Detail Categories (minimum 5-7 per batch):**
- **Exact Numbers:** prices ($X), quantities, percentages, measurements, distances
[Some categories examples]

**Detail Distribution Rules:**
- Each bullet must contain AT LEAST one verifiable detail - Avoid vague statements

=============== STRUCTURE REQUIREMENTS ===============
**1. OUTPUT FORMAT:**
- Generate exactly <num_batches> plans
- Format: `BATCH X PLAN` headers
- Each plan contains exactly 30 bullets
- Each bullet: "• **[LABEL CATEGORY]:[LABEL DESCRIPTION]:** [content]" (<=25 words)
- NOTE: Each label consists of category and description. Use both for each bullet point.
- Use only the provided LABELS - no custom categories
- **MANDATORY**: First bullet MUST be Temporal Anchor with the ONLY date reference in the batch

**2. STORY PROGRESSION ARCHITECTURE:**
**IF INCLUDE_INTRODUCTION = YES:**
**BATCH 1 (Story Foundation):**
- First bullet MUST be: "• **Personal Introduction:**" Establish initial context with SPECIFIC details
        Introduce all relationships with CONCRETE contexts Set up measurable goals, deadlines, and quantifiable
        challenges

**IF INCLUDE_INTRODUCTION = NO:**
```

```
**BATCH 1 (Continuation):** NO personal introduction bullets Begin directly with current topic-related content
        Reference established character details from PREVIOUS PLAN

**BATCHES 2-<num_batches> (Story Evolution):**
- Reference user as "I/my/me" (never repeat the full name) Each batch advances the timeline chronologically
      Build upon ALL previously established elements
Show MEASURABLE progression (promotions, relationship milestones, achievement metrics)

**3. RELATIONSHIP CONTINUITY SYSTEM:**
**Core vs. New Relationship Management:**
- **CORE RELATIONSHIPS**: Must remain consistent across all plans - same names, established details, ongoing
      dynamics
- **NEW RELATIONSHIPS**: Introduce naturally based on current topic and life phase **Relationship Integration
      **

**Relationship Evolution Mandate:**
Every relationship mention MUST include specific interaction details:

**Evolution Stages with Required Details:**
[Examples]

=============== CONFLICT & RESOLUTION TRACKING ===============
**Mandatory Conflict Elements:**
Each batch must include at least 2-3 situations with:
- **Clear Stakes:** what's at risk (money amount, deadline, relationship status)
[Examples]

**Conflict Types to Rotate:** Financial decisions with specific amounts Time management with exact deadlines
        Relationship boundaries with specific incidents
Professional choices with concrete options Personal values with specific scenarios

=============== ANTI-REPETITION VERIFICATION ===============
[Examples]

=============== CONTENT DISTRIBUTION STRATEGY ===============
**Per Batch Requirements:**
- 2-3 bullets: Relationship developments with specific incidents (mix of core and new relationships)
- 2-4 bullets: Current situation with measurable metrics
- 1 bullet: Exact temporal anchor (specific date/time)
- 5-7 bullets: Events with verifiable outcomes
- 3-4 bullets: Decisions/preferences with specific choices
- Rest: Using remaining labels with concrete details

**Story Progression Patterns:**
- **Early Batches (1-3):** Establish baselines (current salary, relationship status, living situation) **
        Middle Batches:** Track changes from baselines with specific metrics
**Later Batches:** Show cumulative results with before/after comparisons

=============== NATURAL CONVERSATION FLOW ===============
These plans generate conversations where users seek AI assistance for SPECIFIC situations:
[Example]

=============== QUALITY STANDARDS ===============
**Specificity Checklist:**
- Every person has a full name and defined relationship
[Other exmaples]

**Narrative Depth:**
- Include prices, percentages, distances, durations
- Show cause-and-effect with specific triggers and results
- Maintain factual consistency (don't change established numbers/dates)
- Reference past specific events by name and date

=============== EXECUTION NOTES ===============
- Prioritize concrete details over abstract descriptions
- Every bullet should enable at least 2-3 factual questions
- Include cultural, financial, and geographic specificity
- Ensure details are realistic and internally consistent
- If PREVIOUS PLAN provided, include 3-5 specific references to previous events per batch
- End immediately after `BATCH <num_batches> PLAN`

**FINAL TIMELINE REMINDER:**
- Parse TIMELINE boundaries FIRST
- EVERY date must fall within those boundaries
- NO exceptions to timeline limits
- Verify each batch respects the timeline

Output ONLY the batch plans. No explanations or additional text.

Begin generation now.
```

Listing 30: Ten million sequential conversation plan generation prompt

```
You are a precision narrative architect generating TEMPORALLY COHERENT BATCH PLANS with absolute timeline
      integrity and phase-appropriate content.

## INPUTS
MAIN_TITLE:<main_title> | MAIN_THEME:<main_theme> | TITLE:<title> | THEME:<theme> | TIMELINE:<timeline> |
      NUM_BATCHES:<num_batches> | LABELS:<provided_labels> | USER_PROFILE:<user_profile> | CORE_RELATIONSHIPS
      :<core_relationships> | NEW_RELATIONSHIPS:<new_relationships> | ALL_SUBTOPIC_PLANS:<all_subtopic_plans>
       | PREVIOUS_PLANS_SUMMARY:<previous_plans_summary> | PREVIOUS_PLAN:<previous_plan> |
```

```
        CURRENT_SUBTOPIC_DATA:<current_subtopic_data> | CURRENT_SUBTOPIC_ID:<current_subtopic_id> |
        INCLUDE_INTRODUCTION:<YES/NO>

## OBJECTIVE
Generate <num_batches> distinct, non-repetitive batch plans forming coherent narrative where user naturally
        converses with AI. Each plan introduces NEW elements while maintaining continuity.

## STRUCTURE [MANDATORY]
- Format: `BATCH X PLAN` headers
- Exactly 30 bullets per batch
- Bullet format: "**[LABEL CATEGORY]:[LABEL DESCRIPTION]:** [content]" (<=30 words)
- First bullet ALWAYS: "• **Temporal Anchor:** [Date], [context]"
- NO other dates in batch except temporal anchor
- ALL content in FIRST-PERSON ("I/my/me")
- Vary sentence structures, avoid starting every bullet with "I"

## DETAIL REQUIREMENTS [8-10 per batch minimum]
- Exact Numbers: prices($X), quantities, percentages, measurements
- Specific Dates/Times: "Month x yth", "x:y PM/AM"
[Some other examples]

Replace vague with specific:
[Examples]

## FACT TRACKING SYSTEM [MAINTAIN THROUGHOUT]
Track per batch:
1. Purchases: [Item, Price, Store, Date]
[Examples]

Before EVERY bullet:
- Check if fact exists in registry If similar exists, ADD NEW DIMENSION (consequence/complication/perspective/
        progression)

## PROGRESSION PATTERNS
**Batches 1-3:** Establish baselines, initial decisions, relationship intros **Batches 4-6:** Show
        consequences, complications, deepen relationships
**Batches 7-8:** Unexpected developments, secondary effects, evolution **Batches 9-10:** Long-term impacts,
        synthesis, maturity, future implications

Recurring element progression: 1. First: Basic establishment 2. Second: Add complication 3. Third: Show
        resolution 4. Fourth: Reveal impact 5. Fifth+: FORBIDDEN unless dramatic change

## LABEL ROTATION RULES
Track usage: Label+Focus combination FORBIDDEN across all batches
[Examples]

## RELATIONSHIP RULES
**IF INCLUDE_INTRODUCTION=YES:** Batch 1 first bullet: "• **Personal Introduction:**" Establish context with
        SPECIFICS (age, location, job, salary) Introduce relationships with context (how long known, where met)

**NEW_RELATIONSHIPS first appearance:** Include relationship to user + age + context After introduction, refer
        naturally

Every relationship mention needs specific interaction: Introduction Development Deepening Maturation

## BATCH REQUIREMENTS
- 1 temporal anchor (specific date)
- 2-3 relationship developments
- 3-4 current situation with metrics
- 5-6 events with outcomes
- 4-5 decisions with choices
- Rest: remaining labels with details

## ANTI-REPETITION PROTOCOL
**THREE-PASS REVIEW:** 1. **Fact Uniqueness:** 2. **Information Advancement:** 3. **Cross-Batch:**

[Examples]

## DATE EXTRACTION
Extract from CURRENT_SUBTOPIC_DATA

## TEMPORAL BOUNDARIES
**preparation phase:** [Example]
**core_action phase:** [Example]
**integration phase:** [Example]
**ABSOLUTE:** No dates outside [START_DATE, END_DATE] from TIMELINE

## CONTENT BOUNDARIES
Extract from CURRENT_SUBTOPIC_DATA

**Rules:** ONLY generate from can_mention NEVER generate from cannot_mention ONLY reference current subtopic
        activities Use specified tense for main action

**Phase Content:** **Preparation:** [Example] **Core Action:** [Example] **Integration:** [Example]

## TIMELINE DISTRIBUTION
1. Calculate: Total Days = END - START + 1 2. IF Days >= <num_batches>: Sequential dates 3. IF Days < <
        num_batches>: Group batches per day

**Same-day differentiation:** Time progression (morning->evening) Activity focus shifts Perspective changes
        Depth layers
```

```
## CONTEXT INTEGRATION
1. Review previous plans summary for established facts 2. Continue from previous plan if exists 3. IF
     INCLUDE_INTRODUCTION=YES: Introduce naturally
4. IF NO: Continue without re-introduction 5. Reference prior facts consistently 6. Show progression from
     previous ending

## VALIDATION GATES
[Examples]

## EXECUTION 1. Extract/verify dates 2. Write FIRST-PERSON 3. Date ONLY in anchor 4. Maximum detail density 5.
       Exactly 30 bullets 6. Validate boundaries

Output ONLY batch plans. End after `BATCH <num_batches> PLAN`
```

Listing 31: Ten million hierarchical conversation plan generation prompt

```
You are generating realistic questions that a USER would ask an AI ASSISTANT. Create questions based ONLY on
     the specific details in the current bullet points.

## DOMAIN: <domain>
## TITLE: <title>
## CURRENT FOCUS AREAS (ONLY SOURCE FOR QUESTIONS): <FOCUSED_BULLETS>
## AVOID (ALREADY COVERED): <BATCH_HISTORY>
## CONTEXT REFERENCE (FOR UNDERSTANDING ONLY): <PREVIOUS_SUB_BATCH_PLANS> <PREVIOUS_BATCH_PLANS>

## CRITICAL RULES:
### 1. MANDATORY DETAIL COVERAGE & TRACKING
**BEFORE GENERATING:** List every detail from CURRENT FOCUS AREAS:
- Names: [extract all names]
- Ages/Numbers: [extract all numbers]
- Locations: [extract all places]
- Facts/Situations: [extract all specific facts]

**USAGE TRACKING:** Mark each detail as used to prevent repetition within current questions.

### 2. ABSOLUTE SOURCE RESTRICTION
**ONLY ALLOWED SOURCE:** Details explicitly written in CURRENT FOCUS AREAS bullet points
**COMPLETELY FORBIDDEN:**
- ANY names, places, facts, or details from CONTEXT REFERENCE sections
- ANY topics or content from BATCH_HISTORY
**CONTEXT REFERENCE RULE:** Use CONTEXT REFERENCE only to understand WHO people are or WHAT things mean when
     they appear in CURRENT FOCUS AREAS. NEVER generate questions about CONTEXT REFERENCE content.

### 3. ZERO REPETITION ENFORCEMENT
**ABSOLUTE REQUIREMENT:** Each specific detail can ONLY be mentioned ONCE across all questions.
**ABSOLUTE PROHIBITIONS:**
- Using ANY detail more than once in current questions
- Mentioning ANY topic/detail from BATCH_HISTORY
- Referencing ANY content from CONTEXT REFERENCE sections
- Asking about broader topics not in current bullets
**VERIFICATION:** Before each question, confirm it doesn't repeat previous content.

### 4. ANTI-REPETITION SYSTEM
**DETAIL USAGE PATTERN:**
- First mention: Use full specific detail from bullet point
- Subsequent references: Use pronouns ("he", "she", "it", "that", "my choice")
**VERIFICATION:** Check each question doesn't repeat:
- Specific names/numbers already used
- Topics from BATCH_HISTORY
- Any details or content from reference sections (CONTEXT REFERENCE)

### 5. REALISTIC CONVERSATION STYLE
**NATURAL LANGUAGE:**
- Contractions: "I'm", "don't", "can't"
- Casual words: "kinda", "sorta", "gonna"
- Fillers: "like", "um", "you know"
- Informal: "...", "??", "!!"

### 6. QUESTION VARIETY
**AVOID REPETITIVE PATTERNS:**
- Don't start multiple questions the same way
- Vary question length and complexity

### 7. QUESTION GENERATION STRATEGY
- **Normal question**
- **Seek advice**
- **Ask for help**
- **Request clarification**
- **Get guidance**
- **Express emotions**
- **Validate decisions**
- **Process thoughts**
- **Explore options**

**QUESTION CLUSTERING:**
- Some bullets get 1 question, others get 2-3
- Deep dive into complex situations
- Quick questions for simple details
- User introducing himself/herself should be first question

## OUTPUT REQUIREMENTS:
```

```
Generate exactly <SUB_BATCH_SIZE> questions that:
1. **USE EVERY DETAIL** from current bullet points exactly once
2. Sound like genuine human requests for AI help
3. Focus on specific personal situations mentioned
4. Avoid all repetition from previous batches
5. Show realistic emotional responses to bullet situations
6. Follow natural conversation flow
7. **NEVER repeat specific details within current questions**
8. **NO repetitive "and" chains** in any message

**SUCCESS CRITERIA:**
- Every name, age, location, fact from bullets MUST appear and appear ONLY ONCE
- No repetition of BATCH_HISTORY topics
- Questions sound like real people texting for advice
- All questions trace back to specific bullet details
- Subsequent references use pronouns/generic terms only
- ZERO content from CONTEXT REFERENCE sections

**OUTPUT FORMAT:**
For each question, use this exact format:
[question text] ->-> [bullet_number]

**CRITICAL:**
- Each question MUST end with "->-> [number]" where [number] is the bullet point it's based on
- Use bullet numbers 1, 2, 3, etc. as they appear in CURRENT FOCUS AREAS
- If a question combines details from multiple bullets, use the primary bullet number
- If the question is not generated from any bulletpoints, put N/A
- Generate exactly <SUB_BATCH_SIZE> questions

**Format:** One question per line, natural length, no numbering or extra text.
```

Listing 32: Question generation general domain prompt

```
You are generating realistic coding questions that a DEVELOPER would ask an AI ASSISTANT. Create questions
    based ONLY on the specific details in CURRENT FOCUS AREAS.

## CURRENT FOCUS AREAS (STRICT SCOPE - ONLY SOURCE FOR QUESTIONS):
<FOCUSED_BULLETS>

## QUESTIONS ALREADY COVERED IN THIS BATCH (AVOID THESE):
<BATCH_HISTORY>

## CONTEXT REFERENCE (FOR UNDERSTANDING ONLY - DO NOT GENERATE QUESTIONS ABOUT THIS):
<PREVIOUS_SUB_BATCH_PLANS>
<PREVIOUS_BATCH_PLANS>

## BULLET TYPE DETECTION
**MANDATORY FIRST STEP - CHECK EACH BULLET:**
- If bullet contains "**Time Anchor:**" -> ABSOLUTELY NO CODE, ONLY project/scheduling questions
- If bullet contains "**Personal Introduction:**" -> ABSOLUTELY NO CODE, ONLY career/personal questions
- Otherwise -> Technical bullet, GENERATE CODE

### 1. CURRENT FOCUS AREAS BULLET TYPE IDENTIFICATION - CHECK FIRST
**BEFORE DOING ANYTHING:** Identify the bullet type in CURRENT FOCUS AREAS:
- **Time Anchor:** bullets (contain "Time Anchor:" in title) -> NO CODE GENERATION
- **Personal Introduction:** bullets (contain "Personal Introduction:" in title) -> NO CODE GENERATION
- **Technical bullets:** (all others) -> CODE GENERATION REQUIRED

### 2. MANDATORY DETAIL COVERAGE & TRACKING
**STEP 1 - MANDATORY EXTRACTION:** Before writing ANY questions, you MUST extract and list EVERY SINGLE detail
    from CURRENT FOCUS AREAS:
**Extract ALL of these categories:**
- **Names:** [list EVERY name - developer names, company names, project names, client names]
- **Numbers/Versions:** [list EVERY version number, date, time, quantity, port, ID, measurement]
[Other categories examples]

**STEP 2 - VERIFICATION:** Count total extracted details. You MUST use 100% of them.
**STEP 3 - TRACKING:** As you write each question, mark which specific details it uses.
**STEP 4 - FINAL CHECK:** Before submitting, verify EVERY extracted detail appears in at least one question.
**ABSOLUTE REQUIREMENT:** Every single extracted detail MUST appear in at least one question across the batch.
    NO EXCEPTIONS.

### 3. ABSOLUTE SOURCE RESTRICTION
**ONLY ALLOWED:** Details explicitly written in CURRENT FOCUS AREAS bullet points
**COMPLETELY FORBIDDEN:**
- ANY content from BATCH_HISTORY
- ANY content from CONTEXT REFERENCE sections
- Generic programming questions
- Details not explicitly mentioned in current bullets

**CONTEXT REFERENCE RULE:** Use only to understand WHAT technologies/components mean when they appear in
    CURRENT FOCUS AREAS.

### 4. DETAIL USAGE PATTERN (ANTI-REPETITION)
- **First mention:** Use full specific detail from bullet point (exact names, versions, error messages)
- **Subsequent references:** Use pronouns ("it", "that", "my React app", "the API")
- **VERIFICATION:** Each specific detail appears ONLY ONCE across all questions

### 5. MANDATORY COMPLEX CODE GENERATION
**CRITICAL: 85% of questions MUST include substantial code snippets (20-60+ lines)**
ONLY IF bullet's title in CURRENT FOCUS AREAS is not time anchor or personal introduction
```

```
**ABSOLUTE EXCEPTIONS - NO CODE GENERATION:**
- **Time Anchor:** bullets - NEVER EVER generate any code, programming solutions, or technical implementations
- **Personal Introduction:** bullets - NEVER EVER generate any code, programming solutions, or technical
    implementations
- **FOR PERSONAL INTRODUCTION BULLETS:** Generate questions FROM the perspective of the person introducing
    themselves
The person in the bullet is the USER asking the questions
These are contextual/personal details, NOT technical coding scenarios

**STOP AND VERIFY: If the bullet contains "Time Anchor:" or "Personal Introduction:" in the title, you MUST
    NOT generate ANY code blocks, programming solutions, scripts, or technical implementations. Period.**

**FOR TIME ANCHOR/PERSONAL INTRODUCTION BULLETS:**
- Focus on project management, scheduling, personal goals
- Ask about deadlines, meeting coordination, project planning
- NO code blocks, NO programming solutions, NO technical implementations
- Use natural conversation about timing, goals, and context

**CODE COMPLEXITY REQUIREMENTS (for all other bullets):**
- **Minimum 20-60+ lines per code block**
- **Multiple functions/methods/classes (4-6 minimum)**
- **Realistic imports and dependencies (3-5 minimum)**
- **Proper error handling, validation, edge cases**
- **Complex business logic, database operations, API calls**
- **Production-level structure and realistic variable names**

**REQUIRED PATTERNS (for coding bullets only):**
- **Debugging (40%):** Generate buggy code with realistic, hard-to-spot errors
- **Code Review (25%):** Generate working but suboptimal code needing improvements
- **Implementation (20%):** Generate partial implementations with detailed TODOs
- **Optimization (15%):** Generate slow/inefficient but functional code

**NEVER use simple examples or tutorial-style code - always production-level complexity**

### 6. AUTHENTIC DEVELOPER STYLE
**Language:** Use contractions ("I'm", "don't"), dev slang ("lol", "btw"), fillers ("like", "um"), informal
    punctuation ("...", "??", "!!")
**Emotion:** Show genuine feelings - frustration with bugs, excitement about features
**Natural Flow:** Mix question lengths, include rambling, thinking out loud
**Technical Authenticity:** Include actual error messages, file names, version numbers from bullets

### 7. CHRONOLOGICAL ORDER
Process bullet points in exact order provided. Earlier bullet details appear in earlier questions.

### 8. CODING QUESTION STRATEGY
- **Implementation:** "Help me build [specific feature from bullet]"
- **Debugging:** "I'm getting this error: [specific error]. How do I fix it?"
- **Code Review:** "Can you review this [specific code] and suggest improvements?"
- **Optimization:** "How can I make [specific implementation] faster?"

## OUTPUT REQUIREMENTS:
Generate exactly <SUB_BATCH_SIZE> questions that:

1. **MANDATORY:** Use EVERY SINGLE detail from FOCUSED_BULLETS at least once (names, versions, errors, files,
    specs, etc.)
2. **85% MUST include substantial code snippets (20-60+ lines) with production-level complexity**
3. Sound like genuine developer requests with realistic technical scenarios
4. Follow chronological order of bullet points
5. **ALWAYS include complete, complex code - NEVER use simple examples**
6. Match one of the four coding patterns with appropriate complexity
7. Stay strictly within bullet point scope
8. **MANDATORY VERIFICATION:** Before submitting, confirm every extracted detail appears in the questions

**OUTPUT FORMAT:**
- **CRITICAL: Generate ONLY the developer messages, nothing else**
- Do NOT include question numbers, headers, or organizational text
- **MANDATORY: Separate each complete message with "---MESSAGE_SEPARATOR---"**
- Each message can span multiple lines and include code blocks
- **CRITICAL: Each message MUST end with "->> [number]" where [number] is the bullet point it's based on**
- **MANDATORY: End with "### COMPLETE ###"**

**REQUIRED FORMAT PATTERN:**
[Output format example]

**CRITICAL FORMATTING RULES:**
[Output formatting rules]

**CRITICAL VERIFICATION:** Before each question:
1. **MANDATORY FIRST CHECK: What type of bullet is this?**
    - Time Anchor bullet -> Generate project management/scheduling questions, ABSOLUTELY NO CODE
    - Personal Introduction bullet -> Generate career/personal questions FROM their perspective, ABSOLUTELY NO
        CODE
    - Technical bullet -> Generate coding questions WITH substantial code
2. Does this use a specific detail from current bullets?
3. Have I included substantial, complex code (not simple examples) for technical bullets?
4. Does this sound like a real developer asking for help?
5. Am I following one of the four coding patterns correctly for technical bullets?

**FINAL VERIFICATION BEFORE SUBMITTING:**
Count how many extracted details appear in your questions. It MUST be 100% of all details from CURRENT FOCUS
    AREAS.
```

```
**Generate exactly <SUB_BATCH_SIZE> questions in the format above.**
```

Listing 33: Question generation coding domain prompt

```
You are generating realistic math questions that a USER would ask an AI ASSISTANT. Create questions based ONLY
      on the specific details in CURRENT FOCUS AREAS.

## CURRENT FOCUS AREAS (STRICT SCOPE - ONLY SOURCE FOR QUESTIONS):
<FOCUSED_BULLETS>

## QUESTIONS ALREADY COVERED IN THIS BATCH (AVOID THESE):
<BATCH_HISTORY>

## CONTEXT REFERENCE (FOR UNDERSTANDING ONLY - DO NOT GENERATE QUESTIONS ABOUT THIS):
<PREVIOUS_SUB_BATCH_PLANS>
<PREVIOUS_BATCH_PLANS>

## CRITICAL RULES:
### 1. CURRENT FOCUS AREAS BULLET TYPE IDENTIFICATION - CHECK FIRST
**BEFORE DOING ANYTHING:** Identify the bullet type in CURRENT FOCUS AREAS:
- **Time Anchor:** bullets (contain "Time Anchor:" in title) -> NO MATHEMATICAL WORK GENERATION
- **Personal Introduction:** bullets (contain "Personal Introduction:" in title) -> NO MATHEMATICAL WORK
      GENERATION
- **Mathematical bullets: (all others) -> MATHEMATICAL WORK GENERATION REQUIRED

### 2. MANDATORY DETAIL COVERAGE & TRACKING
**STEP 1 - MANDATORY EXTRACTION:** Before writing ANY questions, you MUST extract and list EVERY SINGLE detail
      from CURRENT FOCUS AREAS:

**Extract ALL of these categories:**
- **Names:** [list EVERY name mentioned - people, places, institutions, etc.]
- **Numbers:** [list EVERY number, age, percentage, score, quantity, measurement]
[Other categories example]

**STEP 2 - VERIFICATION:** Count total extracted details. You MUST use 100% of them.
**STEP 3 - TRACKING:** As you write each question, mark which specific details it uses.
**STEP 4 - FINAL CHECK:** Before submitting, verify EVERY extracted detail appears in at least one question.

**ABSOLUTE REQUIREMENT:** Every single extracted detail MUST appear in at least one question across the batch.
      NO EXCEPTIONS.
### 3. ABSOLUTE SOURCE RESTRICTION
**ONLY ALLOWED:** Details explicitly written in CURRENT FOCUS AREAS bullet points
**COMPLETELY FORBIDDEN:**
- ANY content from BATCH_HISTORY
- ANY content from CONTEXT REFERENCE sections
- Generic questions about mathematical fields
- Details not explicitly mentioned in current bullets

**CONTEXT REFERENCE RULE:** Use only to understand WHO/WHAT things mean when they appear in CURRENT FOCUS
      AREAS.

### 4. DETAIL USAGE PATTERN (ANTI-REPETITION)
- **First mention:** Use full specific detail from bullet point
- **Subsequent references:** Use pronouns ("it", "that", "my homework")
- **VERIFICATION:** Each specific detail appears ONLY ONCE across all questions

### 5. MANDATORY MATHEMATICAL WORK INCLUSION
**CRITICAL:** When referencing problems, equations, or mathematical work that isn't explicitly provided in
      bullets, you MUST generate and include the complete mathematical content.

**ABSOLUTE EXCEPTIONS - NO MATHEMATICAL WORK GENERATION:**
- **Time Anchor:** bullets - Generate questions about scheduling, deadlines, timing without any MATHEMATICAL
      WORK
- **Personal Introduction:** bullets - Generate questions FROM the person introducing themselves about their
      background, career, goals without any MATHEMATICAL WORK
- **FOR PERSONAL INTRODUCTION BULLETS:** Generate questions FROM the perspective of the person introducing
      themselves
The person in the bullet is the USER asking the questions
These are contextual/personal details, NOT MATHEMATICAL WORK scenarios

**FOR TIME ANCHOR/PERSONAL INTRODUCTION BULLETS:**
- Focus on study scheduling, academic deadlines, learning goals
- Ask about exam preparation, study coordination, academic planning
- NO mathematical equations, NO problem-solving, NO calculations
- Use natural conversation about timing, goals, and academic context

**REQUIRED PATTERNS:**
- **Completely Stuck (40%):** NO work shown, just describe what you're trying to solve
- **Partially Stuck (30%):** Show ONLY initial 2-4 steps where you got stuck
- **Need Verification (20%):** Show ONLY final answer/result
- **Conceptual Confusion (10%):** NO calculations, concept-focused questions

**WORK GENERATION REQUIREMENTS:**
- Match mathematical level mentioned in bullet points
- Include specific numbers, variables, expressions
- Create realistic problems users would encounter
- **NEVER use placeholders like "... (insert work)" - ALWAYS generate actual mathematical work**

### 6. AUTHENTIC USER STYLE
```

```
**Language:** Use contractions ("I'm", "don't"), casual slang ("lol", "btw"), fillers ("like", "um"), informal
    punctuation ("...", "??", "!!")
**Emotion:** Show genuine feelings - confusion, frustration, excitement
**Natural Flow:** Mix question lengths, include rambling, thinking out loud

### 7. CHRONOLOGICAL ORDER
Process bullet points in exact order provided. Earlier bullet details appear in earlier questions.

### 8. QUESTION GENERATION STRATEGY
- **Problem-Solving:** "Help me solve [specific problem]"
- **Concept Clarification:** "I don't understand [specific concept]"
- **Solution Verification:** "Can you check if my solution is correct?"
- **Method Explanation:** "Why does [specific method] work?"

## OUTPUT REQUIREMENTS:
Generate exactly <SUB_BATCH_SIZE> questions that:

1. **MANDATORY:** Use EVERY SINGLE detail from FOCUSED_BULLETS at least once (names, ages, dates, traits,
    goals, timeframes, etc.)
2. **80% MUST include substantial mathematical work** (equations, calculations, solution attempts)
3. Sound like genuine user requests with realistic mathematical content
4. Follow chronological order of bullet points
5. **ALWAYS include complete mathematical problems/equations - NEVER use placeholders**
6. Match one of the four behavioral patterns with appropriate work shown
7. Stay strictly within bullet point scope
8. **MANDATORY VERIFICATION:** Before submitting, confirm every extracted detail appears in the questions

**OUTPUT FORMAT:**
- **CRITICAL: Generate ONLY the user messages, nothing else**
- Do NOT include question numbers, headers, or organizational text
- **MANDATORY: Separate each complete message with "---MESSAGE_SEPARATOR---"**
- Each message can span multiple lines and include mathematical expressions
- **CRITICAL: Each message MUST end with "->-> [number]" where [number] is the bullet point it's based on**

**REQUIRED FORMAT PATTERN:**
[Output format example]

**CRITICAL FORMATTING RULES:**
[Output formatting rules]

**CRITICAL VERIFICATION:** Before each question:
1. Does this use a specific detail from current bullets?
2. **Is this a Time Anchor or Personal Introduction bullet? If YES, do NOT include any mathematical work**
3. **For Personal Introduction: Am I generating questions FROM the person's perspective (they are the user)?**
4. Have I included actual mathematical work (not placeholders) for mathematical bullets?
5. Does this sound like a real user asking for help?
6. Am I following one of the four behavioral patterns correctly for mathematical bullets?

**FINAL VERIFICATION BEFORE SUBMITTING:**
Count how many extracted details appear in your questions. It MUST be 100% of all details from CURRENT FOCUS
    AREAS.

**Generate exactly <SUB_BATCH_SIZE> questions in the format above.**
```

Listing 34: Question generation math domain prompt

```
You will receive an AI assistant's reply. Determine whether it contains a **direct, specific question** that
    the user must answer next by providing new information, preferences, or a decision.

- **YES** if, and only if, the assistant's reply asks for a concrete user response (e.g. "What's your budget
    for this trip?", "Which option would you prefer?").
- **NO** for generic or rhetorical prompts (e.g. "Any questions?", "Would you like to dive deeper?", "Consider
    your budget") that do not demand an immediate, specific answer.

AI ASSISTANT RESPONSE:
<assistant_response>

CRITICAL NOTE: Respond only in English. Do not include any Chinese.
Output exactly **YES** or **NO**, nothing else.
```

Listing 35: Check assistant's response include question prompt

```
You are simulating a typical user in conversation. Decide if you would ask a follow-up question after the AI's
    response.

**CONVERSATION CONTEXT:**
- DOMAIN: <domain>
- TITLE: <title>
- THEME: <theme>
- SUBTOPICS: <subtopics>
- Recent History: <formatted_history>
- AI's Last Response: <assistant_response>

**ASK FOLLOW-UP ("yes") WHEN:**
1. **Missing Info**: The response lacks details you genuinely need to proceed
    - Specific steps for a process you're trying to follow
    - Key parameters (dates, amounts, requirements) for a decision you're making
    - Clarification on which option applies to your specific situation

2. **Genuine Confusion**: Something is unclear or contradictory
```

```
   - Technical terms used without explanation that block understanding
   - Conflicting information that affects your next action
   - Ambiguous instructions where the wrong interpretation has consequences

3. **Incomplete Practical Guidance**: You asked "how to" but can't actually do it yet
   - Missing steps in a procedure
   - Lacks specifics needed for implementation
   - Assumes knowledge you don't have

**NO FOLLOW-UP ("no") WHEN:**
1. **Good Enough to Proceed**: You have what you need

**OUTPUT:** Only "yes" or "no"
```

Listing 36: Check need for followup prompt

```
You must respond only in English. Never switch to Chinese or any other language mid-sentence. All responses
     should be entirely in English.
Respond to the user's message by either:
  · Fully answering their question
  . Provide a comprehensive answer
  · Answering plus asking at most ONE follow-up question if you need more detail

Always honor these rules:
1. Do NOT ask questions the user already answered.
2. Only ask a question if you genuinely need context to provide a complete, actionable answer.
3. Keep your main answer clear and comprehensive before you ask.
4. Use the following system inputs:

DOMAIN: <domain>
TITLE: <title>
THEME: <theme>
SUBTOPICS: <subtopics>
PREVIOUS PLANS SUMMARY: <previous_plans_summary>
PREVIOUS BATCHES OF THIS PLAN: <previous_batches>
CURRENT HISTORY: <current_batch_messages>

CRITICAL NOTE: Respond only in English. Do not include any Chinese.

**Output**
Return exactly what you'd say to the user—no tags, no internal notes.
```

Listing 37: Assistant LLM answer generation prompt

```
You are role-playing as a real user having an authentic conversation with an AI chat assistant.
The AI assistant has just asked you a question, and you need to provide a natural, human-like response.

#### Input Context You'll Receive:
- **Current Batch Message History**: The conversation flow leading to the AI's question
- **Domain, Title, Theme & Subtopics**: The main subject of this conversation
- **Previous Plans**: Summary of earlier conversation contexts for continuity
- **Current Plan**: The overarching narrative direction for this conversation batch
- **AI's Question**: The specific question the assistant asked that requires your response

#### INPUTS:
**Current Batch Message History**: <current_batch_messages>
**DOMAIN: <domain>
**TITLE: <title>
**THEME: <theme>
**SUBTOPICS: <subtopics>
**Previous Plans Summary**: <previous_plans_summary>
**Previous Batches of This Plan**: <previous_batches>
**Current Plan**: <current_plan>
**AI's Question**: <ai_last_message>

#### CRITICAL: Keep Responses SHORT and Natural
**Real users give brief, to-the-point answers to AI questions**

#### Your Role & Behavior:
You are a real human user with:
- Personal experiences, opinions, and emotions
- Natural speech patterns and conversational habits
- Realistic knowledge limitations and curiosity
- Consistent personality traits across the conversation

**Language Authenticity:**
- Use lots of contractions: "I'm", "don't", "can't", "it's", "that's"
- Include casual slang: "lol", "btw", "tbh", "kinda", "sorta", "gonna", "wanna"
- Add filler words: "like", "um", "you know", "I mean", "so", "well"
- Use informal punctuation: multiple periods "...", question marks "??", exclamation points "!!"

**Imperfect Natural Speech:**
- Include minor typos and informal grammar
- Add rambling elements: "I mean, ..."
- Include thinking out loud: "hmm", "actually", "oh", "maybe"

**Emotional Authenticity:**
- Show genuine feelings: excitement, frustration, uncertainty, hope
- Use emotional language
- Add personal reactions: "ugh", "omg", "yay", "oof", "blah"
```

```
#### Response Guidelines:
**STEP 1 - Check Plans for Existing Information**:
- **First**, carefully review the Current Plan and Previous Plans for any information that answers the AI's
    question
- **If found in plans**: Base your response on that established information to maintain story continuity
- **If not found in plans**: Create a new answer that aligns with the topic, theme, and existing storyline

**STEP 2 - Answer the AI's Question Directly**:
- Keep your reply focused on answering; do **not** introduce new questions.
- Give a direct answer to what the AI asked
- Don't over-explain or provide unnecessary details
- Answer like you would in a real text conversation
- Include personal context or examples when natural
- **CRITICAL**: Ensure your answer doesn't contradict anything established in previous or current plans

**Stay Consistent with Context**:
- Maintain the same personality and circumstances throughout
- Keep your responses aligned with the current topic and theme

**Response Characteristics**:
- **Tone**: Match the conversation's emotional tone and your established personality
- **Authenticity**: Sound like a real person, not an AI trying to sound human

#### Critical Instructions:
- **Be concise** - Real people don't write long in chat
- Keep your reply focused on answering; do **not** introduce new questions.
- **ALWAYS check plans first** - Look for any information that answers the AI's question before creating new
    details
- **Maintain consistency** - Never contradict information established in current or previous plans
- **Fill gaps naturally** - If plans don't have the answer, create responses that fit the established
    storyline
- ONLY provide your response as the user - no meta-commentary
- Stay in character as a human user throughout
- Answer the question but don't feel obligated to ask a question back (the AI asked YOU)
```

Listing 38: User LLM answer generation prompt

```
You must respond **only in English**. Do not include any Chinese characters or phrases in your response.
You are a real person having a conversation with an AI assistant. Based on the conversation history and the AI
    's last response, ask ONE natural follow-up question.

## CONTEXT:
**Current Batch Message History**: <current_batch_messages>
**DOMAIN: <domain>
**TITLE: <title>
**THEME: <theme>
**SUBTOPICS: <subtopics>
**Previous Plans Summary**: <previous_plans_summary>
**Previous Batches of This Plan**: <previous_batches>
**Current Plan**: <current_plan>
**AI's Response**: <ai_last_message>

## YOUR TASK
Ask a follow-up question (10-20 words) that a real person would naturally ask after receiving the AI's
    response.

## CRITICAL RULES TO PREVENT REPETITION
Before generating your question:
1. **Scan the Current Batch Message History** for all topics already discussed
2. **Check Previous Batches** for questions already asked
3. **Never ask about something already covered**

If you notice your question seeks information already provided in the conversation history, STOP and generate
    a completely different question.

## HOW REAL PEOPLE ASK FOLLOW-UPS
### Natural conversation starters:
- "oh wait..." / "hmm..." / "actually..." / "btw..." / "ok but..."
- "that's cool but..." / "makes sense, though..." / "yeah but what about..."

### Authentic reaction patterns:
**Building on the last ai response:**
- Ask about a specific topic not yet covered
- Connect it to your personal situation from the Current Plan

**Showing genuine reactions:**
- If AI gave good news -> "nice! but does that mean..."
[Other examples]

### Question types that feel natural:
- **Practical concerns**: "how long does that usually take?"
[Other types]

## NATURAL SPEECH PATTERNS
Include these elements to sound human:
- Contractions: "don't", "can't", "won't", "that's"
- Casual words: "kinda", "sorta", "gonna", "like"
- Emotional reactions: "ugh", "hmm", "oh", "yikes"
- Informal punctuation: "..." or "??" or "!"
```

```
## CONVERSATION FLOW AWARENESS
Based on where you are in the Current Batch Message History:
Ask broader exploratory questions
Ask for specific details or comparisons
Ask about implementation or next steps

## AUTHENTICITY CHECKLIST
[Some examples]

CRITICAL NOTE: Respond only in English. Do not include any Chinese.

## YOUR RESPONSE:
[Generate only the follow-up question, nothing else]
```

Listing 39: User LLM ask followup question prompt

```
I provide you with a text. Your task it to identify all the details stated in the text,
and output that in key: value format.
E.g.:
Key 1: Value 1,
Key 2: Value 2,
Key 3: Value 3,
....

Also at the end, I want you to provide a brief summary of what this text was about in this format: Summary: '
    summarized text'

Note: only output key-values and the summary. DO NOT provide any explanation before or after that.
Note: Do not output Key 1, Key 2, ...

**Previous Context:**
{history}

text: {text}
```

Listing 40: Key-value extraction prompt

```
You are a highly analytical AI assistant. Your task is to analyze the latest conversation exchange and produce
    a structured summary of key information and insights.

**Your Internal Process:**
To ensure maximum accuracy, you must first think step-by-step.
1.   **Analyze:** Break down the user's latest message.
2.   **Identify:** Pinpoint all facts, instructions, and updates.
3.   **Deduce:** Reason about the implications of the new information in the context of the conversation
     history. What is the user's underlying goal or state?
4.   **Format:** After completing your internal analysis, format the conclusions into the 'Extracted Facts'
     structure.

**Crucial Instruction:** Your final output must **ONLY** be the 'Extracted Facts' block. **DO NOT** include
    your step-by-step reasoning or any other text in your response. Strictly follow the format shown in the
    example's output.

---
**EXAMPLE**

**Conversation Context:**
* **Recent Conversation History:**
    USER: Hey, I need some help with the "Project Phoenix" launch plan.
    ASSISTANT: Of course. What do you need?
    USER: The launch date is set for September 15th, 2025. I'm responsible for the marketing materials.
* **Latest Exchange to Analyze:**
    USER: Okay, the final budget for the social media campaign is $7,500. The client, Innovate Corp, just
         approved it. Please find me three case studies of successful B2B SaaS launches by tomorrow, August
         28th. And don't include any of our direct competitors in the examples.
    ASSISTANT: Understood. I will find three case studies of successful B2B SaaS launches, excluding
         competitors, and have them for you by tomorrow, August 28th. The approved budget of $7,500 for the
         social media campaign has been noted.

**Example of Correct Final Output:**
    * The client's name is "Innovate Corp".
    * The project is related to a "B2B SaaS launch".
    * The final budget for the social media campaign is \$7,500.
    * A deadline is set for "tomorrow, August 28th".
    * User intends to review three case studies for the project.
    * Instruction: Find three case studies.
    * Constraint: Do not include direct competitors in the examples.
    * The budget for the social media campaign has been approved by the client.
    * The user is under a deadline and needs the case studies urgently to inform their work on the marketing
         materials.

---
**ACTUAL TASK**

**Recent Conversation History:**
{history}

**Latest Exchange to Analyze:**
USER: {latest_user_message}
ASSISTANT: {latest_assistant_message}
```

```
**Extracted Facts:**
```

Listing 41: Scratchpad creation prompt

```
You are tasked with summarizing and compressing scratch pad content to fit within a specific token limit.
**Input Content:**
{content}

**Target Length:** {tokens_limit} tokens

**Your Task:**
Compress this content by clustering related information, removing redundancy, and prioritizing the most
    important details.

**Process:**
1. **Cluster**: Group related information by topic, entity, or theme
2. **Deduplicate**: Remove redundant or repetitive information
3. **Prioritize**: Keep the most important and contextually relevant details
4. **Compress**: Condense while maintaining essential meaning and context

**Output Format:**
Return ONLY the compressed content organized as:

**KEY ENTITIES & RELATIONSHIPS:**
- [Most important people, organizations, systems mentioned]

**CORE DECISIONS & PREFERENCES:**
- [Critical decision points, requirements, constraints]

**PROCESSES & WORKFLOWS:**
- [Essential procedural information and methodologies]

**USER PREFERENCES:**
- [User's stated likes, dislikes, preferred methods, settings, choices]

**USER INSTRUCTIONS:**
- [Specific directions, commands, or guidance provided by the user]

**IMPORTANT DATES:**
- [Deadlines, milestones, scheduled events, time-sensitive information]

**CRITICAL CONTEXT:**
- [Background information necessary for understanding]

**ACTIONABLE ITEMS:**
- [Next steps, pending actions, deadlines]

**IMPORTANT DEVELOPMENTS:**
- [Significant events, changes, milestones]

**Requirements:**
- Stay within {tokens_limit} tokens
- Eliminate redundancy while preserving essential information
- Eliminate older values when there is newer and updated value for a thing
- Maintain chronological context where important
- Prioritize information with ongoing relevance"

**CRITICAL LENGTH REQUIREMENT:**
- Your response should be approximately {tokens_limit} tokens
- If your draft is significantly shorter than {tokens_limit} tokens, ADD MORE DETAIL
```

Listing 42: Scratchpad summarization prompt

```
I provide you with a user query and a text chunk.
You need to decide if the text chunk is nesseccery for answering user question.
If we need the text chunk to answer the user question, or if the text chunk is part of the answer to user
    question return 'yes'
If the text chunk is noise and not relevant to user question, return 'no'.
Output format: Return only 'yes' or 'no', without any explantion before or after that.

User query: {query} \n\n
Text chunk: {doc_text}
```

Listing 43: Scratchpad noise filtering prompt

```
You are an assistant that MUST answer questions using ONLY the information provided in the context below.

STRICT INSTRUCTIONS:
1. Answer ONLY based on the provided context
2. Do NOT use your internal knowledge

CONTEXT:
<context>

QUESTION:
<question>
```

```
ANSWER REQUIREMENTS:
- Be direct and concise
- Only output the answer to the question without any explanation

RESPONSE:
```

Listing 44: Answer generation with RAG prompt

## I  LLM USAGE

We used ChatGPT[4] as a writing assistant. Specifically, we first drafted the paper and then employed ChatGPT to refine the text/assist with rephrasing and grammar. The suggestions were manually reviewed and edited before inclusion in the final version.

---

[4]https://chatgpt.com

