# OpenReview forum: "Beyond a Million Tokens: Benchmarking and Enhancing Long-Term Memory in LLMs"
_ICLR.cc/2026/Conference — ICLR 2026 Poster_

### Official Review · Reviewer_dWNe · 2025-10-23

**Soundness:** 3
**Presentation:** 2
**Contribution:** 4
**Rating:** 6
**Confidence:** 4

**Summary:**

The paper proposes BEAM, a benchmark consisting of 100 synthetic conversations that span up to 10M tokens along with corresponding probing questions. BEAM aims to evaluate the abilities of large language models (LLMs) to reason over long-context input. Unlike existing benchmarks, it is more diverse, more realistic and covers multiple memory subtasks. Moreover, the paper introduces LIGHT, a framework that mimics episodic, working, and scratchpad memories to improve the long-context performance of LLMs. Existing LLMs do not perform well on BEAM. Augmenting LLMs with LIGHT improves their performance compared to standard RAG.

**Strengths:**

- The benchmark contains more realistic conversations, is more diverse, covers many memory subtasks, and contains very long context scenarios. It is challenging even for the best existing LLMs.
- The paper contains all the required details to understand and possibly replicate the dataset creation. The appendix provides even more details about the dataset creation.
- The proposed method LIGHT outperforms standard RAG, which is a strong baseline.

**Weaknesses:**

- There is no qualitative analysis about the kind of errors that the LLMs make.
- Regarding readability, some sections of the paper are very dense and harder to read. The level of details describing the dataset creation and the propose method is adequate, but the flow of some paragraphs could be improved using for example more bullet points.
- All the figures are tiny. They cannot be properly read without zooming. The captions of all figures are very short and do not clearly describe them. Figure 1, which describes dataset creation, and Figure 2, which explains the proposed method, should be highlighted more. The texts and notations used in those figures should be as self-explanatory as possible to allow a quick overview of the contributions.
- The paper could present a few baselines besides RAG at least for the small models if it is too expensive.
- Concerning related work, you might want to consider similar work https://arxiv.org/abs/2502.13791 that also relies on plans/templates to generate synthetic conversations in order to evaluate LLMs' ability to follow instructions in long-context.
- Minor typo: the generated dialogues in the appendix contains typos ("Trunicated" in red) .

**Questions:**

- Could you clearly list all the human interventions during dataset creation?
- Have you manually reviewed the generated conversations (not the probing questions or plans) ? Do you have metrics that asses the quality of these conversations? Do you observe hallucinations (not following the plans) in the generated conversations?
- What are the main limitations that prevent the LLMs from performing well on this task (even using LIGHT)? What are the specific issues that LIGHT can or cannot address?
- How does LIGHT compare to other recent long-context baselines other than RAG? This could be done only on the small models if too expensive to run.

---

> ### Author Response · Authors · 2025-11-22
> **Response to Reviewer dWNe - Part 1**
>
> We are grateful for your careful evaluation and for highlighting the strengths of both BEAM and LIGHT. We appreciate the constructive suggestions and questions you raised, and we address them point by point below.
>
> > There is no qualitative analysis about the kind of errors that the LLMs make.
>
> We added a qualitative error analysis for all the memory abilities, which is shown in Appendix G, Page 46.
>
> > Regarding readability, some sections of the paper are very dense and harder to read. The level of details describing the dataset creation and the propose method is adequate, but the flow of some paragraphs could be improved using for example more bullet points.
>
> We did not use many bullet points due to space limitations. However, in light of your suggestion, we have applied this valuable recommendation in the dataset creation section.
>
> > All the figures are tiny. They cannot be properly read without zooming. The captions of all figures are very short and do not clearly describe them. Figure 1, which describes dataset creation, and Figure 2, which explains the proposed method, should be highlighted more. The texts and notations used in those figures should be as self-explanatory as possible to allow a quick overview of the contributions.
>
> We increased the size of all figures, provided comprehensive captions, referenced Figure 1 more clearly in the text, and ensured that the notations used in the figure are explained in the text.
>
> > The paper could present a few baselines besides RAG at least for the small models if it is too expensive.
>
> Alongside the long-context LLMs and RAG, we also incorporated ReadAgent [1], a method for enhancing the long-term memory ability of LLMs, and evaluated it on BEAM in comparison with LIGHT. The results show that our proposed model performs better than ReadAgent across all conversation lengths. The results of this comparison can be found in Appendix C.5, Page 33.
>
> > Concerning related work, you might want to consider similar work [1] that also relies on plans/templates to generate synthetic conversations in order to evaluate LLMs' ability to follow instructions in long-context.
>
> We added this work to the related work section.
>
> > Minor typo: the generated dialogues in the appendix contains typos ("Trunicated" in red) .
>
> We corrected the typos.
>
> > Could you clearly list all the human interventions during dataset creation?
>
> The steps involving human annotation were:
> 1. As explained in the paragraph starting at Line 165, human annotators specify the target domains of the conversations, and after GPT-4.1 generates candidate titles, themes, and subtopics, the annotators refine these outputs for topical diversity.
> 2. The next involvement of the human annotators is evaluating the quality of the generated conversations. They reviewed all conversations and, based on the guidelines, annotated each conversation according to three metrics.
> 3. The next involvement occurs after the probing questions are generated. The annotators go through the generated probing questions to discard invalid ones and select two probing questions for each memory ability in each conversation from the validated set.
> 4. After determining the final set of probing questions, the annotators design rubrics for each probing question to enable the nugget evaluation. In this step, they create a list of nuggets for each probing question.

---

> ### Author Response · Authors · 2025-11-22
> **Response to Reviewer dWNe - Part 2**
>
> > Have you manually reviewed the generated conversations (not the probing questions or plans) ? Do you have metrics that asses the quality of these conversations? Do you observe hallucinations (not following the plans) in the generated conversations?
>
> Yes, human annotators manually reviewed the generated conversations.
> For this task, the annotators were again provided with guidelines on how to evaluate the generated conversations and assign scores. Because the conversations are very long and it is impractical to read them in full, annotators were instructed to read the initial 25 dialogue turns to understand what the conversation is about and to become familiar with the theme and flow of the chat. They were instructed to check the follow-up questions proposed by both the user and the assistant, the adherence of the conversation to the conversation plan and user utterances, and whether any hallucinations or unexpected behaviors occur. After that, they were asked to read several random turns between the initial 25 turns and the middle of the conversation to assess whether the conversation is progressing according to the conversation plan. They then had to read 25 dialogue turns in the middle of the conversation, followed by additional random turns between the middle and the end, and finally 25 dialogue turns near the end of the conversation. In this way, annotators read different parts of each conversation and ensured good coverage. For every conversation, they were required to rate the three metrics mentioned in Appendix B.2, Page 17, Line 908, on a scale from 1 to 5.
>
> Before performing the evaluation task, the annotators went through 20 conversations and scored them separately, and we measured their agreement using Cohen’s Kappa. Their agreement for “Coherence and Flow” was 70.44 percent, for “Dialogue Realism” was 73.91 percent, and for “Complexity and Depth” was 78.49 percent.
>
> According to the annotators’ observations, they did not identify cases of hallucination where the generated dialogue did not follow the conversation plans. This is because our data generation pipeline is systematic and controllable. After generating the conversation plans, user utterances are generated directly from these plans, and the assistant utterances and follow-up turns are generated based on the user utterances and the same plans. This ensures that the resulting conversations are coherent and adhere closely to the initial conversation plan or plans.
>
> > What are the main limitations that prevent the LLMs from performing well on this task (even using LIGHT)? What are the specific issues that LIGHT can or cannot address?
>
> Thank you for asking this. We conducted a qualitative analysis, which is provided in Appendix G, Page 46. The main failure pattern for LIGHT overall occurs when the response to the probing question is present in the context, but the LLM is still unable to answer correctly. For example, in abstention evaluation, if we have similar concepts and information to what the question is asking, the LLM becomes confused and hallucinates. These are limitations of the reader LLM, and LIGHT cannot address them.
>
> Regarding the issues LIGHT can address, our results show that LIGHT provides consistent and substantial improvements across most memory abilities. Its advantages are most apparent in recall-intensive settings such as multi-hop reasoning and summarization, where the model must retrieve and integrate information scattered across long spans of dialogue. In these tasks, LIGHT shows the largest gains because the scratchpad, episodic memory, and retrieval filtering work together to surface the most relevant evidence.
>
> Beyond these abilities, LIGHT also improves performance in preference following, temporal reasoning, knowledge update, instruction following, information extraction, and event ordering. Across these categories, LIGHT outperforms all four LLMs used in our experiments in most settings, demonstrating that its memory components help the model maintain coherence, track evolving information, and correctly ground its reasoning even as the context length increases.

---

> ### Author Response · Authors · 2025-11-22
> **Response to Reviewer dWNe - Part 3**
>
> > How does LIGHT compare to other recent long-context baselines other than RAG? This could be done only on the small models if too expensive to run.
>
> We added another baseline and compared it with LIGHT. The results are provided in Appendix C.5, Page 33.
>
> Regarding other recent baselines, there is a line of work in context compression that aims to compress long context into fewer tokens, either in text form or as dense vectors. In this category, we tested AutoCompressor [2]. It accepts up to 6K tokens and returns a dense embedding of the input, which then must be passed to the LLM they have finetuned. We evaluated it on BEAM by recursively compressing the text, but the results were very poor even at the 100K length, so we removed it from our evaluation.
>
> Another line of work includes memory-augmented transformers, which require training and fine-tuning on the data, and this was not feasible with our computational resources. We expect that LIGHT will outperform this line of work as well.
>
> ​​We hope that the additional experiments and revisions help present the work more clearly, and we appreciate your consideration of these updates in your final assessment.

---

> ### Author Response · Authors · 2025-11-22
> **Response to Reviewer dWNe - Part 4**
>
> [1] Lee KH, Chen X, Furuta H, Canny J, Fischer I. A human-inspired reading agent with gist memory of very long contexts. arXiv preprint arXiv:2402.09727. 2024 Feb 15.
>
> [2] Chevalier A, Wettig A, Ajith A, Chen D. Adapting language models to compress contexts. arXiv preprint arXiv:2305.14788. 2023 May 24.

---

### Official Review · Reviewer_77HX · 2025-10-31

**Soundness:** 2
**Presentation:** 2
**Contribution:** 2
**Rating:** 4
**Confidence:** 3

**Summary:**

The paper tackles the challenge of evaluating and improving long-term conversational memory in LLMs by introducing BEAM, a benchmark of 100 coherent, topically diverse dialogues (100K–10M tokens) paired with 2,000 validated probing questions spanning ten memory abilities. It also proposes LIGHT, a cognition-inspired inference framework that combines (i) episodic memory via retrieval over the full conversation, (ii) short-term working memory of recent turns, and (iii) a maintained scratchpad of salient facts filtered for relevance at query time. Experiments across multiple backbones show consistent gains of LIGHT over long-context and RAG baselines; ablations indicate each memory component contributes to performance.

**Strengths:**

- BEAM spans multi-domain dialogues at four lengths (128K, 500K, 1M, 10M tokens) and evaluates 10 distinct memory abilities, filling gaps in prior single-domain, recall-heavy datasets.
- Component-wise ablations substantiate that each memory module contributes meaningfully to performance.
- Results across context scales (100K – 10M) and models show consistent improvements; the paper also includes component-wise ablations and sensitivity to retrieval budget K.

**Weaknesses:**

- LIGHT’s episodic memory relies on FAISS with BGE-small embeddings; the paper does not systematically explore robustness to alternative embedding models or indexing setups beyond a brief note.
- While a human validation step is described for probes and a separate human evaluation of conversation quality is reported, rater pool size/protocols are not clearly specified, making it hard to assess annotation reliability and potential residual biases over very long chats.
- LIGHT maintains an episodic index and a growing scratchpad over extremely long contexts. The paper varies K but does not report end-to-end latency/memory/GPU budgets for 1M–10M-token settings, limiting practical interpretability.

**Questions:**

- Can you clarify which model you are using for the ablation study? Since the results on Table 8 do not match the results on Table1.
- Why was BAAI/bge-small-en-v1.5 selected for the episodic index? Was the choice driven by speed/latency, availability, or prior retrieval effectiveness on dialogue passages? Any comparisons versus larger BGE variants, E5, or modern instruct-tuned embedding models like openai text embedding 001.

---

> ### Author Response · Authors · 2025-11-22
> **Response to Reviewer 77HX - Part 1**
>
> Thank you for your review and for your understanding of the motivations and strengths of our work. We value the constructive comments you provided and respond to each point below.
>
> > LIGHT’s episodic memory relies on FAISS with BGE-small embeddings; the paper does not systematically explore robustness to alternative embedding models or indexing setups beyond a brief note.
>
> Based on your request, we ran experiments with a larger embedding model (bge-large) and compared LIGHT with the baseline RAG. With the larger embedding model, LIGHT obtained greater improvements than the baseline RAG, which shows the superiority of our episodic retrieval and demonstrates the robustness of LIGHT to the choice of embedding model. Using this stronger embedding model, LIGHT achieves good improvements at 1M and 10M. In comparison with the baseline RAG, the improvement for LIGHT is also higher at 100K, 1M, and 10M. The results for this experiment are reported in Appendix C.3, Page 31.
>
> Regarding the indexing setup, we also experimented with an alternative index. In our main experiments, we used IndexFlatIP, and in the new experiment we tested IndexHNSWFlat. With IndexHNSWFlat, LIGHT showed decreased performance at 500K and 1M, which is expected, but improved performance at 10M. The reason for the improvement at 10M is the increase in abstention, while performance on most memory abilities requiring long-term reasoning decreased. According to the error analysis in Appendix G, Page 46, this increase in abstention is justified because one of the main failure modes of abstention occurs when the context provided to the LLM contains highly similar concepts to the query, leading the model to provide an answer. The results for this experiment are reported in Appendix C.4, Page 32.
>
> > While a human validation step is described for probes and a separate human evaluation of conversation quality is reported, rater pool size/protocols are not clearly specified, making it hard to assess annotation reliability and potential residual biases over very long chats.
>
> We used an internal system for the annotators, and the annotations were completed by two students with a CS background.
> Steps involved human are as follows:
>
> 1. **Validating probing questions**: The annotators were first provided with guidelines and instructions on how to perform the task, and a training session was held. They went through a calibration set together, completed multiple examples, and discussed their opinions on each example.
>
> 2. **Designing rubrics**: The annotators were given guidelines on how to design rubrics along with examples and templates to make the task easier and more unified. A training and calibration session was then held, during which annotators reviewed many samples together, asked questions, and received clarification on ambiguous and edge cases.
>
> 3. **Evaluating generated conversations qualities**: For this task, the annotators were again provided with guidelines on how to evaluate the generated conversations and assign scores. Because the conversations are very long and it is impractical to read them in full, annotators were instructed to read the initial 25 dialogue turns to understand what the conversation is about and to become familiar with the theme and flow of the chat. They were instructed to check the follow-up questions proposed by both the user and the assistant, the adherence of the conversation to the conversation plan and user utterances, and whether any hallucinations or unexpected behaviors occur. After that, they were asked to read several random turns between the initial 25 turns and the middle of the conversation to assess whether the conversation is progressing according to the conversation plan. They then had to read 25 dialogue turns in the middle of the conversation, followed by additional random turns between the middle and the end, and finally 25 dialogue turns near the end of the conversation. In this way, annotators read different parts of each conversation and ensured good coverage. For every conversation, they were required to rate the three metrics mentioned in Appendix B.2, Page 17, Line 908, on a scale from 1 to 5.
> Before performing the evaluation task, the annotators went through 20 conversations and scored them separately, and we measured their agreement using Cohen’s Kappa. Their agreement for “Coherence and Flow” was 70.44 percent, for “Dialogue Realism” was 73.91 percent, and for “Complexity and Depth” was 78.49 percent.

---

> ### Author Response · Authors · 2025-11-22
> **Response to Reviewer 77HX - Part 2**
>
> > LIGHT maintains an episodic index and a growing scratchpad over extremely long contexts. The paper varies K but does not report end-to-end latency/memory/GPU budgets for 1M–10M-token settings, limiting practical interpretability.
>
> In LIGHT, the episodic memory and the scratchpad can be constructed offline, and therefore they do not add to latency. They can also be constructed in parallel, which likewise does not contribute to latency. For completeness, we measured the cost of each step: reasoning over the preceding dialogue turn and updating the scratchpad takes on average 1.93 seconds, and extracting the key-value pairs for the episodic memory takes on average 2.03 seconds. The only step that adds to the latency at inference time is filtering the uncorrelated irrelevant parts of the scratchpad with respect to the query, which takes on average 1.21 seconds.
>
> Regarding the memory, the size of the scratchpad is fixed, and we keep it at 15K tokens.
> In terms of memory usage for embeddings, the memory required for RAG is 3 to 6 times larger than the memory needed for LIGHT’s episodic memory because RAG embeds full chunks that make up very long conversations, whereas LIGHT stores only extracted key–value pairs, which are much shorter than the chunks used by RAG.
> Regarding GPU resources, for our experiments we used two A100 80G SXM GPUs for running the LLM.
>
> > Can you clarify which model you are using for the ablation study? Since the results on Table 8 do not match the results on Table1.
>
> We really appreciate you pointing this out. For the ablation study, we used the same LLM (Qwen2.5 32B AWQ). The results in Table 1 were correct before and remain correct. The results for 1M and 10M in Table 8 match Table 1, but the results for 100K and 500K did not match. The reason is that we made a small modification in LIGHT and forgot to update the first column of Table 8 for 100K and 500K. The other columns, as well as the values for 1M and 10M, were already correct. We have now updated Table 8 accordingly.
>
> > Why was BAAI/bge-small-en-v1.5 selected for the episodic index? Was the choice driven by speed/latency, availability, or prior retrieval effectiveness on dialogue passages? Any comparisons versus larger BGE variants, E5, or modern instruct-tuned embedding models like openai text embedding 001.
>
> Thank you for your thoughtful question. We selected this embedding model primarily due to computational resource limitations, since larger embedding models require substantially more resources and introduce higher latency. Running the baseline RAG experiments with a larger embedding model takes considerably longer, because RAG embeds very long chats. In terms of memory usage, RAG requires between 3 and 6 times more memory than LIGHT’s episodic memory on average. This is because LIGHT stores only the extracted key–value pairs, which are much shorter than the chunks embedded by RAG.
> Furthermore, bge-small is a good performer on the MTEB leaderboard, so it was a balance of speed and performance.
>
> We trust that the additional experiments and improvements strengthen the overall presentation of the work, and we appreciate your consideration of these enhancements in your final assessment.

---

### Official Review · Reviewer_z6rF · 2025-10-31

**Soundness:** 3
**Presentation:** 2
**Contribution:** 3
**Rating:** 4
**Confidence:** 4

**Summary:**

This paper introduces BEAM, a large-scale benchmark for evaluating long-term memory in conversational settings, featuring dialogues up to 10M tokens and 2,000 probing questions across ten distinct memory abilities—including three newly introduced dimensions such as instruction following, event ordering, and contradiction resolution.
The paper presents LIGHT, a cognitively inspired framework integrating episodic (retrieval), working (recent context), and scratchpad (salient fact accumulation) memory modules.
Experiments on multiple open- and closed-source LLMs demonstrate that LIGHT improves accuracy over RAG and vanilla baselines.

**Strengths:**

1. Good scale and diversity: context length in conversation setting up to 10M across multiple domains. The samples are validated by human.

2. New complimentary metrics (instruction following, event ordering, and contradiction resolution.) to evaluate the behaviour of LLM in long context setting

3. Cognitive framework - LIGHT - shows convincing improvement in comparison to strong RAG baseline

**Weaknesses:**

1. Limited dataset size: only 100 conversations for benchmarking, this could lead to high variance in evaluation results.

2. The paper doesn't deeply analyze why models fail on specific memory abilities. For instance, why do all methods struggle with contradiction resolution? No error analysis or qualitative examination of failure modes beyond the single case study in Appendix F.

3. The combination of retrieval + working memory + external memory is not particularly novel. The main contribution is demonstrating this combination works better than RAG for long conversations. When retrieval is removed (Fig 3), it appears that under 1M context length where the models were trained, the retrieval module doesn’t benefit much but may harm the performance.

4. Unfair claiming at 10M context length evaluation between LIGHT and Vanilla. Models anyways were not trained to perform at 10M context length. It expected that they perform much worse.

5. It’s unclear whether the author will open-source the data and the framework code. Without these, it’s hard to enable the community to gauge the impact of the work.

**Questions:**

1. Can you provide error analysis showing common failure patterns for each memory ability and also for each evaluated metrics?

2. At 1M context length experiment, LIGHT doen’s seems to outperform RAG at Abstenation, Contradiction Resolution, and Event ordering. Since RAG is similar to the episodic memory module of LIGHT, any reason why this is the case?

---

> ### Author Response · Authors · 2025-11-22
> **Response to Reviewer z6rF - Part 1**
>
> Thank you for your thoughtful review and for recognizing the strengths of our work. We sincerely appreciate the time you spent evaluating the paper and the feedback you provided.
>
> > Limited dataset size: only 100 conversations for benchmarking, this could lead to high variance in evaluation results.
>
> As noted, we created 100 conversations (20 at 128K, 35 at 500K, 35 at 1M, and 10 at 10M), each of which was manually evaluated by humans for quality assurance. However, we believe that focusing solely on the number of conversations does not reflect the full scale of the evaluation. For each conversation, we generated 20 probing questions, resulting in a total of 2000 probing questions, all of which were manually validated by human annotators.
>
> Our scale was also constrained by practical considerations, as each probing question required the development of a corresponding rubric by our annotators. Unlike automated benchmarks, we did high-quality human evaluation and prioritized depth over raw volume, ensuring each of the 2000 probing questions has a verified human-created scoring rubric which significantly reduces noise found in larger, automated datasets.
>
> Finally, our approach is consistent with prior benchmarks. We evaluate models using 2000 probing questions, which compares well to previous work: LongMemEval [1] includes 500 probing questions, MemoryBank [2] includes 194 probing questions, and PerLTQA [3] includes 8593 probing questions. Thus, our benchmark offers a strong and competitive number of probing questions relative to existing work.
>
> > The paper doesn't deeply analyze why models fail on specific memory abilities. For instance, why do all methods struggle with contradiction resolution? No error analysis or qualitative examination of failure modes beyond the single case study in Appendix F.
>
> We added a qualitative error analysis for each memory ability (Appendix G, Page 46).
>
> Regarding contradiction resolution specifically, this ability is inherently challenging because the model must not only retrieve one or more relevant conversation turns but also identify the turns that contradict them.
>
> For LIGHT, one common failure mode occurs when the context contains only one side of the contradiction, leading the LLM to answer based solely on that information. Since the model does not have access to the other side of the story, it cannot detect the conflict. Another common failure occurs when both sides of the contradiction are present in the context, but the LLM still overweighs one side of the contradiction due to position and frequency bias for that side in the context.
>
> > The combination of retrieval + working memory + external memory is not particularly novel. The main contribution is demonstrating this combination works better than RAG for long conversations. When retrieval is removed (Fig 3), it appears that under 1M context length where the models were trained, the retrieval module doesn’t benefit much but may harm the performance.
>
> We agree that some prior works combine retrieval and working memory, such as MemGPT [4] and “Empowering Working Memory for Large Language Model Agents” [5], and that other works focus solely on retrieval, such as MemoryBank [2]. However, to the best of our knowledge, none of the previous work proposes the combination of all three memory components in the way we design them in our LIGHT framework. In LIGHT, we introduce a separate memory component, the scratchpad, which is iteratively updated based on reasoning over the current and preceding dialogue turns, and we do not perform retrieval over the scratchpad.
>
> In addition, the novelty lies in how each memory component is implemented rather than in the high-level labels of the memories themselves. For example, to construct the episodic memory, we extract key–value pairs along with a summary of the interaction and store them in a vector database; the scratchpad is formed through iterative reasoning over the dialogue history; and at inference time, we filter irrelevant content with respect to the query.
>
> Regarding Fig. 3, removing retrieval does impact performance. Removing it at 10M reduces performance by 8.5 percent, at 1M by 2.4 percent, at 500K by 6 percent, and at 100K it does not change the performance but does not improve it either.
>
> Furthermore, as shown in the experiments in Appendix C.3 (Page 31), using a stronger embedding model leads to greater improvements for LIGHT compared to the RAG baseline, demonstrating the robustness and superiority of LIGHT’s episodic memory relative to RAG baseline.

---

> ### Author Response · Authors · 2025-11-22
> **Response to Reviewer z6rF - Part 2**
>
> > Unfair claiming at 10M context length evaluation between LIGHT and Vanilla. Models anyways were not trained to perform at 10M context length. It expected that they perform much worse.
>
> In fact, this is one of the main points of our experiments: to show the limitations of current long-context LLMs. If long-context LLMs are trained at a specific context size, applying them to longer contexts requires truncation, and this demonstrates that they cannot be reliably used beyond their context-window size. At very long lengths, even if the models nominally support that window, they perform much worse than our proposed LIGHT framework.
>
> Also, as noted in Line 431 (footnote), the only model that can support this context length is Llama-4-Scout, which requires a whole cluster with approximately 80 A100 80G GPUs for deployment. We were unable to find any provider that could offer access to this model at that context length.
>
> > It’s unclear whether the author will open-source the data and the framework code. Without these, it’s hard to enable the community to gauge the impact of the work.
>
> Thank you for pointing this out. We will open source the code (including all implementations), the data, and the data generation framework, which will allow users to create long conversations ranging from 100K to beyond 10M tokens based on their own topics and in any domain they choose.
>
> > Can you provide error analysis showing common failure patterns for each memory ability and also for each evaluated metrics?
>
> We added a qualitative error analysis for each memory ability, which is included in Appendix G, Page 46.
>
> > At 1M context length experiment, LIGHT doen’s seems to outperform RAG at Abstenation, Contradiction Resolution, and Event ordering. Since RAG is similar to the episodic memory module of LIGHT, any reason why this is the case?
>
> At 1M context, for event ordering, LIGHT outperforms RAG at 3 out of 4 LLMs, with average of 0.95% improvement.
>
> For contradiction resolution, where LIGHT’s accuracy is lower than RAG by a subtle 0.7 percent, the reason is that LIGHT injects more relevant information into the prompt, and the reader LLM may focus on only one side of the contradiction and fail to consider the conflicting information.
>
> For abstention, the reason RAG outperforms LIGHT is due to limitations of the reader LLM. Because LIGHT is stronger than RAG in integrating more relevant and closely related facts into the context (high recall), the increased presence of related information can cause the reader LLM to provide an answer and be unable to abstain. According to the error analysis, this is one of the main causes of failure for this memory ability.
>
> We hope that our responses resolve the concerns raised, and we appreciate your consideration in the final evaluation

---

> ### Author Response · Authors · 2025-11-22
> **Response to Reviewer z6rF - Part 3**
>
> [1] Wu D, Wang H, Yu W, Zhang Y, Chang KW, Yu D. Longmemeval: Benchmarking chat assistants on long-term interactive memory. arXiv preprint arXiv:2410.10813. 2024 Oct 14.
>
> [2] Zhong W, Guo L, Gao Q, Ye H, Wang Y. Memorybank: Enhancing large language models with long-term memory. InProceedings of the AAAI Conference on Artificial Intelligence 2024 Mar 24 (Vol. 38, No. 17, pp. 19724-19731).
>
> [3] Du Y, Wang H, Zhao Z, Liang B, Wang B, Zhong W, Wang Z, Wong KF. PerLTQA: A personal long-term memory dataset for memory classification, retrieval, and fusion in question answering. InProceedings of the 10th SIGHAN Workshop on Chinese Language Processing (SIGHAN-10) 2024 Aug (pp. 152-164).
>
> [4] Packer C, Fang V, Patil S, Lin K, Wooders S, Gonzalez J. MemGPT: Towards LLMs as Operating Systems.
>
> [5] Guo J, Li N, Qi J, Yang H, Li R, Feng Y, Zhang S, Xu M. Empowering working memory for large language model agents. arXiv preprint arXiv:2312.17259. 2023 Dec 22.

---

### Official Review · Reviewer_WLZS · 2025-11-03

**Soundness:** 3
**Presentation:** 2
**Contribution:** 2
**Rating:** 4
**Confidence:** 4

**Summary:**

The paper makes two main contributions: a BEAM benchmark for evaluating long-term memory in LLMs and LIGHT framework that augments LLMs with extra tools to better handle long-context tasks. The authors provide a number of empirical results, evaluating a number of models on the new benchmark and ablating the new LIGHT framework components.

**Strengths:**

The paper focuses on a highly relevant problem.
The data generation method is sufficiently original.
Experimental evaluation reasonably complements the data-generation effort and the ablation studies help better understand the contributions of different LIGHT framework components.

**Weaknesses:**

My biggest issues with the paper are:

- Insufficient coverage of related work + I'm unsure about how this work fits into the broader research context in this domain.
The paper is positioned as addressing the problem of long-term memory, but is focused solely on conversation-like context. The way the authors brush off existing work seems a bit dismissive, especially since there are many works that do not suffer from the limitations the authors quote as disqualifying.

For example, Narrative QA and Narrative XL datasets do not consist of concatenated unrelated conversations and do require engagement with the material beyond simple retrieval.

The data generation approach is reasonably original, but I do not think that the authors provide a thorough enough justification for why this approach should be taken over the alternatives.

If the paper were explicitly targeting conversation-like usage contexts, it might have been more defensible. But still, when there are many datasets that are far from being solved, a lot of attention needs to be devoted to justify introduction of another one, with thorough proof (both empirical and theoretical) that the new dataset catches LLM performance gaps that can't be investigated using already existing datasets.

I found it especially unfortunate that in the related work, the paper focuses on the efforts in developing long-context models, while the primary focus of the paper is on long-context LLM evaluation & benchmarking. The latter is covered very sparsely.

- Empirical ablation results (e.g. Figure 3) don't seem to have confidence intervals, and, more imporantly, the results are not very consistent or strong. I.e. it seems that the proposed LIGHT model does not consistently improve performance, and that the helpfulness of its specific components depends on context length (and likely on many other factors like the nature of the task that were not investigated in as much depth).

Overall, I feel that the paper proposes a number of very promising ideas, but requires a bit more depth to properly place them in the broader research context and to make the results more reliably interpretable.

**Questions:**

Is the focus of the paper intended to be on conversation-like tasks only, or is the claim broader (long-context LLM application/evaluation in general)?

Minor suggestion: I believe that including a table of illustrative examples of probing questions into the main paper would greatly aid readability. It will clarify the approach for the readers who want to understand the paper without having to read the appendix.

---

> ### Author Response · Authors · 2025-11-22
> **Response to Reviewer WLZS - Part 1**
>
> Thank you for your valuable comments. We sincerely appreciate your time and thoughtful evaluation of our work, as well as your acknowledgement that the paper proposes several promising ideas.
>
> > My biggest issues with the paper are:
> Insufficient coverage of related work + I'm unsure about how this work fits into the broader research context in this domain. The paper is positioned as addressing the problem of long-term memory, but is focused solely on conversation-like context. The way the authors brush off existing work seems a bit dismissive, especially since there are many works that do not suffer from the limitations the authors quote as disqualifying.
> For example, Narrative QA and Narrative XL datasets do not consist of concatenated unrelated conversations and do require engagement with the material beyond simple retrieval.
>
> As you noted, our work focuses specifically on the problem of long-term memory within conversation-style context. We carefully discuss related work and prior benchmarks within the long-term memory, and our benchmark aims to contribute several aspects that prior work does not address, including: (1) extending the context length to an unprecedented 10M tokens, (2) covering multiple domains such as coding, mathematics, and health, (3) evaluating a comprehensive set of memory abilities, including newly introduced abilities, and (4) constructing coherent, natural, and realistic long conversations.
>
> Regarding the focus on conversation-style context: for evaluating long-term memory abilities, such datasets are the most suitable. They simulate real-world user interactions with chatbots, where conversations can naturally become very long, and where long-term memory capabilities are essential. This setting allows us to define a meaningful and rigorous memory-evaluation protocol that depends on continuity of interaction, hence our decision to target conversation-style benchmarks.
>
> In terms of the broader impact, long-term memory capabilities are critical across various applications, including conversational AI, retrieval-augmented systems, and code assistants. Therefore, having a high-quality and reliable benchmark that can accurately measure long-term memory is essential. Likewise, developing methods that enhance long-term memory in LLMs is an important direction for advancing the field.
>
> > The data generation approach is reasonably original, but I do not think that the authors provide a thorough enough justification for why this approach should be taken over the alternatives.
>
> We appreciate your acknowledgement of the originality of our data generation framework. As noted in the paper, we will release the full pipeline so that others can generate conversations tailored to their own domains, topics, and requirements.
>
> The current alternatives to our approaches are: one set of work commonly pursued in prior studies provides a topic and a short background passage to an LLM and instructs it to generate multi-turn dialogues. They have these limitations: (i) the model tends to produce redundant, template-based questions, exhausting topics after a few turns; (ii) the resulting dialogues rarely exceed 10K tokens; and (iii) to extend dialogue length, prior benchmarks concatenate these limited conversations with uncorrelated  or noisy content. Moreover, the latest LLMs output tokens around 128K on average, which means we can’t use this approach at all for generating longer conversations.
>
> To overcome these limitations, we propose our data generation framework, which begins with a conversation seed containing the domain, title and theme, subtopics, narratives that shape the evolving aspects of the conversation, and a user profile with its relationships graph to increase realism. Using this seed, we create conversation plans that serve as a scaffold and outline the storyline, enabling diverse, non-template-based, high-quality user questions. Without such plans, LLM-generated user utterances become template-based and repetitive. With the plans, we generate user utterances, followed by the assistant’s responses. Since realistic conversations also require follow-up questions, we include two controllers: one responsible for generating user follow-ups to the assistant, and one responsible for answering assistant follow-up questions.
>
> This process yields long, coherent, and natural conversations without the common issues of redundancy or disjoint content. As a result, the benchmark provides a more faithful approximation of real-world interactions, leading to a stronger and more reliable evaluation of long-term memory abilities.

---

> ### Author Response · Authors · 2025-11-22
> **Response to Reviewer WLZS - Part 2**
>
> > If the paper were explicitly targeting conversation-like usage contexts, it might have been more defensible. But still, when there are many datasets that are far from being solved, a lot of attention needs to be devoted to justify introduction of another one, with thorough proof (both empirical and theoretical) that the new dataset catches LLM performance gaps that can't be investigated using already existing datasets.
>
> As explained earlier, the focus of our paper is a conversation-style dataset because it is the most suitable setting for evaluating the long-term memory abilities of LLMs. As described previously, our work addresses the limitations of prior benchmarks by proposing a multi-domain, multi-scale dataset that evaluates comprehensive memory abilities and is more realistic to real-world use cases due to the high-quality, coherent, and interactive conversations between the user and the assistant. Our benchmark is also more challenging than existing benchmarks, which justifies the need for introducing it in order to achieve a more accurate, fair, and reliable evaluation of long-term memory abilities in LLMs and long-term memory methods.
>
> > I found it especially unfortunate that in the related work, the paper focuses on the efforts in developing long-context models, while the primary focus of the paper is on long-context LLM evaluation & benchmarking. The latter is covered very sparsely.
>
> We did cover extensive prior work on the evaluation and benchmarking of long-term memory and long-context methods in the related work section (Page 10, Line 519 and Page 16, Line 847), and our intention was to situate our contributions within that line of research. If there are specific articles you believe should be included, we would be glad to incorporate them and improve the related work section accordingly.
>
> > Empirical ablation results (e.g. Figure 3) don't seem to have confidence intervals, and, more imporantly, the results are not very consistent or strong. I.e. it seems that the proposed LIGHT model does not consistently improve performance, and that the helpfulness of its specific components depends on context length (and likely on many other factors like the nature of the task that were not investigated in as much depth).
>
> **Re: confidence intervals**
> We agree with the reviewer that including confidence intervals is valuable and should be encouraged in the literature. Unfortunately, each experimental run requires significant computation resources. We do not have resources available to allow multiple repeated experiments.
>
> **Re: Performance Trends**
> We would like to gently push back on the claim that the performance trends are not consistent.
> Regarding the consistency of LIGHT, we observe that removing the scratchpad, retrieval, or noise-filtering components harms performance across all conversation lengths, indicating that each of these components is important regardless of context size. Similarly, the effectiveness of working memory depends on how many answers to the probing questions fall within the window stored in working memory. This component becomes particularly helpful when the probing questions originate from that range, and removing it at the 10M-token length leads to a 5.7% performance drop.
> Overall, keeping all components of the LIGHT framework together yields clear benefits as the conversation length increases. This robustness to context length is crucial, and at 10M tokens, removing any component leads to a substantial degradation in performance.
>
> **Re: Strong Results**
> Our proposed LIGHT framework performs significantly better than the Long-context LLM and RAG baselines, as well as the additional baseline reported in Appendix C.5 (Page 33). Across four conversation lengths and four LLMs (both closed-source and open-source), LIGHT consistently outperforms the baselines, with only one exception where RAG slightly exceeds LIGHT at the 10M length with Gemini 2 Flash.
> On average, the performance improvements of LIGHT compared to Long-context LLMs, RAG, and ReadAgent are as follows:
> * 100K: 31.51%, 10.71%, 50.97%
> * 500K: 42.78%, 8.35%, 65.44%
> * 1M: 52.11%, 8.47%, 66.12%
> * 10M: 99.85%, 7.75%, 60.81%
>
> These results demonstrate that LIGHT offers strong and consistent gains across models and context lengths.

---

> ### Author Response · Authors · 2025-11-22
> **Response to Reviewer WLZS - Part 3**
>
> > Is the focus of the paper intended to be on conversation-like tasks only, or is the claim broader (long-context LLM application/evaluation in general)?
>
> Our focus is only on conversation-style tasks because they provide a realistic setting for evaluating long-term memory, as they simulate user-chatbot interaction where conversations can naturally extend for long durations, and evaluating long-term memory in this setup is both critical and challenging. This focus is also aligned with real-world systems such as Mem0 and the Alexa Prize SocialBot challenges, which are designed around long, evolving conversational interactions.
>
> > Minor suggestion: I believe that including a table of illustrative examples of probing questions into the main paper would greatly aid readability. It will clarify the approach for the readers who want to understand the paper without having to read the appendix.
>
> Because of the page limit, we placed these examples in the appendix, but in the camera-ready version we will incorporate your valuable and thoughtful suggestion.
>
> We hope the clarifications provided here offer a clearer understanding of the work, and we appreciate your reconsideration of the paper in light of these updates as you complete your final assessment.

---

### Author Response · Authors · 2025-11-28
**Follow-Up on Author Response**

Dear Reviewers,

We sincerely thank you for the time and effort you have invested in reviewing our work and for the constructive comments you provided. We have submitted our responses, and we would greatly appreciate hearing whether our clarifications have satisfactorily addressed your concerns.

We kindly wanted to follow up to ask whether there are any further questions or additional points you would like us to clarify. We truly value your insights and appreciate your consideration during the final evaluation process.

---

### Public Comment · ~Pradyumna_Shyama_Prasad1 · 2025-11-28
**Statistical significance of LIGHT vs RAG improvements**

Thank you for this work on long-context evaluation. The BEAM benchmark addresses a real gap. I have a concern about the method contribution.

At ≤1M tokens: LIGHT outperforms RAG by 1.3–4.2%. These are consistent gains, which is encouraging. However, with n=40 questions per ability at 100K and n=70 at 500K/1M, I was curious whether the authors have verified statistical significance? A rough power analysis suggests per-ability comparisons at these sample sizes may require effect sizes of ~10% to reliably detect. Paired tests or bootstrap confidence intervals would help clarify whether these improvements reflect a true effect.

At 10M tokens: Since vanilla baselines cannot access most of the conversation at this length, the LIGHT vs RAG comparison seems more relevant than the reported gains over vanilla. I noticed the results here are mixed. For instance, Gemini RAG outperforms LIGHT (0.216 vs 0.192). With n=10 conversations at this scale, could this inconsistency suggest high variance?

---

> ### Author Response · Authors · 2025-12-03
> **Authors’ Response**
>
> Thank you for reading and engaging with our paper and finding its impact and importance.
>
> > At ≤1M tokens: LIGHT outperforms RAG by 1.3–4.2%. These are consistent gains, which is encouraging. However, with n=40 questions per ability at 100K and n=70 at 500K/1M, I was curious whether the authors have verified statistical significance? A rough power analysis suggests per-ability comparisons at these sample sizes may require effect sizes of ~10% to reliably detect. Paired tests or bootstrap confidence intervals would help clarify whether these improvements reflect a true effect.
>
> For these lengths, LIGHT achieves relative improvements over RAG of 4.79–15.61%. To assess whether these gains reflect a true effect, we calculated confidence intervals (95%) for the per-ability differences:
>
> 100K: (0.44% - 8.02%)
> 500K: (-0.30% - 5.40%)
> 1M: (5.78% - 11.70%)
>
> These intervals indicate strong and stable statistical improvements of LIGHT over RAG at 100K and 1M. At 500K, though the effect is not as strong, it remains largely positive, consistent, and aligned with the overall improvement trend.
>
> However, as discussed in the error analysis in Appendix G (Page 46), the main failure pattern of LIGHT for abstention arises from limitations of the reader LLM. With stronger memory and better context, the performance on this ability would be lower. Excluding this memory ability, the results become:
>
> 100K: (2.96% - 10.34%)
> 500K: (1.81% - 7.32%)
> 1M: (8.33% - 13.79%)
>
> Which demonstrates a strong and consistent improvement of LIGHT over RAG at all lengths.
>
> > At 10M tokens: Since vanilla baselines cannot access most of the conversation at this length, the LIGHT vs RAG comparison seems more relevant than the reported gains over vanilla. I noticed the results here are mixed. For instance, Gemini RAG outperforms LIGHT (0.216 vs 0.192). With n=10 conversations at this scale, could this inconsistency suggest high variance?
>
> For this length, and as discussed in the error analysis of LIGHT’s failure modes in Appendix G (Page 46), the most common failure arises from limitations of the reader LLM. We therefore attribute the better performance of RAG compared to LIGHT with Gemini to this limitation of the reader LLM rather than to variance.
>
> Variance is also naturally higher at 10M due to the smaller number of conversations. Generating additional 10M-token dialogues is extremely resource-intensive; each 10M conversation is roughly equivalent in cost to producing ~10 conversations at the 1M scale, making larger samples infeasible under our compute and annotation budget.

---

### Author Response · Authors · 2025-12-03
**Summary of Contributions, Discussion, and Revisions for AC**

Dear AC and reviewers,

We would like to sincerely thank the reviewers for the time and effort they dedicated to evaluating our work, for their constructive feedback, and for recognizing the strengths, importance, and potential impact of our contributions. Throughout the rebuttal period, we **applied all reviewer suggestions** in detail and carried out a **substantial number of additional experiments**, **analyses**, and **clarifications**. These updates **strengthened the paper considerably** and directly addressed the concerns raised. The manuscript has been updated accordingly.

---

# Recognized Contributions Across Reviewers
Across reviewers, several contributions and positive aspects of the paper were consistently highlighted:
## Benchmark Contributions (BEAM):
- **Methodology, Scale and Diversity**: Introduces a novel, original, human-validated data-generation method that produces realistic, diverse, multi-domain conversations spanning four context lengths (128K, 500K, 1M, and 10M tokens).
- **Gap-Filling and Difficulty**: Fills gaps in prior single-domain, recall-heavy datasets by evaluating 10 distinct memory abilities and subtasks, creating scenarios that are challenging even for the best existing LLMs.
- **Metrics**: Proposes new complimentary metrics (instruction following, event ordering, and contradiction resolution) to evaluate LLM behavior in long-context settings.
- **Reproducibility**: The paper and appendix provide all necessary details to understand and replicate the dataset creation process.

## Method Contributions (LIGHT):
- **Performance and Scalability**: LIGHT demonstrates convincing, consistent improvements over the strong standard RAG baseline across various models and context scales (100K – 10M).
- **Ablation and Analysis**: Component-wise ablations substantiate that each memory module contributes meaningfully to performance, and the evaluation includes analysis of sensitivity to retrieval budget K.
- **Evaluation Quality**: The experimental evaluation reasonably complements the data-generation effort.

We thank the reviewers for recognizing the novelty and contributions of both BEAM and LIGHT.

---

# Summary of Major Revisions Made in Response to Reviewers Feedback
We summarize the key revisions below:
- Added a **new baseline** (requested by reviewer `dWNe`), reported in Appendix C.5 (Page 33), demonstrating that LIGHT outperforms ReadAgent with a **60.83%** average improvement.
- Added a **new experiment** using a larger embedding model (requested by reviewer `77HX`), in Appendix C.3 (Page 31), confirming the **robustness of LIGHT** to embedding model choice and showing **larger gains compared to RAG**.
- Added an **experiment** on alternative retrieval indexing setups (requested by reviewer `77HX`), in Appendix C.4 (Page 32), showing **robustness of LIGHT** under different indexing configurations.
- Added a **qualitative error analysis** covering all memory abilities (requested by reviewers `dWNe` and `z6rF`), in Appendix G (Page 46), detailing failure patterns of LIGHT.
- Reported latency, memory usage, and GPU budget for LIGHT (reviewer `77HX`).
- Added detailed guidelines and protocols for human annotators (reviewer `77HX`).
- Increased figure sizes and expanded captions for clarity (reviewer `dWNe`).
- Improved readability of the dataset creation section (reviewer `dWNe`).
- Added the suggested related work (reviewer `dWNe`).
- Provided a comprehensive explanation of all human interventions in the dataset creation pipeline (reviewer `dWNe`).
- Clarified the scope and intended focus of the benchmark (reviewer `WLZS`).

---

During the discussion phase, **reviewers were unable to respond to our rebuttal**, and therefore could not acknowledge the extensive new experiments, expanded analyses, and clarifications added during revision. **We kindly ask the AC** to **take these updates into consideration**, as they address the feedback directly and could have positively influenced the reviewers’ scores had the conversation continued.

In light of these improvements, **we respectfully request that the AC** consider the potential upward revision of the reviewers’ evaluations when forming the meta-review. We believe the strengthened experimental results, expanded analysis, and clearer presentation enhance both the rigor and the potential impact of the work, and we hope this will be reflected in the final assessment.

We sincerely appreciate your time and thoughtful consideration.

Best regards, The Authors

---

### Meta-Review · Area_Chair_zFNu · 2026-01-04

**Summary:**

All reviewers gave borderline scores for this work, three 4s and one 6.

Most concerns are related to the quality of the proposed datasets, including 1) the number of conversations [Reviwer z6rF], 2) the representativeness of the conversational long context [eviwer WLZS], 3) the informativeness of evaluations for future development (what errors made by existing LLMs) [Reviwer z6rF, dWNe].

Overall, the authors provide sufficient support during the rebuttal for addressing those major concerns.

Besides the above major conerns, all reviewers agreed on the importance and originality of the proposed dataset for evaluating long-context LLMs.

**Reviewer Concerns:**

In addition to those above major concerns addressed by the rebuttal, the authors' responses also sufficiently addressed other minor concerns regarding the presentation and the experiments carried out for the proposed datasets, including 1) the proposed approach over existing baselines [Reviwer dWNe], 2) the experiment design choice (e.g., embedding model and latency) [Reviwer 77HX].

**Reviewer Scores:**

There are no reviewers' response to authors' rebuttal but I think Reviwer 77HX and dWNE are likely to raise their scores to 6.

---

### Decision · Program_Chairs · 2026-01-26

Accept (Poster)